# Actor-Critic Provably Finds Nash Equilibria of Linear-Quadratic Mean-Field Games

**Zuyue Fu**
Northwestern University
`zuyue.fu@u.northwestern.edu`

**Zhuoran Yang**
Princeton University
`zy6@princeton.edu`

**Yongxin Chen**
Georgia Institute of Technology
`yongchen@gatech.edu`

**Zhaoran Wang**
Northwestern University
`zhaoranwang@gmail.com`

## Abstract

We study discrete-time mean-field Markov games with infinite numbers of agents where each agent aims to minimize its ergodic cost. We consider the setting where the agents have identical linear state transitions and quadratic cost functions, while the aggregated effect of the agents is captured by the population mean of their states, namely, the mean-field state. For such a game, based on the Nash certainty equivalence principle, we provide sufficient conditions for the existence and uniqueness of its Nash equilibrium. Moreover, to find the Nash equilibrium, we propose a mean-field actor-critic algorithm with linear function approximation, which does not require knowing the model of dynamics. Specifically, at each iteration of our algorithm, we use the single-agent actor-critic algorithm to approximately obtain the optimal policy of the each agent given the current mean-field state, and then update the mean-field state. In particular, we prove that our algorithm converges to the Nash equilibrium at a linear rate. To the best of our knowledge, this is the first success of applying model-free reinforcement learning with function approximation to discrete-time mean-field Markov games with provable non-asymptotic global convergence guarantees.

## 1 Introduction

In reinforcement learning (RL) (Sutton and Barto, 2018), an agent learns to make decisions that minimize its expected total cost through sequential interactions with the environment. Multi-agent reinforcement learning (MARL) (Shoham et al., 2003; 2007; Busoniu et al., 2008) aims to extend RL to sequential decision-making problems involving multiple agents. In a non-cooperative game, we are interested in the Nash equilibrium (Nash, 1951), which is a joint policy of all the agents such that each agent cannot decrease its expected total cost by unilaterally deviating from its Nash policy. The Nash equilibrium plays a critical role in understanding the social dynamics of self-interested agents (Ash, 2000; Axtell, 2002) and constructing the optimal policy of a particular agent via fictitious self-play (Bowling and Veloso, 2000; Ganzfried and Sandholm, 2009). With the recent development in deep learning (LeCun et al., 2015), MARL with function approximation achieves tremendous empirical successes in applications, including Go (Silver et al., 2016; 2017), Poker (Heinrich and Silver, 2016; Moravčík et al., 2017), Star Craft (Vinyals et al., 2019), Dota (OpenAI, 2018), autonomous driving (Shalev-Shwartz et al., 2016), multi-robotic systems (Yang and Gu, 2004), and solving social dilemmas (de Cote et al., 2006; Leibo et al., 2017; Hughes et al., 2018). However, since the capacity of the joint state and action spaces grows exponentially in the number of agents, such MARL approaches become computationally intractable when the number of agents is large, which is common in real-world applications (Sandholm, 2010; Calderone, 2017; Wang et al., 2017a).

Mean-field game is proposed by Huang et al. (2003; 2006); Lasry and Lions (2006a;b; 2007) with the idea of utilizing mean-field approximation to model the strategic interactions within a large population. In a mean-field game, each agent has the same cost function and state transition, which depend on the other agents only through their aggregated effect. As a result, the optimal policy of

each agent depends solely on its own state and the aggregated effect of the population, and such an optimal policy is symmetric across all the agents. Moreover, if the aggregated effect of the population corresponds to the Nash equilibrium, then the optimal policy of each agent jointly constitutes a Nash equilibrium. Although such a Nash equilibrium corresponds to an infinite number of agents, it well approximates the Nash equilibrium for a sufficiently large number of agents (Bensoussan et al., 2016). Also, as the aggregated effect of the population abstracts away the strategic interactions between individual agents, it circumvents the computational intractability of the MARL approaches that do not exploit symmetry.

However, most existing work on mean-field games focuses on characterizing the existence and uniqueness of the Nash equilibrium rather than designing provably efficient algorithms. In particular, most existing work considers the continuous-time setting, which requires solving a pair of Hamilton-Jacobi-Bellman (HJB) and Fokker-Planck (FP) equations, whereas the discrete-time setting is more common in practice, e.g., in the aforementioned applications. Moreover, most existing approaches, including the ones based on solving the HJB and FP equations, require knowing the model of dynamics (Bardi and Priuli, 2014), or having the access to a simulator, which generates the next state given any state-action pair and aggregated effect of the population (Guo et al., 2019), which is often unavailable in practice.

To address these challenges, we develop an efficient model-free RL approach to mean-field game, which provably attains the Nash equilibrium. In particular, we focus on discrete-time mean-field games with linear state transitions and quadratic cost functions, where the aggregated effect of the population is quantified by the mean-field state. Such games capture the fundamental difficulties of general mean-field games and well approximates a variety of real-world systems such as power grids (Minciardi and Sacile, 2011), swarm robots (Fang, 2014; Araki et al., 2017; Doerr et al., 2018), and financial systems (Zhou and Li, 2000; Huang and Li, 2018). In detail, based on the Nash certainty equivalence (NCE) principle (Huang et al., 2006; 2007), we propose a mean-field actor-critic algorithm which, at each iteration, given the mean-field state $\mu$, approximately attains the optimal policy $\pi_\mu^*$ of each agent, and then updates the mean-field state $\mu$ assuming that all the agents follow $\pi_\mu^*$. We parametrize the actor and critic by linear and quadratic functions, respectively, and prove that such a parameterization encompasses the optimal policy of each agent. Specifically, we update the actor parameter using policy gradient (Sutton et al., 2000) and natural policy gradient (Kakade, 2002; Peters and Schaal, 2008; Bhatnagar et al., 2009) and update the critic parameter using primal-dual gradient temporal difference (Sutton et al., 2009a;b). In particular, we prove that given the mean-field state $\mu$, the sequence of policies generated by the actor converges linearly to the optimal policy $\pi_\mu^*$. Moreover, when alternatingly update the policy and mean-field state, we prove that the sequence of policies and its corresponding sequence of mean-field states converge to the unique Nash equilibrium at a linear rate. Our approach can be interpreted from both "passive" and "active" perspectives: (i) Assuming that each self-interested agent employs the single-agent actor-critic algorithm, the policy of each agent converges to the unique Nash policy, which characterizes the social dynamics of a large population of model-free RL agents. (ii) For a particular agent, our approach serves as a fictitious self-play method for it to find its Nash policy, assuming the other agents give their best responses. To the best of our knowledge, our work establishes the first efficient model-free RL approach with function approximation that provably attains the Nash equilibrium of a discrete-time mean-field game. As a byproduct, we also show that the sequence of policies generated by the single-agent actor-critic algorithm converges at a linear rate to the optimal policy of a linear-quadratic regulator (LQR) problem in the presence of drift, which may be of independent interest.

**Related Work.** Mean-field game is first introduced in Huang et al. (2003; 2006); Lasry and Lions (2006a;b; 2007). In the last decade, there is growing interest in understanding continuous-time mean-field games. See, e.g., Guéant et al. (2011); Bensoussan et al. (2013); Gomes et al. (2014); Carmona and Delarue (2013; 2018) and the references therein. Due to their simple structures, continuous-time linear-quadratic mean-field games are extensively studied under various model assumptions. See Li and Zhang (2008); Bardi (2011); Wang and Zhang (2012); Bardi and Priuli (2014); Huang et al. (2016a;b); Bensoussan et al. (2016; 2017); Caines and Kizilkale (2017); Huang and Huang (2017); Moon and Başar (2018); Huang and Zhou (2019) for examples of this line of work. Meanwhile, the literature on discrete-time linear-quadratic mean-field games remains relatively scarce. Most of this line of work focuses on characterizing the existence of a Nash equilibrium and the behavior of such a Nash equilibrium when the number of agents goes to infinity

(Gomes et al., 2010; Tembine and Huang, 2011; Moon and Başar, 2014; Biswas, 2015; Saldi et al., 2018a;b; 2019). See also Yang et al. (2018a), which applies maximum entropy inverse RL (Ziebart et al., 2008) to infer the cost function and social dynamics of discrete-time mean-field games with finite state and action spaces. Our work is most related to Guo et al. (2019), where they propose a mean-field Q-learning algorithm (Watkins and Dayan, 1992) for discrete-time mean-field games with finite state and action spaces. Such an algorithm requires the access to a simulator, which, given any state-action pair and mean-field state, outputs the next state. In contrast, both our state and action spaces are infinite, and we do not require such a simulator but only observations of trajectories under given mean-field state. Correspondingly, we study the mean-field actor-critic algorithm with linear function approximation, whereas their algorithm is tailored to the tabular setting. Also, our work is closely related to Mguni et al. (2018), which focuses on a more restrictive setting where the state transition does not involve the mean-field state. In such a setting, mean-field games are potential games, which is, however, not true in more general settings (Li et al., 2017; Briani and Cardaliaguet, 2018). In comparison, we allow the state transition to depend on the mean-field state. Meanwhile, they propose a fictitious self-play method based on the single-agent actor-critic algorithm and establishes its asymptotic convergence. However, their proof of convergence relies on the assumption that the single-agent actor-critic algorithm converges to the optimal policy, which is unverified therein. In addition, our work is related to Jayakumar and Aditya (2019), where the proposed algorithm is only shown to converge asymptotically to a stationary point of the mean-field game.

Our work also extends the line of work on finding the Nash equilibria of Markov games using MARL. Due to the computational intractability introduced by the large number of agents, such a line of work focuses on finite-agent Markov games (Littman, 1994; 2001; Hu and Wellman, 1998; Bowling, 2001; Lagoudakis and Parr, 2002; Hu and Wellman, 2003; Conitzer and Sandholm, 2007; Perolat et al., 2015; Pérolat et al., 2016b;a; 2018; Wei et al., 2017; Zhang et al., 2018; Zou et al., 2019; Casgrain et al., 2019). See also Shoham et al. (2003; 2007); Busoniu et al. (2008); Li (2018) for detailed surveys. Our work is related to Yang et al. (2018b), where they combine the mean-field approximation of actions (rather than states) and Nash Q-learning (Hu and Wellman, 2003) to study general-sum Markov games with a large number of agents. However, the Nash Q-learning algorithm is only applicable to finite state and action spaces, and its convergence is established under rather strong assumptions. Also, when the number of agents goes to infinity, their approach yields a variant of tabular Q-learning, which is different from our mean-field actor-critic algorithm.

For policy optimization, based on the policy gradient theorem, Sutton et al. (2000); Konda and Tsitsiklis (2000) propose the actor-critic algorithm, which is later generalized to the natural actor-critic algorithm (Peters and Schaal, 2008; Bhatnagar et al., 2009). Most existing results on the convergence of actor-critic algorithms are based on stochastic approximation using ordinary differential equations (Bhatnagar et al., 2009; Castro and Meir, 2010; Konda and Tsitsiklis, 2000; Maei, 2018), which are asymptotic in nature. For policy evaluation, the convergence of primal-dual gradient temporal difference is studied in Liu et al. (2015); Du et al. (2017); Wang et al. (2017b); Yu (2017); Wai et al. (2018). However, this line of work assumes that the feature mapping is bounded, which is not the case in our setting. Thus, the existing convergence results are not applicable to analyzing the critic update in our setting. To handle the unbounded feature mapping, we utilize a truncation argument, which requires more delicate analysis.

Finally, our work extends the line of work that studies model-free RL for LQR. For example, Bradtke (1993); Bradtke et al. (1994) show that policy iteration converges to the optimal policy, Tu and Recht (2017); Dean et al. (2017) study the sample complexity of least-squares temporal-difference for policy evaluation. More recently, Fazel et al. (2018); Malik et al. (2018); Tu and Recht (2018) show that the policy gradient algorithm converges at a linear rate to the optimal policy. See as also Hardt et al. (2016); Dean et al. (2018) for more in this line of work. Our work is also closely related to Yang et al. (2019), where they show that the sequence of policies generated by the natural actor-critic algorithm enjoys a linear rate of convergence to the optimal policy. Compared with this work, when fixing the mean-field state, we use the actor-critic algorithm to study LQR in the presence of drift, which introduces significant difficulties in the analysis. As we show in §3, the drift causes the optimal policy to have an additional intercept, which makes the state- and action-value functions more complicated.

**Notations.** We denote by $\|M\|_*$ the spectral norm, $\rho(M)$ the spectral radius, $\sigma_{\min}(M)$ the minimum singular value, and $\sigma_{\max}(M)$ the maximum singular value of a matrix $M$. We use $\|\alpha\|_2$ to represent the $\ell_2$-norm of a vector $\alpha$, and $(\alpha)_i^j$ to denote the sub-vector $(\alpha_i, \alpha_{i+1}, \ldots, \alpha_j)^\top$, where $\alpha_k$ is the $k$-th entry of the vector $\alpha$. For scalars $a_1, \ldots, a_n$, we denote by $\text{poly}(a_1, \ldots, a_n)$ the polynomial of $a_1, \ldots, a_n$, and this polynomial may vary from line to line. We use $[n]$ to denote the set $\{1, 2, \ldots, n\}$ for any $n \in \mathbb{N}$.

## 2 LINEAR-QUADRATIC MEAN-FIELD GAME

A linear-quadratic mean-field $N_a$-player game involves $N_a \in \mathbb{N}$ agents, whose state transitions are given by

$$x_{t+1}^i = Ax_t^i + Bu_t^i + \overline{A}\overline{x}_t + d^i + \omega_t^i, \qquad \forall t \geq 0, \ i \in [N_a].$$

Here $A \in \mathbb{R}^{m \times m}$, $B \in \mathbb{R}^{m \times k}$, and $\overline{A} \in \mathbb{R}^{m \times m}$ are matrices, $x_t^i \in \mathbb{R}^m$ and $u_t^i \in \mathbb{R}^k$ are the state and action vectors of agent $i$, respectively, the vector $d^i \in \mathbb{R}^m$ is a drift term, $\omega_t^i \in \mathbb{R}^m$ is an independent random noise term following the Gaussian distribution $\mathcal{N}(0, \Psi_\omega)$, and $\overline{x}_t = 1/N_a \cdot \sum_{j=1}^{N_a} x_t^j$ is the mean-field state. The agents are coupled through the mean-field state $\overline{x}_t$. In the linear-quadratic mean-field $N_a$-player game, the cost of agent $i \in [N_a]$ at time $t \geq 0$ is given by

$$c_t^i = (x_t^i)^\top Q x_t^i + (u_t^i)^\top R u_t^i + \overline{x}_t^\top \overline{Q}\overline{x}_t,$$

where $Q \in \mathbb{R}^{m \times m}$, $R \in \mathbb{R}^{k \times k}$, and $\overline{Q} \in \mathbb{R}^{m \times m}$ are matrices, and $u_t^i$ is generated by $\pi^i$, i.e., the policy of agent $i$. To measure the performance of agent $i$ following its policy $\pi^i$ under the influence of the other agents, we define the expected total cost of agent $i$ as

$$J^i(\pi^1, \pi^2, \ldots, \pi^{N_a}) = \lim_{T \to \infty} \mathbb{E}\left(\frac{1}{T}\sum_{t=0}^T c_t^i\right).$$

We are interested in finding a Nash equilibrium $(\pi^1, \pi^2, \ldots, \pi^{N_a})$, which is defined by

$$J^i(\pi^1, \ldots, \pi^{i-1}, \pi^i, \pi^{i+1}, \ldots, \pi^{N_a}) \leq J^i(\pi^1, \ldots, \pi^{i-1}, \widetilde{\pi}^i, \pi^{i+1}, \ldots, \pi^{N_a}), \qquad \forall \widetilde{\pi}^i, \ i \in [N_a].$$

That is, agent $i$ cannot further decrease its expected total cost by unilaterally deviating from its Nash policy.

For the simplicity of discussion, we assume that the drift term $d^i$ is identical for each agent. We consider taking the infinite-population limit $N_a \to \infty$, where each agent has an infinitesimal contribution to the dynamics of the system. Thus, the joint policy of all the agents except agent $i$ can be modeled as a mean-field policy $\pi^\dagger$, and all the agents following such a mean-field policy $\pi^\dagger$ generate the mean-field state $\mathbb{E}x_t^\dagger$, where $\{x_t^\dagger\}_{t \geq 0}$ is generated following the policy $\pi^\dagger$. By the symmetry of the agents in terms of their state transitions and cost functions, we focus on a fixed agent and drop the superscript $i$ hereafter.

Before we formally present the formulation of linear-quadratic mean-field games, we first introduce the following mean-field LQR (MF-LQR) problem, which aims to find an optimal policy for the fixed agent given the mean-field policy $\pi^\dagger$.

**Problem 2.1** (MF-LQR). Given the mean-field policy $\pi^\dagger$, we consider the following formulation,

$$x_{t+1} = Ax_t + Bu_t + \overline{A}\mathbb{E}x_t^\dagger + d + \omega_t,$$
$$c(x_t, u_t) = x_t^\top Q x_t + u_t^\top R u_t + (\mathbb{E}x_t^\dagger)^\top \overline{Q}(\mathbb{E}x_t^\dagger),$$
$$J(\pi, \pi^\dagger) = \lim_{T \to \infty} \mathbb{E}\left[\frac{1}{T}\sum_{t=0}^T c(x_t, u_t)\right],$$

where $x_t \in \mathbb{R}^m$ is the state vector, $u_t \in \mathbb{R}^k$ is the action vector generated by the policy $\pi$, $\{x_t^\dagger\}_{t \geq 0}$ is the trajectory generated by the policy $\pi^\dagger$, $\omega_t \in \mathbb{R}^m$ is an independent random noise term following the Gaussian distribution $\mathcal{N}(0, \Psi_\omega)$, and $d \in \mathbb{R}^m$ is a drift term. Here the expectation $\mathbb{E}x_t^\dagger$ is taken across all the agents. We aim to find $\pi^*$ such that $J(\pi^*, \pi^\dagger) = \inf_{\pi \in \Pi} J(\pi, \pi^\dagger)$.

Note that a controllable linear system using linear quadratic optimal control is always stable. Further combining the fact that our linear closed-loop dynamics in Problem 2.1 is driven by the Gaussian noise term $\omega_t$, we know that the Markov chain of states generated by the policy $\pi^\dagger$ admits a stationary distribution and converges to this stationary distribution. This implies that the mean-field state $\mathbb{E}x_t^\dagger$ converges to a constant vector $\mu^\dagger$ as $t \to \infty$, which serves as a time-invariant mean-field state. As we consider the ergodic setting, it then suffices to study Problem 2.1 with $t$ sufficiently large. Therefore, the influence of the mean-field policy $\pi^\dagger$ is captured by the mean-field state $\mu^\dagger$. By re-formulating Problem 2.1, with slight abuse of notations, we obtain the following drifted-LQR (D-LQR).

**Problem 2.2** (D-LQR). Given a mean-field state $\mu \in \mathbb{R}^m$, we consider the following formulation,

$$x_{t+1} = Ax_t + Bu_t + \overline{A}\mu + d + \omega_t,$$
$$c_\mu(x_t, u_t) = x_t^\top Q x_t + u_t^\top R u_t + \mu^\top \overline{Q}\mu,$$
$$J_\mu(\pi) = \lim_{T \to \infty} \mathbb{E}\left[\frac{1}{T}\sum_{t=0}^{T} c_\mu(x_t, u_t)\right],$$

where $x_t \in \mathbb{R}^m$ is the state vector, $u_t \in \mathbb{R}^k$ is the action vector generated by the policy $\pi$, $\omega_t \in \mathbb{R}^m$ is an independent random noise term following the Gaussian distribution $\mathcal{N}(0, \Psi_\omega)$, and $d \in \mathbb{R}^m$ is a drift term. We aim to find an optimal policy $\pi_\mu^*$ such that $J_\mu(\pi_\mu^*) = \inf_{\pi \in \Pi} J_\mu(\pi)$.

Compared with the most studied LQR problem (Lewis et al., 2012), both the state transition and the cost function in Problem 2.2 have drift terms, which act as the mean-field "force" that drives the states away from zero. Such a mean-field "force" introduces additional challenges when solving Problem 2.2 in the model-free setting (see §3.3 for details). On the other hand, the unique optimal policy $\pi_\mu^*$ of Problem 2.2 admits a linear form $\pi_\mu^*(x_t) = -K_{\pi_\mu^*}x_t + b_{\pi_\mu^*}$ under certain regularity conditions (Anderson and Moore, 2007), where the matrix $K_{\pi_\mu^*} \in \mathbb{R}^{k \times m}$ and the vector $b_{\pi_\mu^*} \in \mathbb{R}^k$ are the parameters of $\pi_\mu^*$. Motivated by such a linear form of the optimal policy, we define the class of linear-Gaussian policies as

$$\Pi = \{\pi(x) = -K_\pi x + b_\pi + \sigma \cdot \eta \colon K_\pi \in \mathbb{R}^{k \times m}, b_\pi \in \mathbb{R}^k\}, \tag{2.1}$$

where $\sigma \in \mathbb{R}$ and the standard Gaussian noise term $\eta \in \mathbb{R}^k$ is included to encourage exploration. To solve Problem 2.2, it suffices to find the optimal policy $\pi_\mu^*$ within $\Pi$. We define $\Lambda_1(\mu) = \pi_\mu^*$ as the optimal policy under the mean-field state $\mu$.

Assume that all the agents follow the linear policy $\pi(x) = -K_\pi x + b_\pi$ under the mean-field state $\mu$. By plugging $u_t = \pi(x_t)$ into the state transition in Problem 2.2, as $t \to \infty$, we know that these agents generate a new mean-field state $\mu_{\text{new}}$ such that

$$\mu_{\text{new}} = (I - A + BK_\pi)^{-1}(Bb_\pi + \overline{A}\mu + d).$$

We define $\Lambda_2(\mu, \pi) = \mu_{\text{new}}$ as such a new mean-field state.

Now, we are ready to present the following linear-quadratic mean-field game (LQ-MFG).

**Problem 2.3** (LQ-MFG). We consider the following formulation,

$$x_{t+1} = Ax_t + Bu_t + \overline{A}\mu + d + \omega_t,$$
$$c(x_t, u_t) = x_t^\top Q x_t + u_t^\top R u_t + \mu^\top \overline{Q}\mu,$$
$$J(\pi, \mu) = \lim_{T \to \infty} \mathbb{E}\left[\frac{1}{T}\sum_{t=0}^{T} c(x_t, u_t)\right],$$

where $x_t \in \mathbb{R}^m$ is the state vector, $u_t \in \mathbb{R}^k$ is the action vector generated by the policy $\pi$, $\mu \in \mathbb{R}^m$ is the mean-field state, $\omega_t \in \mathbb{R}^m$ is an independent random noise term following the Gaussian distribution $\mathcal{N}(0, \Psi_\omega)$, and $d \in \mathbb{R}^m$ is a drift term. We aim to find a pair $(\mu^*, \pi^*)$ such that (i) $J(\pi^*, \mu^*) = \inf_{\pi \in \Pi} J(\pi, \mu^*)$; (ii) $\mathbb{E}x_t^*$ converges to $\mu^*$ as $t \to \infty$, where $\{x_t^*\}_{t \geq 0}$ is the Markov chain of states generated by the policy $\pi^*$.

The formulation in Problem 2.3 is studied by Lasry and Lions (2007); Bensoussan et al. (2016); Saldi et al. (2018a;b). We propose a more general formulation in Problem C.2 (see §C of the appendix for

details), where an additional interaction term between the state vector $x_t$ and the mean-field state $\mu$ is incorporated into the cost function. According to our analysis in §C, up to minor modification, the results in the following sections also carry over to Problem C.2. Therefore, for the sake of simplicity, we focus on Problem 2.3 in the sequel.

In Problem 2.3, condition (i) is equivalent to the optimality of the policy $\pi^*$ under the mean-field state $\mu^*$, namely, $\Lambda_1(\mu^*) = \pi^*$. Meanwhile, condition (ii) is equivalent to the invariance of the mean-field state $\mu^*$ given the policy $\pi^*$, namely, $\Lambda_2(\mu^*, \pi^*) = \mu^*$. Such equivalence follows from the NCE principle (Huang et al., 2006; 2007), which also motivates the following definition of the Nash equilibrium pair (Saldi et al., 2018a;b).

**Definition 2.4** (Nash Equilibrium Pair). The pair $(\mu^*, \pi^*) \in \mathbb{R}^m \times \Pi$ constitutes a Nash equilibrium pair of Problem 2.3 if it satisfies $\pi^* = \Lambda_1(\mu^*)$ and $\mu^* = \Lambda_2(\mu^*, \pi^*)$. Here $\mu^*$ is called the Nash mean-field state and $\pi^*$ is called the Nash policy.

By Definition 2.4, Problem 2.3 aims to find a Nash equilibrium pair $(\mu^*, \pi^*)$.

## 3 MEAN-FIELD ACTOR-CRITIC

We first characterize the existence and uniqueness of the Nash equilibrium pair of Problem 2.3 under mild regularity conditions, and then propose a mean-field actor-critic algorithm to obtain such a Nash equilibrium. As a building block of the mean-field actor-critic, we propose the natural actor-critic to solve Problem 2.2.

### 3.1 EXISTENCE AND UNIQUENESS OF NASH EQUILIBRIUM PAIR

We now establish the existence and uniqueness of the Nash equilibrium pair defined in Definition 2.4. We impose the following regularity conditions.

**Assumption 3.1.** We assume that the following statements hold:

(i) The algebraic Riccati equation $X = A^\top X A + Q - A^\top X B (B^\top X B + R)^{-1} B^\top X A$ admits a unique symmetric positive definite solution $X^*$;

(ii) It holds for $L_0 = L_1 L_3 + L_2$ that $L_0 < 1$, where

$$L_1 = \left\| \left[ (I - A)Q^{-1}(I - A)^\top + B R^{-1} B^\top \right]^{-1} \overline{A} \right\|_2 \cdot \left\| \left[ K^* Q^{-1}(I - A)^\top - R^{-1} B^\top \right] \right\|_2,$$

$$L_2 = \left[ 1 - \rho(A - BK^*) \right]^{-1} \cdot \|\overline{A}\|_2, \qquad L_3 = \left[ 1 - \rho(A - BK^*) \right]^{-1} \cdot \|B\|_2.$$

Here $K^* = -(B^\top X^* B + R)^{-1} B^\top X^* A$.

The first assumption is implied by mild regularity conditions on the matrices $A$, $B$, $Q$, and $R$, which are (1) the positivity of $R$; (2) the non-negativity of $Q = C^\top C$; (3) the observability of $(A, C)$; (4) the stability of $(A, B)$. See De Souza et al. (1986); Lewis et al. (2012) for more details. The second assumption is standard in the literature (Bensoussan et al., 2016; Saldi et al., 2018b), which ensures the stability of the LQ-MFG. In the following proposition, we show that Problem 2.3 admits a unique Nash equilibrium pair.

**Proposition 3.2** (Existence and Uniqueness of Nash Equilibrium Pair). Under Assumption 3.1, the operator $\Lambda(\cdot) = \Lambda_2(\cdot, \Lambda_1(\cdot))$ is $L_0$-Lipschitz, where $L_0$ is given in Assumption 3.1. Moreover, there exists a unique Nash equilibrium pair $(\mu^*, \pi^*)$ of Problem 2.3.

*Proof.* See §E.1 for a detailed proof. □

### 3.2 MEAN-FIELD ACTOR-CRITIC FOR LQ-MFG

The NCE principle motivates a fixed-point approach to solve Problem 2.3, which generates a sequence of policies $\{\pi_s\}_{s \geq 0}$ and mean-field states $\{\mu_s\}_{s \geq 0}$ satisfying the following two properties: (i) Given the mean-field state $\mu_s$, the policy $\pi_s$ is optimal. (ii) The mean-field state becomes $\mu_{s+1}$ as $t \to \infty$, if all the agents follow $\pi_s$ under the current mean-field state $\mu_s$. Here (i) requires solving Problem 2.2 given the mean-field state $\mu_s$, while (ii) requires simulating the agents following the

---

**Algorithm 1** Mean-Field Actor-Critic for solving LQ-MFG.

---

1: **Input:**
  - Initial mean-field state $\mu_0$ and Initial policy $\pi_0$ with parameters $K_0$ and $b_0$.
  - Numbers of iterations $S$, $\{N_s\}_{s\in[S]}$, $\{H_s\}_{s\in[S]}$, $\{\widetilde{T}_{s,n}, T_{s,n}\}_{s\in[S],n\in[N_s]}$, $\{\widetilde{T}^b_{s,h}, T^b_{s,h}\}_{s\in[S],h\in[H_s]}$.
  - Stepsizes $\{\gamma_s\}_{s\in[S]}$, $\{\gamma^b_s\}_{s\in[S]}$, $\{\gamma_{s,n,t}\}_{s\in[S],n\in[N_s],t\in[T_{s,n}]}$, $\{\gamma^b_{s,h,t}\}_{s\in[S],h\in[H_s],t\in[T^b_{s,h}]}$.
2: **for** $s = 0, 1, 2, \ldots, S-1$ **do**
3:   **Policy Update:** Solve for the optimal policy $\pi_{s+1}$ with parameters $K_{s+1}$ and $b_{s+1}$ of Problem 2.2 via Algorithm 2 with $\mu_s$, $\pi_s$, $N_s$, $H_s$, $\{\widetilde{T}_{s,n}, T_{s,n}\}_{n\in[N_s]}$, $\{\widetilde{T}^b_{s,h}, T^b_{s,h}\}_{h\in[H_s]}$, $\gamma_s$, $\gamma^b_s$, $\{\gamma_{s,n,t}\}_{n\in[N_s],t\in[T_{s,n}]}$, and $\{\gamma^b_{s,h,t}\}_{h\in[H_s],t\in[T^b_{s,h}]}$, which gives the estimated mean-field state $\widehat{\mu}_{K_{s+1},b_{s+1}}$.
4:   **Mean-Field State Update:** Update the mean-field state via $\mu_{s+1} \leftarrow \widehat{\mu}_{K_{s+1},b_{s+1}}$.
5: **end for**
6: **Output:** Pair $(\pi_S, \mu_S)$.

---

policy $\pi_s$ given the current mean-field $\mu_s$. Based on such properties, we propose the mean-field actor-critic in Algorithm 1.

Algorithm 1 requires solving Problem 2.2 at each iteration to obtain $\pi_s = \Lambda_1(\mu_s)$ and $\mu_{s+1} = \Lambda_2(\mu_s, \pi_s)$. To this end, we introduce the natural actor-critic in §3.3 that solves Problem 2.2.

### 3.3   NATURAL ACTOR-CRITIC FOR D-LQR

Now we focus on solving Problem 2.2 for a fixed mean-field state $\mu$, we thus drop the subscript $\mu$ hereafter. With slight abuse of notations, we write $\pi_{K,b}(x) = -Kx + b + \sigma \cdot \eta$ to emphasize the dependence on $K$ and $b$, and $J(K,b) = J(\pi_{K,b})$ consequently. Now, we propose the natural actor-critic to solve Problem 2.2.

For any policy $\pi_{K,b} \in \Pi$, by the state transition in Problem 2.2, we have

$$x_{t+1} = (A - BK)x_t + (Bb + \overline{A}\mu + d) + \epsilon_t, \qquad \epsilon_t \sim \mathcal{N}(0, \Psi_\epsilon), \tag{3.1}$$

where $\Psi_\epsilon = \sigma BB^\top + \Psi_\omega$. It is known that if $\rho(A - BK) < 1$, then the Markov chain $\{x_t\}_{t\geq 0}$ induced by (3.1) has a unique stationary distribution $\mathcal{N}(\mu_{K,b}, \Phi_K)$ (Anderson and Moore, 2007), where the mean-field state $\mu_{K,b}$ and the covariance $\Phi_K$ satisfy that

$$\mu_{K,b} = (I - A + BK)^{-1}(Bb + \overline{A}\mu + d), \tag{3.2}$$

$$\Phi_K = (A - BK)\Phi_K(A - BK)^\top + \Psi_\epsilon. \tag{3.3}$$

Meanwhile, the Bellman equation for Problem 2.2 takes the following form

$$P_K = (Q + K^\top RK) + (A - BK)^\top P_K(A - BK). \tag{3.4}$$

Then by calculation (see Proposition B.2 in §B.1 of the appendix for details), it holds that the expected total cost $J(K,b)$ is decomposed as

$$J(K,b) = J_1(K) + J_2(K,b) + \sigma^2 \cdot \text{tr}(R) + \mu^\top \overline{Q}\mu, \tag{3.5}$$

where $J_1(K)$ and $J_2(K,b)$ are defined as

$$J_1(K) = \text{tr}\big[(Q + K^\top RK)\Phi_K\big] = \text{tr}(P_K \Psi_\epsilon),$$

$$J_2(K,b) = \begin{pmatrix} \mu_{K,b} \\ b \end{pmatrix}^\top \begin{pmatrix} Q + K^\top RK & -K^\top R \\ -RK & R \end{pmatrix} \begin{pmatrix} \mu_{K,b} \\ b \end{pmatrix}. \tag{3.6}$$

Here $J_1(K)$ is the expected total cost in the most studied LQR problems (Yang et al., 2019; Fazel et al., 2018), where the state transition does not have drift terms. Meanwhile, $J_2(K,b)$ corresponds to the expected cost induced by the drift terms. The following two propositions characterize the properties of $J_2(K,b)$.

First, we show that $J_2(K,b)$ is strongly convex in $b$.

**Proposition 3.3.** Given any $K$, the function $J_2(K, b)$ is $\nu_K$-strongly convex in $b$. Here $\nu_K = \sigma_{\min}(Y_{1,K}^\top Y_{1,K} + Y_{2,K}^\top Y_{2,K})$, where $Y_{1,K} = R^{1/2}K(I - A + BK)^{-1}B - R^{1/2}$ and $Y_{2,K} = Q^{1/2}(I - A + BK)^{-1}B$. Also, $J_2(K, b)$ has $\iota_K$-Lipschitz continuous gradient in $b$, where $\iota_K$ is upper bounded as $\iota_K \leq [1 - \rho(A - BK)]^{-2} \cdot (\|B\|_2^2 \cdot \|K\|_2^2 \cdot \|R\|_2 + \|B\|_2^2 \cdot \|Q\|_2)$.

*Proof.* See §E.4 for a detailed proof. □

Second, we show that $\min_b J_2(K, b)$ is independent of $K$.

**Proposition 3.4.** We define $b^K = \operatorname{argmin}_b J_2(K, b)$, where $J_2(K, b)$ is defined in (3.6). It holds that

$$b^K = \left[ KQ^{-1}(I - A)^\top - R^{-1}B^\top \right] \cdot \left[ (I - A)Q^{-1}(I - A)^\top + BR^{-1}B^\top \right]^{-1} \cdot (\overline{A}\mu + d).$$

Moreover, $J_2(K, b^K)$ takes the form of

$$J_2(K, b^K) = (\overline{A}\mu + d)^\top \left[ (I - A)Q^{-1}(I - A)^\top + BR^{-1}B^\top \right]^{-1} \cdot (\overline{A}\mu + d),$$

which is independent of $K$.

*Proof.* See §E.2 for a detailed proof. □

Since $\min_b J_2(K, b)$ is independent of $K$ by Proposition 3.4, it holds that the optimal $K^*$ is the same as $\operatorname{argmin}_K J_1(K)$. This motivates us to minimize $J(K, b)$ by first updating $K$ following the gradient direction $\nabla_K J_1(K)$ to the optimal $K^*$, then updating $b$ following the gradient direction $\nabla_b J_2(K^*, b)$. We now design our algorithm based on this idea.

We define $\Upsilon_K$, $p_{K,b}$, and $q_{K,b}$ as

$$\Upsilon_K = \begin{pmatrix} Q + A^\top P_K A & A^\top P_K B \\ B^\top P_K A & R + B^\top P_K B \end{pmatrix} = \begin{pmatrix} \Upsilon_K^{11} & \Upsilon_K^{12} \\ \Upsilon_K^{21} & \Upsilon_K^{22} \end{pmatrix},$$

$$p_{K,b} = A^\top \left[ P_K \cdot (\overline{A}\mu + d) + f_{K,b} \right], \qquad q_{K,b} = B^\top \left[ P_K \cdot (\overline{A}\mu + d) + f_{K,b} \right], \qquad (3.7)$$

where $f_{K,b} = (I - A + BK)^{-\top}[(A - BK)^\top P_K(Bb + \overline{A}\mu + d) - K^\top Rb]$. By calculation (see Proposition B.3 in §B.1 of the appendix for details), the gradients of $J_1(K)$ and $J_2(K, b)$ take the forms of

$$\nabla_K J_1(K) = 2(\Upsilon_K^{22}K - \Upsilon_K^{21}) \cdot \Phi_K, \qquad \nabla_b J_2(K, b) = \Upsilon_K^{22}(-K\mu_{K,b} + b) + \Upsilon_K^{21}\mu_{K,b} + q_{K,b}.$$

Our algorithm follows the natural actor-critic method (Bhatnagar et al., 2009) and actor-critic method (Konda and Tsitsiklis, 2000). Specifically, (i) To obtain the optimal $K^*$, in the critic update step, we estimate the matrix $\Upsilon_K$ by $\widehat{\Upsilon}_K$ via a policy evaluation algorithm, e.g., Algorithm 3 or Algorithm 4 (see §B.2 and §B.3 of the appendix for details); in the actor update step, we update $K$ via $K \leftarrow K - \gamma \cdot (\widehat{\Upsilon}_K^{22}K - \widehat{\Upsilon}_K^{21})$, where the term $\widehat{\Upsilon}_K^{22}K - \widehat{\Upsilon}_K^{21}$ is the estimated natural gradient. (ii) To obtain the optimal $b^*$ given $K^*$, in the critic update step, we estimate $\Upsilon_{K^*}$, $q_{K^*,b}$, and $\mu_{K^*,b}$ by $\widehat{\Upsilon}_{K^*}$, $\widehat{q}_{K^*,b}$, and $\widehat{\mu}_{K^*,b}$ via a policy evaluation algorithm; In the actor update step, we update $b$ via $b \leftarrow b - \gamma \cdot \widehat{\nabla}_b J_2(K^*, b)$, where $\widehat{\nabla}_b J_2(K^*, b) = \widehat{\Upsilon}_{K^*}^{22}(-K^*\widehat{\mu}_{K^*,b} + b) + \widehat{\Upsilon}_{K^*}^{21}\widehat{\mu}_{K^*,b} + \widehat{q}_{K^*,b}$ is the estimated gradient. Combining the above procedure, we obtain the natural actor-critic for Problem 2.2, which is stated in Algorithm 2.

One may want to apply gradient method to $J(K, b)$ directly in the joint space of $K$ and $b$. However, the gradient dominance property of $J_1(K)$ in the most studied LQR problem (Yang et al., 2019) no longer holds for $J(K, b)$. Therefore, the convergence of the gradient method to $J(K, b)$ is not guaranteed in our problem.

## 4 GLOBAL CONVERGENCE RESULTS

The following theorem establishes the rate of convergence of Algorithm 1 to the Nash equilibrium pair $(\mu^*, \pi^*)$ of Problem 2.3.

---

**Algorithm 2** Natural Actor-Critic Algorithm for D-LQR.
---
1: **Input:**
- Mean-field state $\mu$ and initial policy $\pi_{K_0,b_0}$.
- Numbers of iterations $N$, $H$, $\{\widetilde{T}_n, T_n\}_{n \in [N]}$, $\{\widetilde{T}_h^b, T_h^b\}_{h \in [H]}$.
- Stepsizes $\gamma, \gamma^b, \{\gamma_{n,t}\}_{n \in [N], t \in [T_n]}, \{\gamma_{h,t}^b\}_{h \in [H], t \in [T_h^b]}$.
2: **for** $n = 0, 1, 2, \ldots, N - 1$ **do**
3:      **Critic Update:** Compute $\widehat{\Upsilon}_{K_n}$ via Algorithm 3 with $\pi_{K_n,b_0}, \mu, \widetilde{T}_n, T_n, \{\gamma_{n,t}\}_{t \in [T_n]}, K_0$, and $b_0$ as inputs.
4:      **Actor Update:** Update the parameter via

$$K_{n+1} \leftarrow K_n - \gamma \cdot (\widehat{\Upsilon}_{K_n}^{22} K_n - \widehat{\Upsilon}_{K_n}^{21}).$$

5: **end for**
6: **for** $h = 0, 1, 2, \ldots, H - 1$ **do**
7:      **Critic Update:** Compute $\widehat{\mu}_{K_N,b_h}, \widehat{\Upsilon}_{K_N}, \widehat{q}_{K_N,b_h}$ via Algorithm 3 with $\pi_{K_N,b_h}, \mu, \widetilde{T}_h^b, T_h^b$, $\{\gamma_{h,t}^b\}_{t \in [T_h^b]}, K_0$, and $b_0$.
8:      **Actor Update:** Update the parameter via

$$b_{h+1} \leftarrow b_h - \gamma^b \cdot \left[\widehat{\Upsilon}_{K_N}^{22}(-K_N \widehat{\mu}_{K,b_h} + b_h) + \widehat{\Upsilon}_{K_N}^{21} \widehat{\mu}_{K_N,b_h} + \widehat{q}_{K_N,b_h}\right].$$

9: **end for**
10: **Output:** Policy $\pi_{K,b} = \pi_{K_N,b_H}$, estimated mean-field state $\widehat{\mu}_{K,b} = \widehat{\mu}_{K_N,b_H}$.

---

**Theorem 4.1** (Convergence of Algorithm 1). *For a sufficiently small tolerance $\varepsilon > 0$, we set the number of iterations $S$ in Algorithm 1 such that*

$$S > \frac{\log(\|\mu_0 - \mu^*\|_2 \cdot \varepsilon^{-1})}{\log(1/L_0)}. \tag{4.1}$$

*For any $s \in [S]$, we define*

$$\varepsilon_s = \min\Big\{\left[1 - \rho(A - BK^*)\right]^4 (\|B\|_2 + \|\overline{A}\|_2)^{-4} (\|\mu_s\|_2^{-2} + \|d\|_2^{-2}) \cdot \sigma_{\min}(\Psi_\epsilon) \cdot \sigma_{\min}(R) \cdot \varepsilon^2,$$

$$\nu_{K^*} \cdot \left[1 - \rho(A - BK^*)\right]^4 \cdot \|B\|_2^{-2} \cdot M_b(\mu_s) \cdot \varepsilon^2, \, \varepsilon\Big\} \cdot 2^{-s-10}, \tag{4.2}$$

*where $\nu_{K^*}$ is defined in Proposition 3.3 and*

$$M_b(\mu_s) = 4\Big\|Q^{-1}(I - A)^\top \cdot \left[(I - A)Q^{-1}(I - A)^\top + BR^{-1}B^\top\right]^{-1} \cdot (\overline{A}\mu_s + d)\Big\|_2$$

$$\cdot \left[\nu_{K^*}^{-1} + \sigma_{\min}^{-1}(\Psi_\epsilon) \cdot \sigma_{\min}^{-1}(R)\right]^{1/2}. \tag{4.3}$$

*In the $s$-th policy update step in Line 3 of Algorithm 1, we set the inputs via Theorem B.4 such that $J_{\mu_s}(\pi_{s+1}) - J_{\mu_s}(\pi_{\mu_s}^*) < \varepsilon_s$, where the expected total cost $J_{\mu_s}(\cdot)$ is defined in Problem 2.2, and $\pi_{\mu_s}^* = \Lambda_1(\mu_s)$ is the optimal policy under the mean-field state $\mu_s$. Then it holds with probability at least $1 - \varepsilon^5$ that*

$$\|\mu_S - \mu^*\|_2 \leq \varepsilon, \qquad \|K_S - K^*\|_F \leq \varepsilon, \qquad \|b_S - b^*\|_2 \leq (1 + L_1) \cdot \varepsilon.$$

*Here $\mu^*$ is the Nash mean-field state, $K_S$ and $b_S$ are parameters of the policy $\pi_S$, and $K^*$ and $b^*$ are parameters of the Nash policy $\pi^*$.*

*Proof Sketch.* The proof of the theorem is based on the convergence of the natural actor-critic algorithm 2 and a contraction argument. First, we prove in Theorem B.4 that Algorithm 2 converges linearly to the optimal policy of Problem 2.2. By this, in each iteration of Algorithm 1, we control the error between $\mu_{s+1}$ and $\mu_{s+1}^*$ to be $\varepsilon_s > 0$ with high probability, where $\mu_{s+1}^*$ is the mean-field state generated by the optimal policy $\Lambda_1(\mu_s)$; in other words, $\mu_{s+1}^* = \Lambda(\mu_s)$. Combining the fact from Proposition 3.2 that $\Lambda(\cdot)$ is a contraction, we deduce that

$$\|\mu_{s+1} - \mu^*\|_2 \leq \big\|\Lambda(\mu_s) - \Lambda(\mu^*)\big\|_2 + \widetilde{\varepsilon}_s \leq L_0 \cdot \|\mu_s - \mu^*\|_2 + \widetilde{\varepsilon}_s$$

with high probability, where $\widetilde{\varepsilon}_s > 0$ is some error term and is specified in the detailed proof. Moreover, by telescoping sum, and note that the sum $\sum_{s=1}^{S} \widetilde{\varepsilon}_s$ is upper bounded by the desired error $\varepsilon$, we conclude the theorem. See §D.1 for a detailed proof. $\qquad\square$

We highlight that if the inputs of Algorithm 1 satisfy the conditions stated in Theorem B.4, it holds that $J_{\mu_s}(\pi_{s+1}) - J_{\mu_s}(\pi_{\mu_s}^*) < \varepsilon_s$ for any $s \in [S]$. See Theorem B.4 in §B.1 of the appendix for details. By Theorem 4.1, Algorithm 1 converges linearly to the unique Nash equilibrium pair $(\mu^*, \pi^*)$ of Problem 2.3. To the best of our knowledge, this theorem is the first successful attempt to establish that reinforcement learning with function approximation finds the Nash equilibrium pairs in mean-field games with theoretical guarantee, which lays the theoretical foundations for applying modern reinforcement learning techniques to general mean-field games.

## 5 CONCLUSION

For the discrete-time linear-quadratic mean-field games, we provide sufficient conditions for the existence and uniqueness of the Nash equilibrium pair. Moreover, we propose the mean-field actor-critic algorithm with linear function approximation that is shown converges to the Nash equilibrium pair with linear rate of convergence. Our algorithm can be modified to use other parametrized function classes, including deep neural networks, for solving mean-field games. For future research, we aim to extend our algorithm to other variations of mean-field games including risk-sensitive mean-field games (Saldi et al., 2018a; Tembine et al., 2014), robust mean-field games (Bauso et al., 2016), and partially observed mean-field games (Saldi et al., 2019).

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

# A    NOTATIONS IN THE APPENDIX

In the proof, for convenience, for any invertible matrix $M$, we denote by $M^{-\top} = (M^{-1})^{\top} = (M^{\top})^{-1}$ and $\|M\|_{\text{F}}$ the Frobenius norm. We also denote by $\text{svec}(M)$ the symmetric vectorization of the symmetric matrix $M$, which is the vectorization of the upper triangular matrix of the symmetric matrix $M$, with off-diagonal entries scaled by $\sqrt{2}$. We denote by $\text{smat}(\cdot)$ the inverse operation. For any matrices $G$ and $H$, we denote by $G \otimes H$ the Kronecker product, and $G \otimes_s H$ the symmetric Kronecker product, which is defined as a mapping on a vector $\text{svec}(M)$ such that $(G \otimes_s H)\text{svec}(M) = 1/2 \cdot \text{svec}(HMG^{\top} + GMH^{\top})$.

For notational simplicity, we write $\mathbb{E}_{\pi}(\cdot)$ to emphasize that the expectation is taken following the policy $\pi$.

# B    AUXILIARY ALGORITHMS AND ANALYSIS

## B.1    RESULTS IN D-LQR

In this section, we provide auxiliary results in analyzing Problem 2.2. First, we introduce the value functions of the Markov decision process (MDP) induced by Problem 2.2. We define the state- and action-value functions $V_{K,b}(x)$ and $Q_{K,b}(x,u)$ as follows

$$V_{K,b}(x) = \sum_{t=0}^{\infty}\Big\{\mathbb{E}\big[c(x_t, u_t)\,|\,x_0 = x\big] - J(K,b)\Big\}, \tag{B.1}$$

$$Q_{K,b}(x,u) = c(x,u) - J(K,b) + \mathbb{E}\big[V_{K,b}(x_1)\,|\,x_0 = x, u_0 = u\big], \tag{B.2}$$

where $x_t$ follows the state transition, and $u_t$ follows the policy $\pi_{K,b}$ given $x_t$. In other words, we have $u_t = -Kx_t + b + \sigma\eta_t$, where $\eta_t \sim \mathcal{N}(0, I)$. The following proposition establishes the close forms of these value functions.

**Proposition B.1.** The state-value function $V_{K,b}(x)$ takes the form of

$$V_{K,b}(x) = x^{\top}P_K x - \text{tr}(P_K\Phi_K) + 2f_{K,b}^{\top}(x - \mu_{K,b}) - \mu_{K,b}^{\top}P_K\mu_{K,b}, \tag{B.3}$$

and the action-value function $Q_{K,b}(x,u)$ takes the form of

$$\begin{aligned}
Q_{K,b}(x,u) &= \begin{pmatrix} x \\ u \end{pmatrix}^{\top}\Upsilon_K\begin{pmatrix} x \\ u \end{pmatrix} + 2\begin{pmatrix} p_{K,b} \\ q_{K,b} \end{pmatrix}^{\top}\begin{pmatrix} x \\ u \end{pmatrix} - \text{tr}(P_K\Phi_K) - \sigma^2\cdot\text{tr}(R + P_K BB^{\top}) \\
&\quad - b^{\top}Rb + 2b^{\top}RK\mu_{K,b} - \mu_{K,b}^{\top}(Q + K^{\top}RK + P_K)\mu_{K,b} \\
&\quad + 2f_{K,b}^{\top}\big[(\overline{A}\mu + d) - \mu_{K,b}\big] + (\overline{A}\mu + d)^{\top}P_K(\overline{A}\mu + d), \tag{B.4}
\end{aligned}$$

where $f_{K,b} = (I - A + BK)^{-\top}[(A - BK)^{\top}P_K(Bb + \overline{A}\mu + d) - K^{\top}Rb]$, and $\Upsilon_K$, $p_{K,b}$, and $q_{K,b}$ are defined in (3.7).

*Proof.* See §E.6 for a detailed proof. $\qquad\qquad\square$

By Proposition B.1, we know that $V_{K,b}(x)$ is quadratic in $x$, while $Q_{K,b}(x,u)$ is quadratic in $(x^{\top}, u^{\top})^{\top}$. Now, we show that (3.5) holds.

**Proposition B.2.** The expected total cost $J(K,b)$ defined in Problem 2.2 takes the form of

$$J(K,b) = J_1(K) + J_2(K,b) + \sigma^2\cdot\text{tr}(R) + \mu^{\top}\overline{Q}\mu,$$

where

$$J_1(K) = \text{tr}\big[(Q + K^{\top}RK)\Phi_K\big] = \text{tr}(P_K\Psi_\epsilon),$$

$$J_2(K,b) = \begin{pmatrix} \mu_{K,b} \\ b \end{pmatrix}^{\top}\begin{pmatrix} Q + K^{\top}RK & -K^{\top}R \\ -RK & R \end{pmatrix}\begin{pmatrix} \mu_{K,b} \\ b \end{pmatrix}.$$

Here $\mu_{K,b}$ is defined in (3.2), $\Phi_K$ is defined in (3.3), and $P_K$ is defined in (3.4).

*Proof.* See §E.3 for a detailed proof. □

The following proposition establishes the gradients of $J_1(K)$ and $J_2(K, b)$, respectively.

**Proposition B.3.** The gradient of $J_1(K)$ and the gradient of $J_2(K, b)$ with respect to $b$ take the forms of

$$\nabla_K J_1(K) = 2(\Upsilon_K^{22} K - \Upsilon_K^{21}) \cdot \Phi_K, \qquad \nabla_b J_2(K, b) = 2[\Upsilon_K^{22}(-K\mu_{K,b} + b) + \Upsilon_K^{21}\mu_{K,b} + q_{K,b}],$$

where $\Upsilon_K$ and $q_{K,b}$ are defined in (3.7).

*Proof.* See §E.5 for a detailed proof. □

The following theorem establishes the convergence of Algorithm 2.

**Theorem B.4** (Convergence of Algorithm 2). Assume that $\rho(A - BK_0) < 1$. Let $\varepsilon > 0$ be a sufficiently small tolerance. We set

$$\gamma \leq \big[\|R\|_2 + \|B\|_2^2 \cdot J(K_0, b_0) \cdot \sigma_{\min}^{-1}(\Psi_\epsilon)\big]^{-1},$$

$$N \geq C \cdot \|\Phi_{K^*}\|_2 \cdot \gamma^{-1} \cdot \log\Big\{4\big[J(K_0, b_0) - J(K^*, b^*)\big] \cdot \varepsilon^{-1}\Big\},$$

$$T_n \geq \text{poly}\big(\|K_n\|_F, \|b_0\|_2, \|\mu\|_2, J(K_0, b_0)\big) \cdot \lambda_{K_n}^{-4} \cdot \big[1 - \rho(A - BK_n)\big]^{-9} \cdot \varepsilon^{-5},$$

$$\widetilde{T}_n \geq \text{poly}\big(\|K_n\|_F, \|b_0\|_2, \|\mu\|_2, J(K_0, b_0)\big) \cdot \lambda_{K_n}^{-2} \cdot \big[1 - \rho(A - BK_n)\big]^{-12} \cdot \varepsilon^{-12},$$

$$\gamma_{n,t} = \gamma_0 \cdot t^{-1/2},$$

$$\gamma^b \leq \min\Big\{1 - \rho(A - BK_N), \big[1 - \rho(A - BK_N)\big]^{-2} \cdot \big(\|B\|_2^2 \cdot \|K_N\|_2^2 \cdot \|R\|_2 + \|B\|_2^2 \cdot \|Q\|_2\big)\Big\},$$

$$H \geq C_0 \cdot \nu_{K_N}^{-1} \cdot (\gamma^b)^{-1} \cdot \log\Big\{4\big[J(K_N, b_0) - J(K_N, b^{K_N})\big] \cdot \varepsilon^{-1}\Big\},$$

$$T_h^b \geq \text{poly}\big(\|K_N\|_F, \|b_h\|_2, \|\mu\|_2, J(K_N, b_0)\big) \cdot \lambda_{K_N}^{-4} \cdot \nu_{K_N}^{-4} \cdot \big[1 - \rho(A - BK_N)\big]^{-11} \cdot \varepsilon^{-5},$$

$$\widetilde{T}_h^b \geq \text{poly}\big(\|K_N\|_F, \|b_h\|_2, \|\mu\|_2, J(K_N, b_0)\big) \cdot \lambda_{K_N}^{-4} \cdot \nu_{K_N}^{-2} \cdot \big[1 - \rho(A - BK_N)\big]^{-17} \cdot \varepsilon^{-8},$$

$$\gamma_{h,t}^b = \gamma_0 \cdot t^{-1/2},$$

where $C$, $C_0$, and $\gamma_0$ are positive absolute constants, $\{K_n\}_{n\in[N]}$ and $\{b_h\}_{h\in[H]}$ are the sequences generated by Algorithm 2, $\lambda_{K_n}$ is specified in Proposition B.6, and $\nu_{K_N}$ is specified in Proposition 3.3. Then it holds with probability at least $1 - \varepsilon^{10}$ that

$$J(K_N, b_H) - J(K^*, b^*) < \varepsilon, \qquad \|b_H - b^*\|_2 \leq M_b(\mu) \cdot \varepsilon^{1/2},$$

$$\|K_N - K^*\|_F \leq \big[\sigma_{\min}^{-1}(\Psi_\epsilon) \cdot \sigma_{\min}^{-1}(R) \cdot \varepsilon\big]^{1/2}, \qquad \|\widehat{\mu}_{K_N, b_H} - \mu_{K^*, b^*}\|_2 \leq \varepsilon,$$

where $M_b(\mu)$ is defined in (4.3).

*Proof.* See §D.2 for a detailed proof. □

By Theorem B.4, given any mean-field state $\mu$, Algorithm 2 converges linearly to the optimal policy $\pi_\mu^*$ of Problem 2.2.

## B.2 PRIMAL-DUAL POLICY EVALUATION ALGORITHM

Note that the critic update steps in Algorithm 2 are built upon the estimators of the matrix $\Upsilon_K$ and the vector $q_{K,b}$. We now derive a policy evaluation algorithm to establish the estimators of $\Upsilon_K$ and $q_{K,b}$, which is based on gradient temporal difference algorithm (Sutton et al., 2009a).

We define the feature vector as

$$\psi(x, u) = \begin{pmatrix} \varphi(x, u) \\ x - \mu_{K,b} \\ u - (-K\mu_{K,b} + b) \end{pmatrix}, \tag{B.5}$$

where

$$\varphi(x, u) = \mathrm{svec}\left[\begin{pmatrix} x - \mu_{K,b} \\ u - (-K\mu_{K,b} + b) \end{pmatrix}\begin{pmatrix} x - \mu_{K,b} \\ u - (-K\mu_{K,b} + b) \end{pmatrix}^\top\right].$$

Recall $\mathrm{svec}(M)$ gives the symmetric vectorization of the symmetric matrix $M$. We also define

$$\alpha_{K,b} = \begin{pmatrix} \mathrm{svec}(\Upsilon_K) \\ \Upsilon_K\begin{pmatrix} \mu_{K,b} \\ -K\mu_{K,b} + b \end{pmatrix} + \begin{pmatrix} p_{K,b} \\ q_{K,b} \end{pmatrix} \end{pmatrix}, \tag{B.6}$$

where $\Upsilon_K$, $p_{K,b}$, and $q_{K,b}$ are defined in (3.7). To estimate $\Upsilon_K$ and $q_{K,b}$, it suffices to estimate $\alpha_{K,b}$. Meanwhile, we define

$$\Theta_{K,b} = \mathbb{E}_{\pi_{K,b}}\left\{\psi(x, u)\big[\psi(x, u) - \psi(x', u')\big]^\top\right\}, \tag{B.7}$$

where $(x', u')$ is the state-action pair after $(x, u)$ following the policy $\pi_{K,b}$ and the state transition. The following proposition characterizes the connection between $\Theta_{K,b}$ and $\alpha_{K,b}$.

**Proposition B.5.** It holds that

$$\begin{pmatrix} 1 & 0 \\ \mathbb{E}_{\pi_{K,b}}[\psi(x, u)] & \Theta_{K,b} \end{pmatrix}\begin{pmatrix} J(K, b) \\ \alpha_{K,b} \end{pmatrix} = \begin{pmatrix} J(K, b) \\ \mathbb{E}_{\pi_{K,b}}[c(x, u)\psi(x, u)] \end{pmatrix},$$

where $\psi(x, u)$ is defined in (B.5), $\alpha_{K,b}$ is defined in (B.6), and $\Theta_{K,b}$ is defined in (B.7).

*Proof.* See §E.7 for a detailed proof. □

By Proposition B.5, to obtain $\alpha_{K,b}$, it suffices to solve the following linear system in $\zeta = (\zeta_1, \zeta_2^\top)^\top$,

$$\widetilde{\Theta}_{K,b} \cdot \zeta = \begin{pmatrix} J(K, b) \\ \mathbb{E}_{\pi_{K,b}}[c(x, u)\psi(x, u)] \end{pmatrix}, \tag{B.8}$$

where for notational convenience, we define

$$\widetilde{\Theta}_{K,b} = \begin{pmatrix} 1 & 0 \\ \mathbb{E}_{\pi_{K,b}}[\psi(x, u)] & \Theta_{K,b} \end{pmatrix}. \tag{B.9}$$

The following proposition shows that $\Theta_{K,b}$ is invertible.

**Proposition B.6.** If $\rho(A - BK) < 1$, then the matrix $\Theta_{K,b}$ is invertible, and $\|\Theta_{K,b}\|_2 \leq 4(1 + \|K\|_F^2)^2 \cdot \|\Phi_K\|_2^2$. Also, $\sigma_{\min}(\widetilde{\Theta}_{K,b}) \geq \lambda_K$, where $\lambda_K$ only depends on $\|K\|_2$ and $\rho(A - BK)$.

*Proof.* See §E.8 for a detailed proof. □

By Proposition B.6, $\Theta_{K,b}$ is invertible. Therefore, (B.8) admits the unique solution $\zeta_{K,b} = (J(K, b), \alpha_{K,b}^\top)^\top$.

Now, we present the primal-dual gradient temporal difference algorithm.

**Primal-Dual Gradient Method.** Instead of solving (B.8) directly, we minimize the following loss function with respect to $\zeta = ((\zeta^1)^\top, (\zeta^2)^\top)$,

$$\left[\zeta^1 - J(K, b)\right]^2 + \left\|\mathbb{E}_{\pi_{K,b}}[\psi(x, u)]\zeta^1 + \Theta_{K,b}\zeta^2 - \mathbb{E}_{\pi_{K,b}}[c(x, u)\psi(x, u)]\right\|_2^2. \tag{B.10}$$

By Fenchel's duality, the minimization of (B.10) is equivalent to the following primal-dual min-max problem,

$$\min_{\zeta \in \mathcal{V}_\zeta} \max_{\xi \in \mathcal{V}_\xi} F(\zeta, \xi) = \left\{\mathbb{E}_{\pi_{K,b}}[\psi(x, u)]\zeta^1 + \Theta_{K,b}\zeta^2 - \mathbb{E}_{\pi_{K,b}}[c(x, u)\psi(x, u)]\right\}^\top \xi^2 \tag{B.11}$$

$$+ \left[\zeta^1 - J(K, b)\right] \cdot \xi^1 - \|\xi\|_2^2/2,$$

where we restrict the primal variable $\zeta$ in a compact set $\mathcal{V}_\zeta$ and the dual variable $\xi$ in a compact set $\mathcal{V}_\xi$, which are specified in Definition B.7. It holds that

$$\nabla_{\zeta^1} F = \xi^1 + \mathbb{E}_{\pi_{K,b}}\big[\psi(x,u)\big]^\top \xi^2, \qquad \nabla_{\zeta^2} F = \Theta_{K,b}^\top \xi^2, \qquad \nabla_{\xi^1} F = \zeta^1 - J(K,b) - \xi^1,$$

$$\nabla_{\xi^2} F = \mathbb{E}_{\pi_{K,b}}\big[\psi(x,u)\big]\zeta^1 + \Theta_{K,b}\zeta^2 - \mathbb{E}_{\pi_{K,b}}\big[c(x,u)\psi(x,u)\big] - \xi^2. \tag{B.12}$$

The primal-dual gradient method updates $\zeta$ and $\xi$ via

$$\zeta^1 \leftarrow \zeta^1 - \gamma \cdot \nabla_{\zeta^1} F(\zeta,\xi), \qquad \zeta^2 \leftarrow \zeta^2 - \gamma \cdot \nabla_{\zeta^2} F(\zeta,\xi)$$

$$\xi^1 \leftarrow \xi^1 - \gamma \cdot \nabla_{\xi^1} F(\zeta,\xi), \qquad \xi^2 \leftarrow \xi^2 - \gamma \cdot \nabla_{\xi^2} F(\zeta,\xi). \tag{B.13}$$

**Estimation of Mean-Field State $\mu_{K,b}$.** To utilize the primal-dual gradient method in (B.13), it remains to evaluate the feature vector $\psi(x,u)$. Note that by (B.5), the evaluation of the feature vector $\psi(x,u)$ requires the mean-field state $\mu_{K,b}$. In what follows, we establish the estimator $\widehat{\mu}_{K,b}$ of the mean-field state $\mu_{K,b}$ by simulating the MDP following the policy $\pi_{K,b}$ for $\widetilde{T}$ steps, and calculate the estimated feature vector $\widehat{\psi}(x,u)$ by

$$\widehat{\psi}(x,u) = \begin{pmatrix} \widehat{\varphi}(x,u) \\ x - \widehat{\mu}_{K,b} \\ u - (-K\widehat{\mu}_{K,b} + b) \end{pmatrix}, \tag{B.14}$$

where $\widehat{\varphi}(x,u)$ takes the form of

$$\widehat{\varphi}(x,u) = \mathrm{svec}\left[ \begin{pmatrix} x - \widehat{\mu}_{K,b} \\ u - (-K\widehat{\mu}_{K,b} + b) \end{pmatrix} \begin{pmatrix} x - \widehat{\mu}_{K,b} \\ u - (-K\widehat{\mu}_{K,b} + b) \end{pmatrix}^\top \right].$$

We now define the sets $\mathcal{V}_\zeta$ and $\mathcal{V}_\xi$ in (B.11).

**Definition B.7.** Given $K_0$ and $b_0$ such that $\rho(A - BK_0) < 1$ and $J(K_0, b_0) < \infty$, we define the sets $\mathcal{V}_\zeta$ and $\mathcal{V}_\xi$ as

$$\mathcal{V}_\zeta = \Big\{ \zeta : 0 \le \zeta^1 \le J(K_0, b_0), \|\zeta^2\|_2 \le M_{\zeta,1} + M_{\zeta,2} \cdot (1 + \|K\|_{\mathrm{F}}) \cdot \big[1 - \rho(A - BK)\big]^{-1} \Big\},$$

$$\mathcal{V}_\xi = \Big\{ \xi : |\xi^1| \le J(K_0, b_0), \|\xi^2\|_2 \le M_\xi \cdot \big(1 + \|K\|_{\mathrm{F}}^2\big)^3 \cdot \big[1 - \rho(A - BK)\big]^{-1} \Big\}.$$

Here $M_{\zeta,1}$, $M_{\zeta,2}$, and $M_\xi$ are constants independent of $K$ and $b$, which take the forms of

$$M_{\zeta,1} = \Big[ \big(\|Q\|_{\mathrm{F}} + \|R\|_{\mathrm{F}}\big) + \big(\|A\|_{\mathrm{F}}^2 + \|B\|_{\mathrm{F}}^2\big) \cdot \sqrt{d} \cdot J(K_0, b_0) \cdot \sigma_{\min}^{-1}(\Psi_\omega) \Big]$$

$$+ \big(\|A\|_2 + \|B\|_2\big) \cdot J(K_0, b_0)^2 \cdot \sigma_{\min}^{-1}(\Psi_\omega) \cdot \sigma_{\min}^{-1}(Q),$$

$$+ \Big[ \big(\|Q\|_2 + \|R\|_2\big) + \big(\|A\|_2 + \|B\|_2\big)^2 \cdot J(K_0, b_0) \cdot \sigma_{\min}^{-1}(\Psi_\omega) \Big]$$

$$\cdot J(K_0, b_0) \cdot \big[\sigma_{\min}^{-1}(Q) + \sigma_{\min}^{-1}(R)\big]$$

$$M_{\zeta,2} = \big(\|A\|_2 + \|B\|_2\big) \cdot (\kappa_Q + \kappa_R), \qquad M_\xi = C \cdot (M_{\zeta,1} + M_{\zeta,2}) \cdot J(K_0, b_0)^2 \cdot \sigma_{\min}^{-2}(Q),$$

where $C$ is a positive absolute constant, and $\kappa_Q$ and $\kappa_R$ are condition numbers of $Q$ and $R$, respectively.

We summarize the primal-dual gradient temporal difference algorithm in Algorithm 3. Hereafter, for notational convenience, we denote by $\widehat{\psi}_t$ the estimated feature vector $\widehat{\psi}(x_t, u_t)$.

We now characterize the rate of convergence of Algorithm 3.

**Theorem B.8** (Convergence of Algorithm 3). Given $K_0, b_0, K$, and $b$ such that $\rho(A - BK_0) < 1$ and $J(K,b) \le J(K_0, b_0)$, we define the sets $\mathcal{V}_\zeta$ and $\mathcal{V}_\xi$ through Definition B.7. Let $\gamma_t = \gamma_0 t^{-1/2}$, where $\gamma_0$ is a positive absolute constant. Let $\rho \in (\rho(A - BK), 1)$. For $\widetilde{T} \ge \mathrm{poly}_0(\|K\|_{\mathrm{F}}, \|b\|_2, \|\mu\|_2, J(K_0, b_0)) \cdot (1 - \rho)^{-6}$ and a sufficiently large $T$, it holds with probability at least $1 - T^{-4} - \widetilde{T}^{-6}$ that

$$\|\widehat{\alpha}_{K,b} - \alpha_{K,b}\|_2^2 \le \lambda_K^{-2} \cdot \mathrm{poly}_1\big(\|K\|_{\mathrm{F}}, \|b\|_2, \|\mu\|_2, J(K_0, b_0)\big) \cdot \left[ \frac{\log^6 T}{T^{1/2} \cdot (1 - \rho)^4} + \frac{\log \widetilde{T}}{\widetilde{T}^{1/4} \cdot (1 - \rho)^2} \right],$$

---

**Algorithm 3** Primal-Dual Gradient Temporal Difference Algorithm.

---

1: **Input:** Policy $\pi_{K,b}$, mean-field state $\mu$, numbers of iteration $\widetilde{T}$ and $T$, stepsizes $\{\gamma_t\}_{t \in [T]}$, parameters $K_0$ and $b_0$.
2: Define the sets $\mathcal{V}_\zeta$ and $\mathcal{V}_\xi$ via Definition B.7 with $K_0$ and $b_0$.
3: Initialize the parameters by $\zeta_0 \in \mathcal{V}_\zeta$ and $\xi_0 \in \mathcal{V}_\xi$.
4: Sample $\widetilde{x}_0$ from the the stationary distribution $\mathcal{N}(\mu_{K,b}, \Phi_K)$.
5: **for** $t = 0, \ldots, \widetilde{T} - 1$ **do**
6:      Given the mean-field state $\mu$, take action $\widetilde{u}_t$ following $\pi_{K,b}$ and generate the next state $\widetilde{x}_{t+1}$.
7: **end for**
8: Set $\widehat{\mu}_{K,b} \leftarrow 1/\widetilde{T} \cdot \sum_{t=1}^{\widetilde{T}} \widetilde{x}_t$ and compute the estimated feature vector $\widehat{\psi}$ via (B.14).
9: Sample $x_0$ from the the stationary distribution $\mathcal{N}(\mu_{K,b}, \Phi_K)$.
10: **for** $t = 0, \ldots, T - 1$ **do**
11:      Given the mean-field state $\mu$, take action $u_t$ following $\pi_{K,b}$, observe the cost $c_t$, and generate the next state $x_{t+1}$.
12:      Set $\delta_{t+1} \leftarrow \zeta_t^1 + (\widehat{\psi}_t - \widehat{\psi}_{t+1})^\top \zeta_t^2 - c_t$.
13:      Update parameters via

$$\zeta_{t+1}^1 \leftarrow \zeta_t^1 - \gamma_{t+1} \cdot (\xi_t^1 + \widehat{\psi}_t^\top \xi_t^2), \qquad \zeta_{t+1}^2 \leftarrow \zeta_t^2 - \gamma_{t+1} \cdot \widehat{\psi}_t(\widehat{\psi}_t - \widehat{\psi}_{t+1})^\top \xi_t^2,$$
$$\xi_{t+1}^1 \leftarrow (1 - \gamma_{t+1}) \cdot \xi_t^1 + \gamma_{t+1} \cdot (\zeta_t^1 - c_t), \qquad \xi_{t+1}^2 \leftarrow (1 - \gamma_{t+1}) \cdot \xi_t^2 + \gamma_{t+1} \cdot \delta_{t+1} \cdot \widehat{\psi}_t.$$

14:      Project $\zeta_{t+1}$ and $\xi_{t+1}$ to $\mathcal{V}_\zeta$ and $\mathcal{V}_\xi$, respectively.
15: **end for**
16: Set $\widehat{\alpha}_{K,b} \leftarrow (\sum_{t=1}^T \gamma_t)^{-1} \cdot (\sum_{t=1}^T \gamma_t \cdot \zeta_t^2)$, and

$$\widehat{\Upsilon}_K \leftarrow \mathrm{smat}(\widehat{\alpha}_{K,b,1}), \qquad \begin{pmatrix} \widehat{p}_{K,b} \\ \widehat{q}_{K,b} \end{pmatrix} \leftarrow \widehat{\alpha}_{K,b,2} - \widehat{\Upsilon}_K \begin{pmatrix} \widehat{\mu}_{K,b} \\ -K\widehat{\mu}_{K,b} + b \end{pmatrix},$$
where $\widehat{\alpha}_{K,b,1} = (\widehat{\alpha}_{K,b})_1^{(k+d+1)(k+d)/2}$ and $\widehat{\alpha}_{K,b,2} = (\widehat{\alpha}_{K,b})_{(k+d+1)(k+d)/2+1}^{(k+d+3)(k+d)/2}$.

17: **Output:** Estimators $\widehat{\mu}_{K,b}$, $\widehat{\Upsilon}_K$, and $\widehat{q}_{K,b}$.

---

where $\lambda_K$ is defined in Proposition B.6. Same bounds for $\|\widehat{\Upsilon}_K - \Upsilon_K\|_\mathrm{F}^2$, $\|\widehat{p}_{K,b} - p_{K,b}\|_2^2$, and $\|\widehat{q}_{K,b} - q_{K,b}\|_2^2$ hold. Meanwhile, it holds with probability at least $1 - \widetilde{T}^{-6}$ that

$$\|\widehat{\mu}_{K,b} - \mu_{K,b}\|_2 \leq \frac{\log \widetilde{T}}{\widetilde{T}^{1/4}} \cdot (1 - \rho)^{-2} \cdot \mathrm{poly}_2\big(\|\Phi_K\|_2, \|K\|_\mathrm{F}, \|b\|_2, \|\mu\|_2, J(K_0, b_0)\big).$$

*Proof.* See §D.3 for a detailed proof. $\qquad\qquad\qquad\qquad\qquad\qquad\qquad\qquad\qquad\qquad\qquad\qquad$ $\square$

### B.3   TEMPORAL DIFFERENCE POLICY EVALUATION ALGORITHM

Besides the primal-dual gradient temporal difference algorithm, we can also evaluate $\alpha_{K,b}$ by TD(0) method (Sutton and Barto, 2018) in practice, which is presented in Algorithm 4.

Note that in related literature (Bhandari et al., 2018; Korda and La, 2015), non-asymptotic convergence analysis of TD(0) method with linear function approximation is only applied to discounted MDP. As for our ergodic setting, the convergence of TD(0) method is only shown asymptotically (Borkar and Meyn, 2000; Kushner and Yin, 2003) using ordinary differential equation method. Therefore, in the convergence theorem proposed in §3, we only focus on the primal-dual gradient temporal difference method (Algorithm 3) to establish non-asymptotic convergence result.

## C   GENERAL FORMULATION

Compared with Problem 2.3, a more general formulation includes an additional term $x_t^\top P\mu$ in the cost function. For the completeness of this paper, we define this general formulation here. Following from a same argument as in §2, it suffices to study the setting where $t$ is sufficiently large. First, we propose the following general drifted LQR (general D-LQR) problem, which is parallel to Problem 2.2.

---

**Algorithm 4** Temporal Difference Policy Evaluation Algorithm.

1: **Input:** Policy $\pi_{K,b}$, number of iteration $\widetilde{T}$ and $T$, stepsizes $\{\gamma_t\}_{t\in[T]}$.
2: Sample $\widetilde{x}_0$ from the stationary distribution $\mathcal{N}(\mu_{K,b}, \Phi_K)$.
3: **for** $t = 0, \ldots, \widetilde{T} - 1$ **do**
4:     Take action $\widetilde{u}_t$ under the policy $\pi_{K,b}$ and generate the next state $\widetilde{x}_{t+1}$.
5: **end for**
6: Set $\widehat{\mu}_{K,b} \leftarrow 1/\widetilde{T} \cdot \sum_{t=1}^{\widetilde{T}} \widetilde{x}_t$.
7: Sample $x_0$ from the the stationary distribution $\mathcal{N}(\mu_{K,b}, \Phi_K)$.
8: **for** $t = 0, \ldots, T$ **do**
9:     Given the mean-field state $\mu$, take action $u_t$ following $\pi_{K,b}$, observe the cost $c_t$, and generate the next state $x_{t+1}$.
10:     Set $\delta_{t+1} \leftarrow \zeta_t^1 + (\widehat{\psi}_t - \widehat{\psi}_{t+1})^\top \zeta_t^2 - c_t$.
11:     Update parameters via $\zeta_{t+1}^1 \leftarrow (1 - \gamma_{t+1}) \cdot \zeta_t^1 + \gamma_{t+1} \cdot c_t$ and $\zeta_{t+1}^2 \leftarrow \zeta_t^2 - \gamma_{t+1} \cdot \delta_{t+1} \cdot \widehat{\psi}_t$.
12:     Project $\zeta_t$ to $\mathcal{V}'_\zeta$, where $\mathcal{V}'_\zeta$ is a compact set.
13: **end for**
14: Set $\widehat{\alpha}_{K,b} \leftarrow (\sum_{t=1}^{T} \gamma_t)^{-1} \cdot (\sum_{t=1}^{T} \gamma_t \cdot \zeta_t^2)$, and

$$\widehat{\Upsilon}_K \leftarrow \mathrm{smat}(\widehat{\alpha}_{K,b,1}), \qquad \begin{pmatrix} \widehat{p}_{K,b} \\ \widehat{q}_{K,b} \end{pmatrix} \leftarrow \widehat{\alpha}_{K,b,2} - \widehat{\Upsilon}_K \begin{pmatrix} \widehat{\mu}_{K,b} \\ -K\widehat{\mu}_{K,b} + b \end{pmatrix},$$

    where $\widehat{\alpha}_{K,b,1} = (\widehat{\alpha}_{K,b})_1^{(k+d+1)(k+d)/2}$ and $\widehat{\alpha}_{K,b,2} = (\widehat{\alpha}_{K,b})_{(k+d+1)(k+d)/2+1}^{(k+d+3)(k+d)/2}$.

15: **Output:** Estimators $\widehat{\mu}_{K,b}$, $\widehat{\Upsilon}_K$, and $\widehat{q}_{K,b}$.

---

**Problem C.1** (General D-LQR). For any given mean-field state $\mu \in \mathbb{R}^m$, consider the following formulation

$$x_{t+1} = Ax_t + Bu_t + \overline{A}\mu + d + \omega_t,$$
$$\widetilde{c}_\mu(x_t, u_t) = x_t^\top Q x_t + u_t^\top R u_t + \mu^\top \overline{Q} \mu + 2x_t^\top P \mu,$$
$$\widetilde{J}_\mu(\pi) = \lim_{T\to\infty} \mathbb{E}\left[\frac{1}{T} \sum_{t=0}^{T} \widetilde{c}_\mu(x_t, u_t)\right],$$

where $x_t \in \mathbb{R}^m$ is the state vector, $u_t \in \mathbb{R}^k$ is the action vector generated by the policy $\pi$, $\omega_t \in \mathbb{R}^m$ is an independent random noise term following the Gaussian distribution $\mathcal{N}(0, \Psi_\omega)$, and $d \in \mathbb{R}^m$ is a drift term. We aim to find an optimal policy $\pi_\mu^*$ such that $\widetilde{J}_\mu(\pi_\mu^*) = \inf_{\pi\in\Pi} \widetilde{J}_\mu(\pi)$.

In Problem C.1, the unique optimal policy $\pi_\mu^*(\cdot)$ still admits a linear form $\pi_\mu^*(x_t) = -K_{\pi_\mu^*} x_t + b_{\pi_\mu^*}$ (Anderson and Moore, 2007), where the matrix $K_{\pi_\mu^*} \in \mathbb{R}^{k\times m}$ and the vector $b_{\pi_\mu^*} \in \mathbb{R}^k$ are the parameters of the policy $\pi$. It then suffices to find the optimal policy in the class $\Pi$ defined in (2.1).

Parallel to Problem 2.3, we define the general LQ-MFG problem as follows.

**Problem C.2** (General LQ-MFG). We consider the following formulation

$$x_{t+1} = Ax_t + Bu_t + \overline{A}\mu + d + \omega_t,$$
$$\widetilde{c}(x_t, u_t) = x_t^\top Q x_t + u_t^\top R u_t + \mu^\top \overline{Q} \mu + 2x_t^\top P \mu,$$
$$\widetilde{J}(\pi, \mu) = \lim_{T\to\infty} \mathbb{E}\left[\frac{1}{T} \sum_{t=0}^{T} \widetilde{c}(x_t, u_t)\right],$$

where $x_t \in \mathbb{R}^m$ is the state vector, $u_t \in \mathbb{R}^k$ is the action vector generated by the policy $\pi$, $\mu \in \mathbb{R}^m$ is the mean-field state, $\omega_t \in \mathbb{R}^m$ is an independent random noise term following the Gaussian distribution $\mathcal{N}(0, \Psi_\omega)$, and $d \in \mathbb{R}^m$ is a drift term. We aim to find a pair $(\mu^*, \pi^*)$ such that (i) $\widetilde{J}(\pi^*, \mu^*) = \inf_{\pi\in\Pi} \widetilde{J}(\pi, \mu^*)$; (ii) $\mathbb{E}x_t^*$ converges to $\mu^*$ as $t \to \infty$, where $\{x_t^*\}_{t\geq 0}$ is the Markov chain of states generated by the policy $\pi^*$.

One can see that Problem C.2 aims to find a Nash equilibrium pair $(\mu^*, \pi^*)$.

Similar to the discussions in §3.2, to solve Problem C.2, one can design an algorithm similar to Algorithm 1, which solves Problem C.1 and obtain the new mean-field state at each iteration. We omit the detailed algorithm here. Now, we focus on solving Problem C.1 in the sequel.

Similar to §3.3, we drop the subscript $\mu$ when we focus on Problem C.1 for a fixed $\mu$. We write $\pi_{K,b}(x) = -Kx + b + \sigma \cdot \eta$ to emphasize the dependence on $K$ and $b$, and $\widetilde{J}(K,b) = \widetilde{J}(\pi_{K,b})$ consequently. We derive an explicit form of the expected total cost $\widetilde{J}(K,b)$ in the following proposition.

**Proposition C.3.** The expected total cost $\widetilde{J}(K,b)$ in Problem C.1 is decomposed as

$$\widetilde{J}(K,b) = \widetilde{J}_1(K) + \widetilde{J}_2(K,b) + \sigma^2 \cdot \text{tr}(R) + \mu^\top \overline{Q}\mu,$$

where $\widetilde{J}_1(K)$ and $\widetilde{J}_2(K,b)$ take the forms of

$$\widetilde{J}_1(K) = \text{tr}\big[(Q + K^\top RK)\Phi_K\big] = \text{tr}(P_K \Psi_\epsilon),$$

$$\widetilde{J}_2(K,b) = \begin{pmatrix} \mu_{K,b} \\ b \end{pmatrix}^\top \begin{pmatrix} Q + K^\top RK & -K^\top R \\ -RK & R \end{pmatrix} \begin{pmatrix} \mu_{K,b} \\ b \end{pmatrix} + 2\mu^\top P \mu_{K,b}.$$

Here $\mu_{K,b}$ is given in (3.2), $\Phi_K$ is given in (3.3), and $P_K$ is given in (3.4).

*Proof.* The proof is similar to the one of Proposition B.2. Thus we omit it here. $\square$

Compared with the form of $J(K,b)$ given in (3.5), we see that the only difference is that $\widetilde{J}(K,b)$ contains an extra term $2\mu^\top P \mu_{K,b}$ in $\widetilde{J}_2(K,b)$, which is only a linear term in $b$ (recall that $\mu_{K,b}$ is linear in $b$ by (3.2)). Thus, $\widetilde{J}_2(K,b)$ is still strongly convex in $b$, as shown in the proposition below.

**Proposition C.4.** Given any $K$, the function $\widetilde{J}_2(K,b)$ is $\nu_K$-strongly convex in $b$, here $\nu_K = \sigma_{\min}(Y_{1,K}^\top Y_{1,K} + Y_{2,K}^\top Y_{2,K})$, where $Y_{1,K} = R^{1/2}K(I - A + BK)^{-1}B - R^{1/2}$ and $Y_{2,K} = Q^{1/2}(I - A + BK)^{-1}B$. Also, $\widetilde{J}_2(K,b)$ has $\iota_K$-Lipschitz continuous gradient in $b$, where $\iota_K$ is upper bounded such that $\iota_K \leq [1 - \rho(A - BK)]^{-2} \cdot (\|B\|_*^2 \cdot \|K\|_*^2 \cdot \|R\|_* + \|B\|_*^2 \cdot \|Q\|_*)$.

*Proof.* The proof is similar to the one of Proposition 3.3. Thus we omit it here. $\square$

Parallel to Proposition 3.4, we derive a similar proposition in the sequel.

**Proposition C.5.** Denote by $\widetilde{b}^K = \text{argmin}_b \widetilde{J}_2(K,b)$, then $\widetilde{J}_2(K,\widetilde{b}^K)$ takes the form

$$\widetilde{J}_2(K,\widetilde{b}^K) = \begin{pmatrix} \overline{A}\mu + d \\ P^\top \mu \end{pmatrix}^\top \begin{pmatrix} S & S(I-A)Q^{-1} \\ Q^{-1}(I-A)^\top S & 3Q^{-1}(I-A)^\top S(I-A)Q^{-1} - Q^{-1} \end{pmatrix} \begin{pmatrix} \overline{A}\mu + d \\ P^\top \mu \end{pmatrix},$$

which is independent of $K$. Here $S = [(I - A)Q^{-1}(I - A)^\top + BR^{-1}B^\top]^{-1}$. And $\widetilde{b}^K$ takes the form

$$\widetilde{b}^K = \big[KQ^{-1}(I-A)^\top - R^{-1}B^\top\big] \cdot S \cdot \big[(\overline{A}\mu + d) + (I-A)Q^{-1}P^\top \mu\big] - KQ^{-1}P^\top \mu.$$

*Proof.* The proof is similar to the one of Proposition 3.4. Thus we omit it here. $\square$

Similar to Problem 2.2, we define the state- and action-value functions as

$$\widetilde{V}_{K,b}(x) = \sum_{t=0}^\infty \Big\{\mathbb{E}\big[\widetilde{c}(x_t, u_t) \,|\, x_0 = x, u_t = -Kx_t + b + \sigma\eta_t\big] - \widetilde{J}(K,b)\Big\},$$

$$\widetilde{Q}_{K,b}(x,u) = \widetilde{c}(x,u) - \widetilde{J}(K,b) + \mathbb{E}\big[\widetilde{V}_{K,b}(x') \,|\, x, u\big],$$

where the $x'$ is the state generated by the state transition after the state-action pair $(x, u)$. A slight modification of Proposition B.1 gives the proposition below.

**Proposition C.6.** For Problem C.1, the state-value function $\widetilde{V}_{K,b}(x)$ takes the form

$$\widetilde{V}_{K,b}(x) = x^\top P_K x - \text{tr}(P_K \Phi_K) + 2\widetilde{f}_{K,b}^\top(x - \mu_{K,b}) - (\mu_{K,b})^\top P_K \mu_{K,b},$$

and the action-value function $\widetilde{Q}_{K,b}(x, u)$ takes the form

$$
\begin{aligned}
\widetilde{Q}_{K,b}(x,u) &= \begin{pmatrix} x \\ u \end{pmatrix}^\top \Upsilon_K \begin{pmatrix} x \\ u \end{pmatrix} + 2\begin{pmatrix} \widetilde{p}_{K,b} \\ \widetilde{q}_{K,b} \end{pmatrix}^\top \begin{pmatrix} x \\ u \end{pmatrix} - \text{tr}(P_K \Phi_K) - \sigma^2 \cdot \text{tr}(R + P_K BB^\top) - b^\top R b \\
&\quad + 2b^\top RK\mu_{K,b} - (\mu_{K,b})^\top (Q + K^\top RK + P_K)\mu_{K,b} + 2\widetilde{f}_{K,b}^\top [(\overline{A}\mu + d) - \mu_{K,b}] \\
&\quad + (\overline{A}\mu + d)^\top P_K(\overline{A}\mu + d) - 2\mu^\top P\mu_{K,b}.
\end{aligned}
$$

Here the matrix $\Upsilon_K$ is given in (3.7), and the vectors $\widetilde{p}_{K,b}, \widetilde{q}_{K,b}$ are given as

$$\begin{pmatrix} \widetilde{p}_{K,b} \\ \widetilde{q}_{K,b} \end{pmatrix} = \begin{pmatrix} A^\top [P_K \cdot (\overline{A}\mu + d) + \widetilde{f}_{K,b}] + P\mu \\ B^\top [P_K \cdot (\overline{A}\mu + d) + \widetilde{f}_{K,b}] \end{pmatrix}, \tag{C.1}$$

where the vector $\widetilde{f}_{K,b} = (I - A + BK)^{-\top}[(A - BK)^\top P_K(Bb + \overline{A}\mu + d) - K^\top Rb + P\mu]$.

*Proof.* The proof is similar to the one of Proposition B.1. Thus we omit it here. $\qquad\square$

Now we establish the gradients of $\widetilde{J}(K, b)$ for Problem C.1.

**Proposition C.7.** The gradient of $\widetilde{J}_1(K)$ and the gradient of $\widetilde{J}_2(K, b)$ w.r.t. $b$ takes the form

$$\nabla_K \widetilde{J}_1(K) = 2(\Upsilon_K^{22} K - \Upsilon_K^{21}) \cdot \Phi_K, \qquad \nabla_b \widetilde{J}_2(K, b) = 2\big[\Upsilon_K^{22}(-K\mu_{K,b} + b) + \Upsilon_K^{21}\mu_{K,b} + \widetilde{q}_{K,b}\big],$$

where the matrix $\Upsilon_K$ is given in (3.7), and the vector $\widetilde{q}_{K,b}$ is given in (C.1).

*Proof.* The proof is similar to the one of Proposition B.3. Thus we omit it here. $\qquad\square$

Equipped with above results, parallel to the analysis in §3, it is clear that by slight modification of Algorithms 1, 2, and 3, we derive similar actor-critic algorithms to solve both Problem C.2 and Problem C.1, where all the non-asymptotic convergence results hold. We omit the algorithms and the convergence results here.

# D PROOFS OF THEOREMS

## D.1 PROOF OF THEOREM 4.1

We define $\mu_{s+1}^* = \Lambda(\mu_s)$, which is the mean-field state generated by the optimal policy $\pi_{K^*(\mu_s), b^*(\mu_s)} = \Lambda_1(\mu_s)$ under the current mean-field state $\mu_s$. By Proposition 3.4, the optimal $K^*(\mu)$ is independent of the mean-field state $\mu$. Therefore, we write $K^* = K^*(\mu)$ hereafter for notational convenience. By (3.2), we know that

$$\mu_{s+1}^* = (I - A + BK^*)^{-1} \cdot \big[Bb^*(\mu_s) + \overline{A}\mu_s + d\big].$$

We define

$$\widetilde{\mu}_{s+1} = (I - A + BK_s)^{-1}(Bb_s + \overline{A}\mu_s + d),$$

which is the mean-field state generated by the policy $\pi_s$ under the current mean-field state $\mu_s$, where $K_s$ and $b_s$ are the parameters of the policy $\pi_s$. By triangle inequality, we have

$$\|\mu_{s+1} - \mu^*\|_2 \le \underbrace{\|\mu_{s+1} - \widetilde{\mu}_{s+1}\|_2}_{E_1} + \underbrace{\|\widetilde{\mu}_{s+1} - \mu_{s+1}^*\|_2}_{E_2} + \underbrace{\|\mu_{s+1}^* - \mu^*\|_2}_{E_3}, \tag{D.1}$$

where $\mu_{s+1}$ is generated by Algorithm 1. We upper bound $E_1$, $E_2$, and $E_3$ in the sequel.

**Upper Bound of $E_1$.** By Theorem B.4, it holds with probability at least $1 - \varepsilon^{10}$ that

$$E_1 = \|\mu_{s+1} - \widetilde{\mu}_{s+1}\|_2 < \varepsilon_s \le \varepsilon/8 \cdot 2^{-s}, \tag{D.2}$$

where $\varepsilon_s$ is given in (4.2).

**Upper Bound of $E_2$.** By the triangle inequality, we have

$$
\begin{aligned}
E_2 &= \left\| (I - A + BK_s)^{-1}(Bb_s + \overline{A}\mu_s + d) - (I - A + BK^*)^{-1} \cdot \left[ Bb^*(\mu_s) + \overline{A}\mu_s + d \right] \right\|_2 \\
&\leq \left\| Bb^*(\mu_s) + \overline{A}\mu_s + d \right\|_2 \cdot \left\| \left[ I - A + BK^* + B(K_s - K^*) \right]^{-1} - (I - A + BK^*)^{-1} \right\|_2 \\
&\quad + \left\| (I - A + BK_s)^{-1} \right\|_2 \cdot \|B\|_2 \cdot \left\| b_s - b^*(\mu_s) \right\|_2.
\end{aligned}
\tag{D.3}
$$

By Taylor's expansion, we have

$$
\begin{aligned}
&\left\| \left[ I - A + BK^* + B(K_s - K^*) \right]^{-1} - (I - A + BK^*)^{-1} \right\|_2 \\
&= \left\| (I - A + BK^*)^{-1} \left[ I + (I - A + BK^*)^{-1} B(K_s - K^*) \right]^{-1} - (I - A + BK^*)^{-1} \right\|_2 \\
&\leq 2 \left\| (I - A + BK^*)^{-1} B(K_s - K^*)(I - A + BK^*)^{-1} \right\|_2.
\end{aligned}
\tag{D.4}
$$

Meanwhile, by Taylor's expansion, it holds with probability at least $1 - \varepsilon^{10}$ that

$$
\begin{aligned}
\left\| (I - A + BK_s)^{-1} \right\|_2 &= \left\| \left( I - A + BK^* + B(K_s - K^*) \right)^{-1} \right\|_2 \\
&= \left\| (I - A + BK^*)^{-1} \left( I + (I - A + BK^*)^{-1} B(K_s - K^*) \right)^{-1} \right\|_2 \\
&\leq \left[ 1 - \rho(A - BK^*) \right]^{-1} \cdot \left( 1 + \left\| (I - A + BK^*)^{-1} B \right\|_2 \cdot \|K^* - K_s\|_2 \right) \\
&\leq 2 \left[ 1 - \rho(A - BK^*) \right]^{-2},
\end{aligned}
\tag{D.5}
$$

where the last inequality comes from Theorem B.4. By plugging (D.4) and (D.5) in (D.3), it holds with probability at least $1 - \varepsilon^{10}$ that

$$
\begin{aligned}
E_2 &\leq 2 \left\| Bb^*(\mu_s) + \overline{A}\mu_s + d \right\|_2 \cdot \left\| (I - A + BK^*)^{-1} B(K_s - K^*)(I - A + BK^*)^{-1} \right\|_2 \\
&\quad + \left\| (I - A + BK_s)^{-1} \right\|_2 \cdot \|B\|_2 \cdot \left\| b_s - b^*(\mu_s) \right\|_2 \\
&\leq 2 \left\| Bb^*(\mu_s) + \overline{A}\mu_s + d \right\|_2 \cdot \left[ 1 - \rho(A - BK^*) \right]^{-2} \cdot \|B\|_2 \cdot \|K_s - K^*\|_2 \\
&\quad + 2 \left[ 1 - \rho(A - BK^*) \right]^{-2} \cdot \|B\|_2 \cdot \left\| b_s - b^*(\mu_s) \right\|_2.
\end{aligned}
\tag{D.6}
$$

By Proposition 3.4, it holds that

$$
\begin{aligned}
\left\| Bb^*(\mu_s) + \overline{A}\mu_s + d \right\|_2 &\leq L_1 \cdot \|B\|_2 \cdot \|\mu_s\|_2 + \|\overline{A}\|_2 \cdot \|\mu_s\|_2 + \|d\|_2 \\
&\leq \left( L_1 \cdot \|B\|_2 + \|\overline{A}\|_2 \right) \cdot \|\mu_s\|_2 + \|d\|_2,
\end{aligned}
\tag{D.7}
$$

where the scalar $L_1$ is defined in Assumption 3.1. Meanwhile, by Theorem B.4, it holds with probability at least $1 - \varepsilon^{10}$ that

$$
\|K_s - K^*\|_{\mathrm{F}} \leq \left[ \sigma_{\min}^{-1}(\Psi_\epsilon) \cdot \sigma_{\min}^{-1}(R) \cdot \varepsilon_s \right]^{1/2}, \qquad \left\| b_s - b^*(\mu_s) \right\|_2 \leq M_b(\mu_s) \cdot \varepsilon_s^{1/2}, \tag{D.8}
$$

where $M_b(\mu_s)$ is defined in (4.3). Combining (D.6), (D.7), (D.8), and the choice of $\varepsilon_s$ in (4.2), it holds with probability at least $1 - \varepsilon^{10}$ that

$$
E_2 \leq \varepsilon/8 \cdot 2^{-s}. \tag{D.9}
$$

**Upper Bound of $E_3$.** By Proposition 3.2, we have

$$
E_3 = \|\mu_{s+1}^* - \mu^*\|_2 = \left\| \Lambda(\mu_s) - \Lambda(\mu^*) \right\|_2 \leq L_0 \cdot \|\mu_s - \mu^*\|_2, \tag{D.10}
$$

where $L_0 = L_1 L_3 + L_2$ by Assumption 3.1.

By plugging (D.2), (D.9), and (D.10) in (D.1), we know that

$$
\|\mu_{s+1} - \mu^*\|_2 \leq L_0 \cdot \|\mu_s - \mu^*\|_2 + \varepsilon \cdot 2^{-s-2}, \tag{D.11}
$$

which holds with probability at least $1 - \varepsilon^{10}$. Following from (D.11) and a union bound argument with $S = \mathcal{O}(\log(1/\varepsilon))$, it holds with probability at least $1 - \varepsilon^5$ that

$$\|\mu_S - \mu^*\|_2 \leq L_0^S \cdot \|\mu_0 - \mu^*\|_2 + \varepsilon/2,$$

where we use the fact that $L_0 < 1$ by Assumption 3.1. By the choice of $S$ in (4.1), it further holds with probability at least $1 - \varepsilon^6$ that

$$\|\mu_S - \mu^*\| \leq \varepsilon. \tag{D.12}$$

By Theorem B.4 and the choice of $\varepsilon_s$ in (4.2), it holds with probability at least $1 - \varepsilon^5$ that

$$\|K_S - K^*\|_F = \|K_S - K^*(\mu_S)\|_F \leq \left[\sigma_{\min}^{-1}(\Psi_\epsilon) \cdot \sigma_{\min}^{-1}(R) \cdot \varepsilon_S\right]^{1/2} \leq \varepsilon. \tag{D.13}$$

Meanwhile, by the triangle inequality and the choice of $\varepsilon_s$ in (4.2), it holds with probability at least $1 - \varepsilon^5$ that

$$
\begin{aligned}
\|b_S - b^*\|_2 &\leq \|b_S - b^*(\mu_S)\|_2 + \|b^*(\mu_S) - b^*\|_2 \\
&\leq M_b(\mu_S) \cdot \varepsilon_S^{1/2} + L_1 \cdot \|\mu_S - \mu^*\|_2 \\
&\leq (1 + L_1) \cdot \varepsilon,
\end{aligned}
\tag{D.14}
$$

where the second inequality comes from Theorem B.4 and Proposition 3.4, and the last inequality comes from (D.12). By (D.12), (D.13), and (D.14), we conclude the proof of the theorem.

## D.2 Proof of Theorem B.4

*Proof.* We first show that $J_1(K_N) - J_1(K^*) < \varepsilon/2$ with a high probability, then show that $J_2(K_N, b_H) - J_2(K^*, b^*) < \varepsilon/2$ with a high probability. Then we have

$$J(K_N, b_N) - J(K^*, b^*) = J_1(K_N) + J_2(K_N, b_H) - J_1(K^*) - J_2(K^*, b^*) < \varepsilon$$

with a high probability, which proves Theorem B.4.

**Part 1.** We show that $J_1(K_N) - J_1(K^*) < \varepsilon/2$ with a high probability.

We first bound $J_1(K_1) - J_1(K_2)$ for any $K_1$ and $K_2$. By Proposition B.2, $J_1(K)$ takes the form of

$$J_1(K) = \mathrm{tr}(P_K \Psi_\epsilon) = \mathbb{E}_{y \sim \mathcal{N}(0, \Psi_\epsilon)}(y^\top P_K y). \tag{D.15}$$

The following lemma calculates $y^\top P_{K_1} y - y^\top P_{K_2} y$ for any $K_1$ and $K_2$.

**Lemma D.1.** Assume that $\rho(A - BK_1) < 1$ and $\rho(A - BK_2) < 1$. For any state vector $y$, we denote by $\{y_t\}_{t \geq 0}$ the sequence generated by the state transition $y_{t+1} = (A - BK_2)y_t$ with initial state $y_0 = y$. It holds that

$$y^\top P_{K_2} y - y^\top P_{K_1} y = \sum_{t \geq 0} D_{K_1, K_2}(y_t),$$

where

$$D_{K_1, K_2}(y) = 2y^\top (K_2 - K_1)(\Upsilon_{K_1}^{22} K_1 - \Upsilon_{K_1}^{21})y + y^\top (K_2 - K_1)^\top \Upsilon_{K_1}^{22}(K_2 - K_1)y.$$

Here $\Upsilon_K$ is defined in (3.7).

*Proof.* See §F.1 for a detailed proof. $\square$

The following lemma shows that $J_1(K)$ is gradient dominant.

**Lemma D.2.** Let $K^*$ be the optimal parameter and $K$ be a parameter such that $J_1(K) < \infty$, then it holds that

$$J_1(K) - J_1(K^*) \leq \sigma_{\min}^{-1}(R) \cdot \|\Phi_{K^*}\|_2 \cdot \mathrm{tr}\left[(\Upsilon_K^{22} K - \Upsilon_K^{21})^\top (\Upsilon_K^{22} K - \Upsilon_K^{21})\right], \tag{D.16}$$

$$J_1(K) - J_1(K^*) \geq \sigma_{\min}(\Psi_\omega) \cdot \|\Upsilon_K^{22}\|_2^{-1} \cdot \mathrm{tr}\left[(\Upsilon_K^{22} K - \Upsilon_K^{21})^\top (\Upsilon_K^{22} K - \Upsilon_K^{21})\right]. \tag{D.17}$$

*Proof.* See §F.2 for a detailed proof. □

Recall that from Algorithm 2, the parameter $K$ is updated via

$$K_{n+1} = K_n - \gamma \cdot (\widehat{\Upsilon}^{22}_{K_n} K_n - \widehat{\Upsilon}^{21}_{K_n}), \tag{D.18}$$

where $\widehat{\Upsilon}_{K_n}$ is the output of Algorithm 3. We upper bound $|J_1(K_{n+1}) - J_1(K^*)|$ in the sequel. First, we show that if $J_1(K_n) - J_1(K^*) \geq \varepsilon/2$ holds for any $n \leq N$, we obtain that

$$J_1(K_N) \leq J_1(K_{N-1}) \leq \cdots \leq J_1(K_0), \tag{D.19}$$

which holds with probability at least $1 - \varepsilon^{13}$. We prove (D.19) by mathematical induction. Suppose that

$$J_1(K_n) \leq J_1(K_{n-1}) \leq \cdots \leq J_1(K_0), \tag{D.20}$$

which holds for $n = 0$. In what follows, we define $\widetilde{K}_{n+1}$ as

$$\widetilde{K}_{n+1} = K_n - \gamma \cdot (\Upsilon^{22}_{K_n} K_n - \Upsilon^{21}_{K_n}), \tag{D.21}$$

where $\Upsilon_{K_n}$ is given in (3.7). By (D.21), we have

$$
\begin{aligned}
J_1(\widetilde{K}_{n+1}) - J_1(K_n) &= \mathbb{E}_{y \sim \mathcal{N}(0, \Psi_\epsilon)} \big[ y^\top (P_{\widetilde{K}_{n+1}} - P_{K_n}) y \big] \\
&= -2\gamma \cdot \operatorname{tr} \big[ \Phi_{\widetilde{K}_{n+1}} \cdot (\Upsilon^{22}_{K_n} K_n - \Upsilon^{21}_{K_n})^\top (\Upsilon^{22}_{K_n} K_n - \Upsilon^{21}_{K_n}) \big] \\
&\quad + \gamma^2 \cdot \operatorname{tr} \big[ \Phi_{\widetilde{K}_{n+1}} \cdot (\Upsilon^{22}_{K_n} K_n - \Upsilon^{21}_{K_n})^\top \Upsilon^{22}_{K_n} (\Upsilon^{22}_{K_n} K_n - \Upsilon^{21}_{K_n}) \big] \\
&\leq -2\gamma \cdot \operatorname{tr} \big[ \Phi_{\widetilde{K}_{n+1}} \cdot (\Upsilon^{22}_{K_n} K_n - \Upsilon^{21}_{K_n})^\top (\Upsilon^{22}_{K_n} K_n - \Upsilon^{21}_{K_n}) \big] \\
&\quad + \gamma^2 \cdot \|\Upsilon^{22}_{K_n}\|_2 \cdot \operatorname{tr} \big[ \Phi_{\widetilde{K}_{n+1}} \cdot (\Upsilon^{22}_{K_n} K_n - \Upsilon^{21}_{K_n})^\top (\Upsilon^{22}_{K_n} K_n - \Upsilon^{21}_{K_n}) \big],
\end{aligned}
\tag{D.22}
$$

where the first equality comes from (D.15), the second equality comes from Lemma D.1, and the last inequality comes from the trace inequality. By the definition of $\Upsilon_K$ in (3.7), we obtain that

$$
\begin{aligned}
\|\Upsilon^{22}_{K_n}\|_2 &\leq \|R\|_2 + \|B\|_2^2 \cdot \|P_{K_n}\|_2 \leq \|R\|_2 + \|B\|_2^2 \cdot J_1(K_n) \cdot \sigma_{\min}^{-1}(\Psi_\epsilon) \\
&\leq \|R\|_2 + \|B\|_2^2 \cdot J_1(K_0) \cdot \sigma_{\min}^{-1}(\Psi_\epsilon),
\end{aligned}
\tag{D.23}
$$

where the second inequality comes from Proposition B.2. By plugging (D.23) and the choice of stepsize $\gamma \leq [\|R\|_2 + \|B\|_2^2 \cdot J_1(K_0) \cdot \sigma_{\min}^{-1}(\Psi_\epsilon)]^{-1}$ into (D.22), we obtain that

$$
\begin{aligned}
J_1(\widetilde{K}_{n+1}) - J_1(K_n) &\leq -\gamma \cdot \operatorname{tr} \big[ \Phi_{\widetilde{K}_{n+1}} \cdot (\Upsilon^{22}_{K_n} K_n - \Upsilon^{21}_{K_n})^\top (\Upsilon^{22}_{K_n} K_n - \Upsilon^{21}_{K_n}) \big] \\
&\leq -\gamma \cdot \sigma_{\min}(\Psi_\epsilon) \cdot \operatorname{tr} \big[ (\Upsilon^{22}_{K_n} K_n - \Upsilon^{21}_{K_n})^\top (\Upsilon^{22}_{K_n} K_n - \Upsilon^{21}_{K_n}) \big] \\
&\leq -\gamma \cdot \sigma_{\min}(\Psi_\epsilon) \cdot \sigma_{\min}(R) \cdot \|\Phi_{K^*}\|_2^{-1} \cdot \big[ J_1(K_n) - J_1(K^*) \big] < 0, \tag{D.24}
\end{aligned}
$$

where the last inequality comes from Lemma D.2.

The following lemma upper bounds $|J_1(\widetilde{K}_{n+1}) - J_1(K_{n+1})|$.

**Lemma D.3.** Assume that $J_1(K_n) \leq J_1(K_0)$. It holds with probability at least $1 - \varepsilon^{15}$ that

$$\big| J_1(\widetilde{K}_{n+1}) - J_1(K_{n+1}) \big| \leq \gamma \cdot \sigma_{\min}(\Psi_\epsilon) \cdot \sigma_{\min}(R) \cdot \|\Phi_{K^*}\|_2^{-1} \cdot \varepsilon/4,$$

where $K_{n+1}$ and $\widetilde{K}_{n+1}$ are defined in (D.18) and (D.21), respectively.

*Proof.* See §F.3 for a detailed proof. □

Combining (D.24) and Lemma D.3, if $J_1(K_n) - J_1(K^*) \geq \varepsilon/2$, it holds with probability at least $1 - \varepsilon^{15}$ that

$$
\begin{aligned}
J_1(K_{n+1}) - J_1(K_n) &\leq J_1(\widetilde{K}_{n+1}) - J_1(K_n) + \big| J_1(\widetilde{K}_{n+1}) - J_1(K_{n+1}) \big| \\
&\leq -\gamma \cdot \sigma_{\min}(\Psi_\epsilon) \cdot \sigma_{\min}(R) \cdot \|\Phi_{K^*}\|_2^{-1} \cdot \varepsilon/4 < 0. \tag{D.25}
\end{aligned}
$$

Combining (D.20) and (D.25), it holds with probability at least $1 - \varepsilon^{15}$ that

$$J_1(K_{n+1}) \leq J_1(K_n) \leq \cdots \leq J_1(K_0).$$

Finally, following from a union bound argument and the choice of $N$ in Theorem B.4, if $J_1(K_n) - J_1(K^*) \geq \varepsilon/2$ holds for any $n \leq N$, we have

$$J_1(K_N) \leq J_1(K_{N-1}) \leq \cdots \leq J_1(K_0),$$

which holds with probability at least $1 - \varepsilon^{13}$. Thus, we complete the proof of (D.19).

Combining (D.24) and (D.25), for $J_1(K_n) - J_1(K^*) \geq \varepsilon/2$, we have

$$J_1(K_{n+1}) - J_1(K^*) \leq \left[1 - \gamma \cdot \sigma_{\min}(\Psi_\epsilon) \cdot \sigma_{\min}(R) \cdot \|\Phi_{K^*}\|_2^{-1}\right] \cdot \left[J_1(K_n) - J_1(K^*)\right],$$

which holds with probability at least $1 - \varepsilon^{13}$. Meanwhile, following from a union bound argument and the choice of $N$ in Theorem B.4, it holds with probability at least $1 - \varepsilon^{11}$ that

$$J_1(K_N) - J_1(K^*) \leq \varepsilon/2. \tag{D.26}$$

The following lemma upper bounds $\|K_N - K^*\|_{\mathrm{F}}$.

**Lemma D.4.** For any $K$, we have

$$\|K - K^*\|_{\mathrm{F}}^2 \leq \sigma_{\min}^{-1}(\Psi_\epsilon) \cdot \sigma_{\min}^{-1}(R) \cdot \left[J_1(K) - J_1(K^*)\right].$$

*Proof.* See §F.4 for a detailed proof. $\qquad\square$

Combining (D.26) and Lemma D.4, we have

$$\|K_N - K^*\|_{\mathrm{F}} \leq \left[\sigma_{\min}^{-1}(\Psi_\epsilon) \cdot \sigma_{\min}^{-1}(R) \cdot \varepsilon/2\right]^{1/2}, \tag{D.27}$$

which holds with probability $1 - \varepsilon^{11}$.

**Part 2.** We show that $J_2(K_N, b_H) - J_2(K^*, b^*) < \varepsilon/2$ with high probability. Following from Proposition 3.4, it holds that $J_2(K^*, b^*) = J_2(K_N, b^{K_N})$. Therefore, it suffices to show that $J_2(K_N, b_H) - J_2(K_N, b^{K_N}) < \varepsilon/2$.

First, we show that if $J_2(K_N, b_h) - J_2(K_N, b^{K_N}) \geq \varepsilon/2$ for any $h \leq H$, we obtain that

$$J_2(K_N, b_H) \leq J_2(K_N, b_{H-1}) \leq \cdots \leq J_2(K_N, b_1) \leq J_2(K_N, b_0), \tag{D.28}$$

which holds with probability at least $1 - \varepsilon^{13}$. We prove (D.28) by mathematical induction. Suppose that

$$J_2(K_N, b_h) \leq J_2(K_N, b_{h-1}) \leq \cdots \leq J_2(K_N, b_0), \tag{D.29}$$

Recall that by Algorithm 2, the parameter $b$ is updated via

$$b_{h+1} = b_h - \gamma^b \cdot \widehat{\nabla}_b J_2(K_N, b_h). \tag{D.30}$$

Here

$$\widehat{\nabla}_b J_2(K_N, b_h) = \widehat{\Upsilon}_{K_N}^{22}(-K_N \widehat{\mu}_{K_N, b_h} + b_h) + \widehat{\Upsilon}_{K_N}^{21} \widehat{\mu}_{K_N, b_h} + \widehat{q}_{K_N, b_h}, \tag{D.31}$$

where $\widehat{\Upsilon}_{K_N}$ and $\widehat{q}_{K_N, b_h}$ are the outputs of Algorithm 3. We define $\widetilde{b}_{h+1}$ as

$$\widetilde{b}_{h+1} = b_h - \gamma^b \cdot \nabla_b J_2(K_N, b_h). \tag{D.32}$$

Here

$$\nabla_b J_2(K_N, b_h) = \Upsilon_{K_N}^{22}(-K_N \mu_{K_N, b_h} + b_h) + \Upsilon_{K_N}^{21} \mu_{K_N, b_h} + q_{K_N, b_h}, \tag{D.33}$$

where $\Upsilon_{K_N}$ and $q_{K_N, b_h}$ are defined in (3.7). We upper bound $J_2(K_N, b_{h+1}) - J_2(K_N, b^{K_N})$ in the sequel. Following from (D.32) and Proposition 3.3, we have

$$\begin{aligned}
J_2(K_N, \widetilde{b}_{h+1}) - J_2(K_N, b_h) &\leq -\gamma^b/2 \cdot \left\|\nabla_b J_2(K_N, b_h)\right\|_2^2 \\
&\leq -\nu_{K_N} \cdot \gamma^b \cdot \left[J_2(K_N, b_h) - J_2(K_N, b^{K_N})\right] \\
&\leq -\nu_{K_N} \cdot \gamma^b \cdot \varepsilon < 0, \tag{D.34}
\end{aligned}$$

where $\nu_{K_N}$ is specified in Proposition 3.3. The following lemma upper bounds $|J_2(K_N, b_{h+1}) - J_2(K_N, \widetilde{b}_{h+1})|$.

**Lemma D.5.** Assume that $J_2(K_N, b_h) \leq J_2(K_N, b_0)$. It holds with probability at least $1 - \varepsilon^{15}$ that

$$\left| J_2(K_N, b_{h+1}) - J_2(K_N, \widetilde{b}_{h+1}) \right| \leq \nu_{K_N} \cdot \gamma^b \cdot \varepsilon/2,$$

where $b_{h+1}$ and $\widetilde{b}_{h+1}$ are defined in (D.30) and (D.32), respectively.

*Proof.* See §F.5 for a detailed proof. □

Combining (D.34) and Lemma D.5, we know that if $J_2(K_N, b_h) - J_2(K_N, b^{K_N}) \geq \varepsilon$, it holds with probability at least $1 - \varepsilon^{15}$ that

$$J_2(K_N, b_{h+1}) - J_2(K_N, b_h) \leq J_2(K_N, \widetilde{b}_{h+1}) - J_2(K_N, b_h) + \left| J_2(K_N, b_{h+1}) - J_2(K_N, \widetilde{b}_{h+1}) \right|$$
$$\leq -\nu_{K_N} \cdot \gamma^b \cdot \varepsilon/2 < 0. \tag{D.35}$$

Combining (D.29) and (D.35), it holds with probability at least $1 - \varepsilon^{15}$ that

$$J_2(K_N, b_{h+1}) \leq J_2(K_N, b_h) \leq \cdots \leq J_2(K_N, b_0).$$

Following from a union bound argument and the choice of $H$ in Theorem B.4, if $J_2(K_N, b_h) - J_2(K_N, b^{K_N}) \geq \varepsilon$ holds for any $h \leq H$, we have

$$J_2(K_N, b_H) \leq J_2(K_N, b_{H-1}) \leq \cdots \leq J_2(K_N, b_0),$$

which holds with probability at least $1 - \varepsilon^{13}$. Thus, we finish the proof of (D.28).

Combining (D.34) and Lemma D.5, for $J_2(K_N, b_h) - J_2(K_N, b^{K_N}) \geq \varepsilon/2$, we have

$$J_2(K_N, b_{h+1}) - J_2(K_N, b^{K_N}) \leq (1 - \nu_{K_N} \cdot \gamma^b) \cdot \left[ J_2(K_N, b_h) - J_2(K_N, b^{K_N}) \right],$$

which holds with probability at least $1 - \varepsilon^{13}$. Meanwhile, following from a union bound argument and the choice of $H$ in Theorem B.4, it holds with probability at least $1 - \varepsilon^{11}$ that

$$J_2(K_N, b_H) - J_2(K_N, b^{K_N}) \leq \varepsilon/2. \tag{D.36}$$

By Proposition 3.3 and (D.36), it holds with probability at least $1 - \varepsilon^{11}$ that

$$\|b_H - b^{K_N}\|_2 \leq (2\varepsilon/\nu_{K^*})^{1/2}. \tag{D.37}$$

Following from Proposition 3.4, we know that

$$b^{K_N} - b^* = (K_N - K^*)Q^{-1}(I - A)^\top \tag{D.38}$$
$$\cdot \left[ (I - A)Q^{-1}(I - A)^\top + BR^{-1}B^\top \right]^{-1} \cdot (\overline{A}\mu + d).$$

Combining (D.27), (D.37), and (D.38), it holds with probability $1 - \varepsilon^{10}$ that

$$\|b_H - b^{K_N}\|_2 \leq M_b \cdot \varepsilon^{1/2},$$

where

$$M_b(\mu) = 4 \left\| Q^{-1}(I - A)^\top \cdot \left[ (I - A)Q^{-1}(I - A)^\top + BR^{-1}B^\top \right]^{-1} \cdot (\overline{A}\mu + d) \right\|_2$$
$$\cdot \left[ \nu_{K^*}^{-1} + \sigma_{\min}^{-1}(\Psi_\epsilon) \cdot \sigma_{\min}^{-1}(R) \right]^{1/2}.$$

We finish the proof of the theorem. □

### D.3 PROOF OF THEOREM B.8

*Proof.* We follow the proof of Theorem 4.2 in Yang et al. (2019), where they only consider LQR without drift terms. Since our proof requires much more delicate analysis, we present it here.

**Part 1.** We denote by $\widehat{\zeta}$ and $\widehat{\xi}$ the primal and dual variables generated by Algorithm 3. We define the primal-dual gap of (B.11) as

$$\text{gap}(\widehat{\zeta}, \widehat{\xi}) = \max_{\xi \in \mathcal{V}_\xi} F(\widehat{\zeta}, \xi) - \min_{\zeta \in \mathcal{V}_\zeta} F(\zeta, \widehat{\xi}). \tag{D.39}$$

In the sequel, we upper bound $\|\widehat{\alpha}_{K,b} - \alpha_{K,b}\|_2$ using (D.39).

We define $\zeta_{K,b}$ and $\xi(\zeta)$ as

$$\zeta_{K,b} = \big(J(K,b), \alpha_{K,b}^\top\big)^\top, \qquad \xi(\zeta) = \operatorname*{argmax}_{\xi} F(\zeta, \xi). \tag{D.40}$$

Following from (B.12), we know that

$$\xi^1(\zeta) = \zeta^1 - J(K,b), \quad \xi^2(\zeta) = \mathbb{E}_{\pi_{K,b}}\big[\psi(x,u)\big]\zeta^1 + \Theta_{K,b}\zeta^2 - \mathbb{E}_{\pi_{K,b}}\big[c(x,u)\psi(x,u)\big]. \tag{D.41}$$

The following lemma shows that $\zeta_{K,b} \in \mathcal{V}_\zeta$ and $\xi(\zeta) \in \mathcal{V}_\xi$ for any $\zeta \in \mathcal{V}_\zeta$.

**Lemma D.6.** Under the assumptions in Theorem B.8, it holds that $\zeta_{K,b} = (J(K,b), \alpha_{K,b}^\top)^\top \in \mathcal{V}_\zeta$. Also, for any $\zeta \in \mathcal{V}_\zeta$, the vector $\xi(\zeta)$ defined in (D.40) satisfies that $\xi(\zeta) \in \mathcal{V}_\xi$.

*Proof.* See §F.6 for a detailed proof. $\square$

By (B.12), we know that $\nabla_\zeta F(\zeta_{K,b}, 0) = 0$ and $\nabla_\xi F(\zeta_{K,b}, 0) = 0$. Combining Lemma D.6, it holds that $(\zeta_{K,b}, 0)$ is a saddle point of the function $F(\zeta, \xi)$ defined in (B.11).

Following from (D.39), it holds that

$$\Big\|\mathbb{E}_{\pi_{K,b}}\big[\psi(x,u)\big]\widehat{\zeta}^1 + \Theta_{K,b}\widehat{\zeta}^2 - \mathbb{E}_{\pi_{K,b}}\big[c(x,u)\psi(x,u)\big]\Big\|_2^2 + \big|\widehat{\zeta}^1 - J(K,b)\big|^2$$
$$= F\big(\widehat{\zeta}, \xi(\widehat{\zeta})\big) = \max_{\xi \in \mathcal{V}_\xi} F(\widehat{\zeta}, \xi) = \operatorname{gap}(\widehat{\zeta}, \widehat{\xi}) + \min_{\zeta \in \mathcal{V}_\zeta} F(\zeta, \widehat{\xi}), \tag{D.42}$$

where the first equality comes from (D.41), and the second equality comes from the fact that $\xi(\widehat{\zeta}) = \operatorname{argmax}_{\xi \in \mathcal{V}_\xi} F(\widehat{\zeta}, \xi)$ by (D.40) and Lemma D.6. We upper bound the RHS of (D.42) and lower bound the LHS of (D.42) in the sequel.

As for the RHS of (D.42), it holds for any $\xi \in \mathcal{V}_\xi$ that

$$\min_{\zeta \in \mathcal{V}_\zeta} F(\zeta, \xi) \le \min_{\zeta \in \mathcal{V}_\zeta} \max_{\xi \in \mathcal{V}_\xi} F(\zeta, \xi) = \min_{\zeta \in \mathcal{V}_\zeta} F\big(\zeta, \xi(\zeta)\big)$$
$$= \frac{1}{2} \min_{\zeta \in \mathcal{V}_\zeta} \left\{ \Big\|\mathbb{E}_{\pi_{K,b}}\big[\psi(x,u)\big]\zeta^1 + \Theta_{K,b}\zeta^2 - \mathbb{E}_{\pi_{K,b}}\big[c(x,u)\psi(x,u)\big]\Big\|_2^2 + \big|\zeta^1 - J(K,b)\big|^2 \right\}$$
$$= 0, \tag{D.43}$$

where the first equality comes from the fact that $\xi(\zeta) = \operatorname{argmax}_{\xi \in \mathcal{V}_\xi} F(\zeta, \xi)$ by (D.40) and Lemma D.6, the second equality comes from (D.41), and the last equality holds by taking $\zeta = \zeta_{K,b} \in \mathcal{V}_\zeta$. Meanwhile, we lower bound the LHS of (D.42) as

$$\Big\|\mathbb{E}_{\pi_{K,b}}\big[\psi(x,u)\big]\widehat{\zeta}^1 + \Theta_{K,b}\widehat{\zeta}^2 - \mathbb{E}_{\pi_{K,b}}\big[c(x,u)\psi(x,u)\big]\Big\|_2^2 + \big|\widehat{\zeta}^1 - J(K,b)\big|^2$$
$$= \big\|\widetilde{\Theta}_{K,b}(\widehat{\zeta} - \zeta_{K,b})\big\|_2^2 \ge \lambda_K^2 \cdot \|\widehat{\zeta} - \zeta_{K,b}\|_2^2 \ge \lambda_K^2 \cdot \|\widehat{\alpha}_{K,b} - \alpha_{K,b}\|_2^2, \tag{D.44}$$

where the first equality comes from the definition of $\widetilde{\Theta}_{K,b}$ in (B.9), and the first inequality comes from Proposition B.6. Here $\lambda_K$ is defined in Proposition B.6. Combining (D.42), (D.43), and (D.44), it holds that

$$\|\widehat{\alpha}_{K,b} - \alpha_{K,b}\|_2^2 \le \lambda_K^{-2} \cdot \operatorname{gap}(\widehat{\zeta}, \widehat{\xi}), \tag{D.45}$$

which finishes the proof of this part.

**Part 2.** We now upper bound $\operatorname{gap}(\widehat{\zeta}, \widehat{\xi})$. We denote by $\widetilde{z}_t = (\widetilde{x}_t^\top, \widetilde{u}_t^\top)^\top$ for $t \in [\widetilde{T}]$, where $\widetilde{x}_t$ and $\widetilde{u}_t$ are generated in Line 6 of Algorithm 3. Following from the state transition in Problem 2.3 and the form of the linear policy, $\{\widetilde{z}_t\}_{t \in [\widetilde{T}]}$ follows the following transition,

$$\widetilde{z}_{t+1} = L\widetilde{z}_t + \nu + \delta_t, \tag{D.46}$$

where

$$\nu = \begin{pmatrix} \overline{A}\mu + d \\ -K(\overline{A}\mu + d) + b \end{pmatrix}, \qquad \delta_t = \begin{pmatrix} \omega_t \\ -K\omega_t + \sigma\eta \end{pmatrix}, \qquad L = \begin{pmatrix} A & B \\ -KA & -KB \end{pmatrix}.$$

Note that we have

$$L = \begin{pmatrix} A & B \\ -KA & -KB \end{pmatrix} = \begin{pmatrix} I \\ -K \end{pmatrix} (A \quad B).$$

Then by the property of spectral radius, it holds that

$$\rho(L) = \rho\left( (A \quad B) \begin{pmatrix} I \\ -K \end{pmatrix} \right) = \rho(A - BK) < 1.$$

Thus, the Markov chain generated by (D.46) admits a unique stationary distribution $\mathcal{N}(\mu_z, \Sigma_z)$, where

$$\mu_z = (I - L)^{-1}\nu, \qquad \Sigma_z = L\Sigma_z L^\top + \begin{pmatrix} \Psi_\omega & -\Psi_\omega K^\top \\ -K\Psi_\omega & K\Psi_\omega K^\top + \sigma^2 I \end{pmatrix}. \qquad (D.47)$$

The following lemma characterizes the average

$$\widehat{\mu}_z = 1/\widetilde{T} \cdot \sum_{t=1}^{\widetilde{T}} \widetilde{z}_t. \qquad (D.48)$$

**Lemma D.7.** It holds that

$$\widehat{\mu}_z \sim \mathcal{N}\left( \mu_z + \frac{1}{\widetilde{T}}\mu_{\widetilde{T}}, \ \frac{1}{\widetilde{T}}\widetilde{\Sigma}_{\widetilde{T}} \right),$$

where $\|\mu_{\widetilde{T}}\|_2 \le M_\mu \cdot (1 - \rho)^{-2} \cdot \|\mu_z\|_2$ and $\|\widetilde{\Sigma}_{\widetilde{T}}\|_F \le M_\Sigma \cdot (1 - \rho)^{-1} \cdot \|\Sigma_z\|_F$. Here $M_\mu$ and $M_\Sigma$ are positive absolute constants. Moreover, it holds with probability at least $1 - \widetilde{T}^{-6}$ that

$$\|\widehat{\mu}_z - \mu_z\|_2 \le \frac{\log \widetilde{T}}{\widetilde{T}^{1/4}} \cdot (1 - \rho)^{-2} \cdot \text{poly}\big(\|\Phi_K\|_2, \|K\|_F, \|b\|_2, \|\mu\|_2\big).$$

*Proof.* See §F.7 for a detailed proof. □

Lemma D.7 gives that

$$\|\widehat{\mu}_{K,b} - \mu_{K,b}\|_2 \le \frac{\log \widetilde{T}}{\widetilde{T}^{1/4}} \cdot (1 - \rho)^{-2} \cdot \text{poly}\big(\|\Phi_K\|_2, \|K\|_F, \|b\|_2, \|\mu\|_2\big),$$

which holds with probability at least $1 - \widetilde{T}^{-6}$.

We now apply a truncation argument to show that $\text{gap}(\widehat{\zeta}, \widehat{\xi})$ is upper bounded. We define the event $\mathcal{E}$ in the sequel. Following from Lemma D.7, it holds for any $z \sim \mathcal{N}(\mu_z, \Sigma_z)$ that

$$z - \widehat{\mu}_z + 1/\widetilde{T} \cdot \mu_{\widetilde{T}} \sim \mathcal{N}(0, \Sigma_z + 1/\widetilde{T} \cdot \widetilde{\Sigma}_{\widetilde{T}}).$$

By Lemma G.3, there exists a positive absolute constant $C_0$ such that

$$\mathbb{P}\Big[ \big| \|z - \widehat{\mu}_z + 1/\widetilde{T} \cdot \mu_{\widetilde{T}}\|_2^2 - \text{tr}(\widetilde{\Sigma}_z) \big| > \tau \Big] \le 2\exp\Big[ -C_0 \cdot \min\big(\tau^2 \|\widetilde{\Sigma}_z\|_F^{-2}, \ \tau \|\widetilde{\Sigma}_z\|_2^{-1}\big) \Big], \quad (D.49)$$

where we write $\widetilde{\Sigma}_z = \Sigma_z + 1/\widetilde{T} \cdot \widetilde{\Sigma}_{\widetilde{T}}$ for notational convenience. By taking $\tau = C_1 \cdot \log T \cdot \|\widetilde{\Sigma}_z\|_F$ in (D.49) for a sufficiently large positive absolute constant $C_1$, it holds that

$$\mathbb{P}\Big[ \big| \|z - \widehat{\mu}_z + 1/\widetilde{T} \cdot \mu_{\widetilde{T}}\|_2^2 - \text{tr}(\widetilde{\Sigma}_z) \big| > C_1 \cdot \log T \cdot \|\widetilde{\Sigma}_z\|_F \Big] \le T^{-6}. \qquad (D.50)$$

We define the event $\mathcal{E}_{t,1}$ for any $t \in [T]$ as

$$\mathcal{E}_{t,1} = \Big\{ \big| \|z_t - \widehat{\mu}_z + 1/\widetilde{T} \cdot \mu_{\widetilde{T}}\|_2^2 - \text{tr}(\widetilde{\Sigma}_z) \big| \le C_1 \cdot \log T \cdot \|\widetilde{\Sigma}_z\|_F \Big\}.$$

Then by (D.50), it holds for any $t \in [T]$ that

$$\mathbb{P}(\mathcal{E}_{t,1}) \geq 1 - T^{-6}. \tag{D.51}$$

Also, we define

$$\mathcal{E}_1 = \bigcap_{t \in [T]} \mathcal{E}_{t,1}. \tag{D.52}$$

Following from a union bound argument and (D.51), it holds that

$$\mathbb{P}(\mathcal{E}_1) \geq 1 - T^{-5}. \tag{D.53}$$

Also, conditioning on $\mathcal{E}_1$, it holds for sufficiently large $\widetilde{T}$ that

$$
\begin{aligned}
\max_{t \in [T]} &\|z_t - \widehat{\mu}_z\|_2^2 \\
&\leq C_1 \cdot \log T \cdot \|\widetilde{\Sigma}_z\|_{\mathrm{F}} + \mathrm{tr}(\widetilde{\Sigma}_z) + \|1/\widetilde{T} \cdot \mu_{\widetilde{T}}\|_2^2 \\
&\leq 2\widetilde{C}_1 \cdot \left[1 + M_\Sigma(1-\rho)^{-1}/\widetilde{T}^2\right] \cdot \log T \cdot \|\Sigma_z\|_2 + M_\mu(1-\rho)^{-2}/\widetilde{T}^2 \cdot \|\mu_z\|_2^2 \\
&\leq C_2 \cdot \log T \cdot \left(1 + \|K\|_{\mathrm{F}}^2\right) \cdot \|\Phi_K\|_2 \cdot (1-\rho)^{-1} + C_3 \cdot \left(\|b\|_2^2 + \|\mu\|_2^2\right) \cdot (1-\rho)^{-4} \cdot \widetilde{T}^{-2} \\
&\leq 2C_2 \cdot \log T \cdot \left(1 + \|K\|_{\mathrm{F}}^2\right) \cdot \|\Phi_K\|_2 \cdot (1-\rho)^{-1},
\end{aligned} \tag{D.54}
$$

where $\widetilde{C}_1$, $C_2$, and $C_3$ are positive absolute constants. Here, the first inequality comes from the definition of $\mathcal{E}_1$ in (D.52), the second inequality comes from Lemma D.7, and the third inequality comes from (D.47). Also, we define the following event

$$\mathcal{E}_2 = \left\{\|\widehat{\mu}_z - \mu_z + 1/\widetilde{T} \cdot \mu_{\widetilde{T}}\|_2 \leq C_1\right\}. \tag{D.55}$$

Then by Lemma D.7, we know that

$$\mathbb{P}(\mathcal{E}_2) \geq 1 - \widetilde{T}^{-6} \tag{D.56}$$

for $\widetilde{T}$ sufficiently large. We define the event $\mathcal{E}$ as

$$\mathcal{E} = \mathcal{E}_1 \bigcap \mathcal{E}_2.$$

Then following from (D.53), (D.56), and a union bound argument, we know that

$$\mathbb{P}(\mathcal{E}) \geq 1 - T^{-5} - \widetilde{T}^{-6}.$$

Now, we define the truncated feature vector $\widetilde{\psi}(x, u)$ as $\widetilde{\psi}(x, u) = \widehat{\psi}(x, u) \mathbb{1}_\mathcal{E}$, the truncated cost function $\widetilde{c}(x, u)$ as $\widetilde{c}(x, u) = c(x, u) \mathbb{1}_\mathcal{E}$, and also the truncated objective function $\widetilde{F}(\zeta, \xi)$ as

$$\widetilde{F}(\zeta, \xi) = \left\{\mathbb{E}(\widetilde{\psi})\zeta^1 + \mathbb{E}\left[(\widetilde{\psi} - \widetilde{\psi}')\widetilde{\psi}^\top\right]\zeta^2 - \mathbb{E}(\widetilde{c}\widetilde{\psi})\right\}^\top \xi^2 + \left[\zeta^1 - \mathbb{E}(\widetilde{c})\right] \cdot \xi^1 - \|\xi\|_2^2/2, \tag{D.57}$$

where we write $\widetilde{\psi} = \widetilde{\psi}(x, u)$ and $\widetilde{c} = \widetilde{c}(x, u)$ for notational convenience. Here the expectation is taken following the policy $\pi_{K,b}$ and the state transition. The following lemma establishes the upper bound of $|F(\zeta, \xi) - \widetilde{F}(\zeta, \xi)|$, where $F(\zeta, \xi)$ and $\widetilde{F}(\zeta, \xi)$ are defined in (B.11) and (D.57), respectively.

**Lemma D.8.** It holds with probability at least $1 - \widetilde{T}^{-6}$ that

$$\left|F(\zeta, \xi) - \widetilde{F}(\zeta, \xi)\right| \leq \left(\frac{1}{2T} + \frac{\log \widetilde{T}}{\widetilde{T}^{1/4}}\right) \cdot (1-\rho)^{-2} \cdot \mathrm{poly}\left(\|K\|_{\mathrm{F}}, \|b\|_2, \|\mu\|_2, J(K_0, b_0)\right).$$

*Proof.* See §F.8 for a detailed proof. □

Following from (D.39) and Lemma D.8, it holds with probability at least $1 - \widetilde{T}^{-6}$ that

$$
\begin{aligned}
\left|\mathrm{gap}(\widehat{\zeta}, \widehat{\xi}) - \widetilde{\mathrm{gap}}(\widehat{\zeta}, \widehat{\xi})\right| \\
\leq \left(\frac{1}{2T} + \frac{\log \widetilde{T}}{\widetilde{T}^{1/4}}\right) \cdot (1-\rho)^{-2} \cdot \mathrm{poly}\left(\|K\|_{\mathrm{F}}, \|b\|_2, \|\mu\|_2, J(K_0, b_0)\right).
\end{aligned} \tag{D.58}
$$

where we define $\widetilde{\mathrm{gap}}(\widehat{\zeta}, \widehat{\xi})$ as

$$\widetilde{\mathrm{gap}}(\widehat{\zeta}, \widehat{\xi}) = \max_{\xi \in \mathcal{V}_\xi} \widetilde{F}(\widehat{\zeta}, \xi) - \min_{\zeta \in \mathcal{V}_\zeta} \widetilde{F}(\zeta, \widehat{\xi}).$$

Therefore, to upper bound of $\mathrm{gap}(\zeta, \xi)$, we only need to upper bound $\widetilde{\mathrm{gap}}(\zeta, \xi)$.

**Part 3.** We upper bound $\widetilde{\mathrm{gap}}(\zeta, \xi)$ in the sequel. We first show that the trajectory generated by the policy $\pi_{K,b}$ and the state transition in Problem 2.2 is $\beta$-mixing.

**Lemma D.9.** Consider a linear system $y_{t+1} = Dy_t + \vartheta + \upsilon_t$, where $\{y_t\}_{t \geq 0} \subset \mathbb{R}^m$, the matrix $D \in \mathbb{R}^{m \times m}$ satisfying $\rho(D) < 1$, the vector $\vartheta \in \mathbb{R}^m$, and $\upsilon_t \sim \mathcal{N}(0, \Sigma)$ is the Gaussians. We denote by $\varpi_t$ the marginal distribution of $y_t$ for any $t \geq 0$. Meanwhile, assume that the stationary distribution of $\{y_t\}_{t \geq 0}$ is a Gaussian distribution $\mathcal{N}((I - D)^{-1}\vartheta, \Sigma_\infty)$, where $\Sigma_\infty$ is the covariance matrix. We define the $\beta$-mixing coefficients for any $n \geq 1$ as follows

$$\beta(n) = \sup_{t \geq 0} \mathbb{E}_{y \sim \varpi_t} \Big[ \big\| \mathbb{P}_{y_n}(\cdot \,|\, y_0 = y) - \mathbb{P}_{\mathcal{N}((I-D)^{-1}\vartheta, \Sigma_\infty)}(\cdot) \big\|_{\mathrm{TV}} \Big].$$

Then, for any $\rho \in (\rho(D), 1)$, the $\beta$-mixing coefficients satisfy that

$$\beta(n) \leq C_{\rho, D, \vartheta} \cdot \big[ \mathrm{tr}(\Sigma_\infty) + m \cdot (1 - \rho)^{-2} \big]^{1/2} \cdot \rho^n,$$

where $C_{\rho, D, \vartheta}$ is a constant, which only depends on $\rho$, $D$, and $\vartheta$. We say that the sequence $\{y_t\}_{t \geq 0}$ is $\beta$-mixing with parameter $\rho$.

*Proof.* See Proposition 3.1 in Tu and Recht (2017) for details. $\square$

Recall that by (3.1), the sequence $\{x_t\}_{t \geq 0}$ follows

$$x_{t+1} = (A - BK)x_t + (Bb + \overline{A}\mu + d) + \epsilon_t, \qquad \epsilon_t \sim \mathcal{N}(0, \Psi_\epsilon),$$

where the matrix $A - BK$ satisfies that $\rho(A - BK) < 1$. Therefore, by Lemma D.9, the sequence $\{z_t\}_{t \geq 0}$ is $\beta$-mixing with parameter $\rho \in (\rho(A - BK), 1)$, where $z_t = (x_t^\top, u_t^\top)^\top$. The following lemma upper bounds the primal-dual gap for a convex-concave problem.

**Lemma D.10.** Let $\mathcal{X}$ and $\mathcal{Y}$ be two compact and convex sets such that $\|x - x'\|_2 \leq M$ and $\|y - y'\|_2 \leq M$ for any $x, x' \in \mathcal{X}$ and $y, y' \in \mathcal{Y}$. We consider solving the following minimax problem

$$\min_{x \in \mathcal{X}} \max_{y \in \mathcal{Y}} H(x, y) = \mathbb{E}_{\epsilon \sim \varpi_\epsilon} \big[ G(x, y; \epsilon) \big],$$

where the objective function $H(x, y)$ is convex in $x$ and concave in $y$. In addition, we assume that the distribution $\varpi_\epsilon$ is $\beta$-mixing with $\beta(n) \leq C_\epsilon \cdot \rho^n$, where $C_\epsilon$ is a constant. Meanwhile, we assume that it holds almost surely that $G(x, y; \epsilon)$ is $\widetilde{L}_0$-Lipschitz in both $x$ and $y$, the gradient $\nabla_x G(x, y; \epsilon)$ is $\widetilde{L}_1$-Lipschitz in $x$ for any $y \in \mathcal{Y}$, the gradient $\nabla_y G(x, y; \epsilon)$ is $\widetilde{L}_1$-Lipschitz in $y$ for any $x \in \mathcal{X}$, where $C_\epsilon, \widetilde{L}_0, \widetilde{L}_1 > 1$. Each step of our gradient-based method takes the following forms,

$$x_{t+1} = \Gamma_\mathcal{X} \big[ x_t - \gamma_{t+1} \cdot \nabla_x G(x_t, y_t; \epsilon_t) \big], \qquad y_{t+1} = \Gamma_\mathcal{Y} \big[ y_t - \gamma_{t+1} \cdot \nabla_y G(x_t, y_t; \epsilon_t) \big],$$

where the operators $\Gamma_\mathcal{X}$ and $\Gamma_\mathcal{Y}$ projects the variables back to $\mathcal{X}$ and $\mathcal{Y}$, respectively, and the stepsizes take the form $\gamma_t = \gamma_0 \cdot t^{-1/2}$ for a constant $\gamma_0 > 0$. Moreover, let $\widehat{x} = (\sum_{t=1}^T \gamma_t)^{-1}(\sum_{t=1}^T \gamma_t x_t)$ and $\widehat{y} = (\sum_{t=1}^T \gamma_t)^{-1}(\sum_{t=1}^T \gamma_t y_t)$ be the final output of the gradient method after $T$ iterations, then there exists a positive absolute constant $C$, such that for any $\delta \in (0, 1)$, the primal-dual gap to the minimax problem is upper bounded as

$$\max_{x \in \mathcal{X}} H(\widehat{x}, y) - \min_{y \in \mathcal{Y}} H(x, \widehat{y}) \leq \frac{C \cdot (M^2 + \widetilde{L}_0^2 + \widetilde{L}_0 \widetilde{L}_1 M)}{\log(1/\rho)} \cdot \frac{\log^2 T + \log(1/\delta)}{\sqrt{T}} + \frac{C \cdot C_\epsilon \widetilde{L}_0 M}{T},$$

which holds with probability at least $1 - \delta$.

*Proof.* See Theorem 5.4 in Yang et al. (2019) for details. $\square$

To use Lemma D.10, we define the function $G(\zeta, \xi; \widetilde{\psi}, \widetilde{\psi}')$ as

$$G(\zeta, \xi; \widetilde{\psi}, \widetilde{\psi}') = \left[\widetilde{\psi}\zeta^1 + (\widetilde{\psi} - \widetilde{\psi}')\widetilde{\psi}^\top \zeta^2 - \widetilde{c}\widetilde{\psi}\right]^\top \xi^2 + (\zeta^1 - \widetilde{c}) \cdot \xi^1 - 1/2 \cdot \|\xi\|_2^2,$$

where $\widetilde{\psi} = \widetilde{\psi}(x, u)$ and $\widetilde{\psi}' = \widetilde{\psi}(x', u')$. Note that the gradients of $G(\zeta, \xi; \widetilde{\psi}, \widetilde{\psi}')$ take the form

$$\nabla_\zeta G(\zeta, \xi; \widetilde{\psi}, \widetilde{\psi}') = \begin{pmatrix} \widetilde{\psi}^\top \xi^2 + \xi^1 \\ \widetilde{\psi}(\widetilde{\psi} - \widetilde{\psi}')^\top \xi^2 \end{pmatrix}, \quad \nabla_\xi G(\zeta, \xi; \widetilde{\psi}, \widetilde{\psi}') = \begin{pmatrix} \zeta^1 - \widetilde{c} - \xi^1 \\ \widetilde{\psi}\zeta^1 + (\widetilde{\psi} - \widetilde{\psi}')\widetilde{\psi}^\top \zeta^2 - \widetilde{c}\widetilde{\psi} - \xi^2 \end{pmatrix}.$$

By Definition B.7 and Lemma D.6, we know that

$$\left\|\nabla_\zeta G(\zeta, \xi; \widetilde{\psi}, \widetilde{\psi}')\right\|_2 \leq \text{poly}\big(\|K\|_{\text{F}}, J(K_0, b_0)\big) \cdot \log^2 T \cdot (1 - \rho)^{-2},$$
$$\left\|\nabla_\xi G(\zeta, \xi; \widetilde{\psi}, \widetilde{\psi}')\right\|_2 \leq \text{poly}\big(\|K\|_{\text{F}}, \|\mu\|_2, J(K_0, b_0)\big) \cdot \log^2 T \cdot (1 - \rho)^{-2}. \tag{D.59}$$

This gives the Lipschitz constant $\widetilde{L}_0$ in Lemma D.10 for $G(\zeta, \xi; \widetilde{\psi}, \widetilde{\psi}')$. Also, the Hessians of $G(\zeta, \xi; \widetilde{\psi}, \widetilde{\psi}')$ take the forms of

$$\nabla^2_{\zeta\zeta} G(\zeta, \xi; \widetilde{\psi}, \widetilde{\psi}') = 0, \qquad \nabla^2_{\xi\xi} G(\zeta, \xi; \widetilde{\psi}, \widetilde{\psi}') = -I,$$

which follows that

$$\left\|\nabla^2_{\zeta\zeta} G(\zeta, \xi; \widetilde{\psi}, \widetilde{\psi}')\right\|_2 = 0, \qquad \left\|\nabla^2_{\xi\xi} G(\zeta, \xi; \widetilde{\psi}, \widetilde{\psi}')\right\|_2 = 1. \tag{D.60}$$

This gives the Lipschitz constant $\widetilde{L}_1$ in Lemma D.10 for $\nabla_\zeta G(\zeta, \xi; \widetilde{\psi}, \widetilde{\psi}')$ and $\nabla_\xi G(\zeta, \xi; \widetilde{\psi}, \widetilde{\psi}')$. Moreover, note that (D.54) provides an upper bound of $M$, combining (D.59), (D.60) and Lemma D.10, it holds with probability at least $1 - T^{-5}$ that

$$\widetilde{\text{gap}}(\widehat{\zeta}, \widehat{\xi}) \leq \frac{\text{poly}\big(\|K\|_{\text{F}}, \|\mu\|_2, J(K_0, b_0)\big) \cdot \log^6 T}{(1 - \rho)^4 \cdot \sqrt{T}}. \tag{D.61}$$

Combining (D.45), (D.58), and (D.61), we know that

$$\|\widehat{\alpha}_{K,b} - \alpha_{K,b}\|_2^2 \leq \lambda_K^{-2} \cdot \text{poly}_1\big(\|K\|_{\text{F}}, \|b\|_2, \|\mu\|_2, J(K_0, b_0)\big) \cdot \left[\frac{\log^6 T}{T^{1/2} \cdot (1 - \rho)^4} + \frac{\log \widetilde{T}}{\widetilde{T}^{1/4} \cdot (1 - \rho)^2}\right].$$

Same bounds for $\|\widehat{\Upsilon}_K - \Upsilon_K\|_{\text{F}}^2$, $\|\widehat{p}_{K,b} - p_{K,b}\|_2^2$, and $\|\widehat{q}_{K,b} - q_{K,b}\|_2^2$ hold. We finish the proof of the theorem. $\square$

# E    PROOFS OF PROPOSITIONS

## E.1    PROOF OF PROPOSITION 3.2

*Proof.* We follow a similar proof as in the one of Theorem 1.1 in Sznitman (1991) and Theorem 3.2 in Bensoussan et al. (2016). Note that for any policy $\pi_{K,b} \in \Pi$, the parameters $K$ and $b$ uniquely determine the policy. We define the following metric on $\Pi$.

**Definition E.1.** For any $\pi_{K_1,b_1}, \pi_{K_2,b_2} \in \Pi$, we define the following metric,

$$\|\pi_{K_1,b_1} - \pi_{K_2,b_2}\|_2 = c_1 \cdot \|K_1 - K_2\|_2 + c_2 \cdot \|b_1 - b_2\|_2,$$

where $c_1$ and $c_2$ are positive constants.

One can verify that Definition E.1 satisfies the requirement of being a metric. We first evaluate the forms of the operators $\Lambda_1(\cdot)$ and $\Lambda_2(\cdot, \cdot)$.

**Forms of the operators $\Lambda_1(\cdot)$ and $\Lambda_2(\cdot, \cdot)$.** By the definition of $\Lambda_1(\mu)$, which gives the optimal policy under the mean-field state $\mu$, it holds that

$$\Lambda_1(\mu) = \pi_\mu^*,$$

where $\pi_\mu^*$ solves Problem 2.2. This gives the form of $\Lambda_1(\cdot)$. We now turn to $\Lambda_2(\mu, \pi)$, which gives the mean-field state $\mu_{\text{new}}$ generated by the policy $\pi$ under the current mean-field state $\mu$. In Problem

2.2, the sequence of states $\{x_t\}_{t \geq 0}$ constitutes a Markov chain, which admits a unique stationary distribution. Thus, by the state transition in Problem 2.2 and the form of the linear-Gaussian policy, we have

$$\mu_{\text{new}} = (A - BK_\pi)\mu_{\text{new}} + (Bb_\pi + \overline{A}\mu + d), \tag{E.1}$$

where $K_\pi$ and $b_\pi$ are parameters of the policy $\pi$. By solving (E.1) for $\mu_{\text{new}}$, it holds that

$$\Lambda_2(\mu, \pi) = \mu_{\text{new}} = (I - A + BK_\pi)^{-1}(Bb_\pi + \overline{A}\mu + d).$$

This gives the form of $\Lambda_2(\cdot, \cdot)$.

Next, we compute the Lipschitz constants for $\Lambda_1(\cdot)$ and $\Lambda_2(\cdot, \cdot)$.

**Lipschitz constant for $\Lambda_1(\cdot)$.** By Proposition 3.4, for any $\mu_1, \mu_2 \in \mathbb{R}^m$, the optimal $K^*$ is fixed for Problem 2.2. Therefore, by the form of the optimal $b^K$ given in Proposition 3.4, it holds that

$$\begin{aligned}
\big\|\Lambda_1(\mu_1) - \Lambda_1(\mu_2)\big\|_2 &\leq c_2 \cdot \Big\|\big[(I - A)Q^{-1}(I - A)^\top + BR^{-1}B^\top\big]^{-1}\overline{A}\Big\|_2 \\
&\qquad \cdot \Big\|\big[K^*Q^{-1}(I - A)^\top - R^{-1}B^\top\big]\Big\|_2 \cdot \|\mu_1 - \mu_2\|_2 \\
&= c_2 L_1 \cdot \|\mu_1 - \mu_2\|_2, \tag{E.2}
\end{aligned}$$

where $L_1$ is defined in Assumption 3.1.

**Lipschitz constants for $\Lambda_2(\cdot, \cdot)$.** By Proposition 3.4, for any $\mu_1, \mu_2 \in \mathbb{R}^m$, the optimal $K^*$ is fixed for Problem 2.2. Thus, for any $\pi \in \Pi$ such that $\pi$ is an optimal policy under some $\mu \in \mathbb{R}^m$, it holds that

$$\begin{aligned}
\big\|\Lambda_2(\mu_1, \pi) - \Lambda_2(\mu_2, \pi)\big\|_2 &= \big\|(I - A + BK_\pi)^{-1} \cdot \overline{A} \cdot (\mu_1 - \mu_2)\big\|_2 \\
&\leq \big[1 - \rho(A - BK^*)\big]^{-1} \cdot \|\overline{A}\|_2 \cdot \|\mu_1 - \mu_2\|_2 \\
&= L_2 \cdot \|\mu_1 - \mu_2\|_2, \tag{E.3}
\end{aligned}$$

where $L_2$ is defined in Assumption 3.1, and $K_\pi = K^*$ is the parameter of the policy $\pi$. Meanwhile, for any mean-field state $\mu \in \mathbb{R}^m$, and any poicies $\pi_1, \pi_2 \in \Pi$ that are optimal under some mean-field states $\mu_1, \mu_2$, respectively, we have

$$\begin{aligned}
\big\|\Lambda_2(\mu, \pi_1) - \Lambda_2(\mu, \pi_2)\big\|_2 &= \big\|(I - A + BK^*)^{-1}B \cdot (b_{\pi_1} - b_{\pi_2})\big\|_2 \\
&\leq \big[1 - \rho(A - BK^*)\big]^{-1} \cdot \|B\|_2 \cdot \|b_{\pi_1} - b_{\pi_2}\|_2 \\
&= c_2^{-1} L_3 \cdot \|\pi_1 - \pi_2\|_2, \tag{E.4}
\end{aligned}$$

where in the last equality, we use the fact that $K_{\pi_1} = K_{\pi_2} = K^*$ by Proposition 3.4. Here $L_3$ is defined in Assumption 3.1, and $b_{\pi_1}$ and $b_{\pi_2}$ are the parameters of the policies $\pi_1$ and $\pi_2$.

Now we show that the operator $\Lambda(\cdot)$ is a contraction. For any $\mu_1, \mu_2 \in \mathbb{R}^m$, it holds that

$$\begin{aligned}
\big\|\Lambda(\mu_1) - \Lambda(\mu_2)\big\|_2 &= \Big\|\Lambda_2\big(\mu_1, \Lambda_1(\mu_1)\big) - \Lambda_2\big(\mu_2, \Lambda_1(\mu_2)\big)\Big\|_2 \\
&\leq \Big\|\Lambda_2\big(\mu_1, \Lambda_1(\mu_1)\big) - \Lambda_2\big(\mu_1, \Lambda_1(\mu_2)\big)\Big\|_2 + \Big\|\Lambda_2\big(\mu_1, \Lambda_1(\mu_2)\big) - \Lambda_2\big(\mu_2, \Lambda_1(\mu_2)\big)\Big\|_2 \\
&\leq c_2^{-1} L_3 \cdot \big\|\Lambda_1(\mu_1) - \Lambda_1(\mu_2)\big\|_2 + L_2 \cdot \|\mu_1 - \mu_2\|_2 \\
&\leq c_2^{-1} L_3 \cdot c_2 L_1 \cdot \|\mu_1 - \mu_2\|_2 + L_2 \cdot \|\mu_1 - \mu_2\|_2 = (L_1 L_3 + L_2) \cdot \|\mu_1 - \mu_2\|_2,
\end{aligned}$$

where the first inequality comes from triangle inequality, the second inequality comes from (E.3) and (E.4), and the last inequality comes from (E.2). By Assumption 3.1, we know that $L_0 = L_1 L_3 + L_2 < 1$, which shows that the operator $\Lambda(\cdot)$ is a contraction. Moreover, by Banach fixed-point theorem, we obtain that $\Lambda(\cdot)$ has a unique fixed point, which gives the unique equilibrium pair of Problem 2.3. We finish the proof of the proposition. $\qquad\square$

### E.2 PROOF OF PROPOSITION 3.4

*Proof.* By the definition of $J_2(K, b)$ in (3.6) and the definition of $\mu_{K,b}$ in (3.2), the problem

$$\min_b J_2(K, b)$$

is equivalent to the following constrained problem,

$$\min_{\mu,b} \quad \begin{pmatrix} \mu \\ b \end{pmatrix}^\top \begin{pmatrix} Q + K^\top R K & -K^\top R \\ -RK & R \end{pmatrix} \begin{pmatrix} \mu \\ b \end{pmatrix}$$

$$\text{s.t.} \quad (I - A + BK)\mu - (Bb + \overline{A}\mu + d) = 0. \tag{E.5}$$

Following from the KKT conditions of (E.5), it holds that

$$2M_K \begin{pmatrix} \mu \\ b \end{pmatrix} + N_K \lambda = 0, \qquad N_K^\top \begin{pmatrix} \mu \\ b \end{pmatrix} + \overline{A}\mu + d = 0, \tag{E.6}$$

where

$$M_K = \begin{pmatrix} Q + K^\top R K & -K^\top R \\ -RK & R \end{pmatrix}, \qquad N_K = \begin{pmatrix} -(I - A + BK)^\top \\ B^\top \end{pmatrix}.$$

By solving (E.6), the minimizer of (E.5) takes the form of

$$\begin{pmatrix} \mu_{K,b^K} \\ b^K \end{pmatrix} = -M_K^{-1} N_K (N_K^\top M_K^{-1} N_K)^{-1} (\overline{A}\mu + d). \tag{E.7}$$

By substituting (E.7) into the definition of $J_2(K, b)$ in (3.6), we have

$$J_2(K, b^K) = (\overline{A}\mu + d)^\top (N_K^\top M_K^{-1} N_K)^{-1} (\overline{A}\mu + d). \tag{E.8}$$

Meanwhile, by calculation, we have

$$M_K^{-1} = \begin{pmatrix} Q^{-1} & Q^{-1}K^\top \\ KQ^{-1} & KQ^{-1}K^\top + R^{-1} \end{pmatrix}.$$

Therefore, the term $N_K^\top M_K^{-1} N_K$ in (E.8) takes the form of

$$N_K^\top M_K^{-1} N_K = (I - A)Q^{-1}(I - A^\top) + BR^{-1}B^\top. \tag{E.9}$$

By plugging (E.9) into (E.8), we have

$$J_2(K, b^K) = (\overline{A}\mu + d)^\top \big[(I - A)Q^{-1}(I - A^\top) + BR^{-1}B^\top\big]^{-1} (\overline{A}\mu + d).$$

Also, by plugging (E.9) into (E.7), we have

$$\begin{pmatrix} \mu_{K,b^K} \\ b^K \end{pmatrix} = \begin{pmatrix} Q^{-1}(I - A)^\top \\ KQ^{-1}(I - A)^\top - R^{-1}B^\top \end{pmatrix} \big[(I - A)Q^{-1}(I - A)^\top + BR^{-1}B^\top\big]^{-1} (\overline{A}\mu + d).$$

We finish the proof of the proposition. $\qquad \square$

### E.3 PROOF OF PROPOSITION B.2

*Proof.* By the definition of the cost function $c(x, u)$ in Problem 2.2 (recall that we drop the subscript $\mu$ when we focus on Problem 2.2), we have

$$\mathbb{E}c_t = \mathbb{E}(x_t^\top Q x_t + u_t^\top R u_t + \mu^\top \overline{Q}\mu)$$

$$= \mathbb{E}(x_t^\top Q x_t + x_t^\top K^\top R K x_t - 2b^\top R K x_t + b^\top R b + \sigma^2 \eta_t^\top R \eta_t + \mu^\top \overline{Q}\mu)$$

$$= \mathbb{E}\big[x_t^\top (Q + K^\top R K)x_t - 2b^\top R K x_t\big] + b^\top R b + \sigma^2 \cdot \text{tr}(R) + \mu^\top \overline{Q}\mu, \tag{E.10}$$

where we write $c_t = c(x_t, u_t)$ for notational convenience. Here in the second line we use $u_t = \pi_{K,b}(x_t) = -Kx_t + b + \sigma\eta_t$. Therefore, combining (E.10) and the definition of $J(K, b)$ in Problem 2.2, we have

$$J(K, b) = \lim_{T \to \infty} \frac{1}{T} \sum_{t=0}^{T} \Big\{ \mathbb{E}\big[x_t^\top (Q + K^\top R K)x_t - 2b^\top R K x_t\big] + b^\top R b + \sigma^2 \cdot \text{tr}(R) + \mu^\top \overline{Q}\mu \Big\}$$

$$= \mathbb{E}_{x \sim \mathcal{N}(\mu_{K,b}, \Phi_K)} \big[x^\top (Q + K^\top R K)x - 2b^\top R K x\big] + b^\top R b + \sigma^2 \cdot \text{tr}(R) + \mu^\top \overline{Q}\mu$$

$$= \text{tr}\big[(Q + K^\top R K)\Phi_K\big] + \mu_{K,b}^\top (Q + K^\top R K)\mu_{K,b} - 2b^\top R K \mu_{K,b} \tag{E.11}$$

$$\qquad + b^\top R b + \sigma^2 \cdot \text{tr}(R) + \mu^\top \overline{Q}\mu.$$

Now, by iteratively applying (3.3) and (3.4), we have
$$\operatorname{tr}\big[(Q + K^\top RK)\Phi_K\big] = \operatorname{tr}(P_K\Psi_\epsilon), \tag{E.12}$$
where $P_K$ is given in (3.4). Combining (E.11) and (E.12), we know that
$$J(K, b) = J_1(K) + J_2(K, b) + \sigma^2 \cdot \operatorname{tr}(R) + \mu^\top \overline{Q}\mu,$$
where
$$J_1(K) = \operatorname{tr}\big[(Q + K^\top RK)\Phi_K\big] = \operatorname{tr}(P_K\Psi_\epsilon),$$
$$J_2(K, b) = \begin{pmatrix} \mu_{K,b} \\ b \end{pmatrix}^\top \begin{pmatrix} Q + K^\top RK & -K^\top R \\ -RK & R \end{pmatrix} \begin{pmatrix} \mu_{K,b} \\ b \end{pmatrix}.$$
We finish the proof of the proposition. □

### E.4 PROOF OF PROPOSITION 3.3

*Proof.* By calculating the Hessian matrix of $J_2(K, b)$, we have
$$\begin{aligned}
\nabla_{bb}^2 J_2(K, b) =& B^\top (I - A + BK)^{-\top}(Q + K^\top RK)(I - A + BK)^{-1}B \\
& - \big[RK(I - A + BK)^{-1}B + B^\top (I - A + BK)^{-\top}K^\top R\big] + R \\
=& \big[R^{1/2}K(I - A + BK)^{-1}B - R^{1/2}\big]^\top \big[R^{1/2}K(I - A + BK)^{-1}B - R^{1/2}\big] \\
& + B^\top (I - A + BK)^{-\top}Q(I - A + BK)^{-1}B,
\end{aligned}$$
which is a positive definite matrix independent of $b$. We denote by its minimum singular value as $\nu_K$. Also, note that $\|\nabla_{bb}^2 J_2(K, b)\|_2$ is upper bounded as
$$\big\|\nabla_{bb}^2 J_2(K, b)\big\|_2 \le \big[1 - \rho(A - BK)\big]^{-2} \cdot \big(\|B\|_2^2 \cdot \|K\|_2^2 \cdot \|R\|_2 + \|B\|_2^2 \cdot \|Q\|_2\big).$$
Therefore, it holds that
$$\iota_K \le \big[1 - \rho(A - BK)\big]^{-2} \cdot \big(\|B\|_2^2 \cdot \|K\|_2^2 \cdot \|R\|_2 + \|B\|_2^2 \cdot \|Q\|_2\big),$$
where $\iota_K$ is the maximum singular value of $\nabla_{bb}^2 J_2(K, b)$. We finish the proof of the proposition. □

### E.5 PROOF OF PROPOSITION B.3

*Proof.* Following from Proposition B.2, it holds that
$$J_1(K) = \operatorname{tr}(P_K\Psi_\epsilon) = \mathbb{E}_{y \sim \mathcal{N}(0, \Psi_\epsilon)}(y^\top P_K y) = \mathbb{E}_{y \sim \mathcal{N}(0, \Psi_\epsilon)}\big[f_K(y)\big], \tag{E.13}$$
where $f_K(y) = y^\top P_K y$. By the definition of $P_K$ in (3.4), we obtain that
$$\begin{aligned}
\nabla_K f_K(y) &= \nabla_K\Big\{y^\top(Q + K^\top RK)y + \big[(A - BK)y\big]^\top P_K\big[(A - BK)y\big]^\top\Big\} \\
&= 2RKyy^\top + \nabla_K\Big[f_K\big((A - BK)y\big)\Big]. \tag{E.14}
\end{aligned}$$
Also, we have
$$\nabla_K\Big[f_K\big((A - BK)y\big)\Big] = \nabla_K f_K\big((A - BK)y\big) - 2B^\top P_K(A - BK)yy^\top. \tag{E.15}$$
By plugging (E.15) into (E.14), we have
$$\nabla_K f_K(y) = 2\big[(R + B^\top P_K B)K - B^\top P_K A\big]yy^\top + \nabla_K f_K\big((A - BK)y\big). \tag{E.16}$$
By iteratively applying (E.16), it holds that
$$\nabla_K f_K(y) = 2\big[(R + B^\top P_K B)K - B^\top P_K A\big] \cdot \sum_{t=0}^{\infty} y_t y_t^\top, \tag{E.17}$$
where $y_{t+1} = (A - BK)y_t$ with $y_0 = y$. Now, combining (E.13) and (E.17), it holds that
$$\nabla_K J_1(K) = 2\big[(R + B^\top P_K B)K - B^\top P_K A\big]\Phi_K = 2(\Upsilon_K^{22}K - \Upsilon_K^{21}) \cdot \Phi_K,$$
where $\Upsilon_K$ is defined in (3.7). Meanwhile, combining the form of $\mu_{K,b}$ in (3.2), it holds by calculation that
$$\nabla_b J_2(K, b) = 2\big[\Upsilon_K^{22}(-K\mu_{K,b} + b) + \Upsilon_K^{21}\mu_{K,b} + q_{K,b}\big],$$
where $q_{K,b}$ is defined in (3.7). We finish the proof of the proposition. □

### E.6 Proof of Proposition B.1

*Proof.* From the definition of $V_{K,b}(x)$ in (B.1) and the definition of the cost function $c(x,u)$ in Problem 2.2, it holds that

$$V_{K,b}(x) = \sum_{t=0}^{\infty} \Big\{ \mathbb{E}\big[x_t^\top (Q + K^\top R K) x_t - 2b^\top R K x_t$$
$$+ b^\top R b + \sigma^2 \eta_t^\top R \eta_t + \mu^\top \overline{Q} \mu \,|\, x_0 = x\big] - J(K,b) \Big\}.$$

Combining (3.1), we know that $V_{K,b}(x)$ is a quadratic function taking the form of $V_{K,b}(x) = x^\top G x + r^\top x + h$, where $G$, $r$, and $h$ are functions of $K$ and $b$. Note that $V_{K,b}(x)$ satisfies that

$$V_{K,b}(x) = c(x, -Kx + b) - J(K,b) + \mathbb{E}\big[V_{K,b}(x') \,|\, x\big], \tag{E.18}$$

by substituting the form of $c(x, -Kx + b)$ in Problem 2.2 and $J(K,b)$ in (3.5) into (E.18), we obtain that

$$x^\top G x + r^\top x + h$$
$$= x^\top (Q + K^\top R K) x - 2b^\top R K x + b^\top R b + \mu^\top \overline{Q} \mu \tag{E.19}$$
$$- \big[\mathrm{tr}(P_K \Psi_\epsilon) + \mu_{K,b}^\top (Q + K^\top R K)\mu_{K,b} - 2b^\top R K \mu_{K,b} + \mu^\top \overline{Q}\mu + b^\top R b\big]$$
$$+ \big[(A - BK)x + (Bb + \overline{A}\mu + d)\big]^\top G\big[(A - BK)x + (Bb + \overline{A}\mu + d)\big]$$
$$+ \mathrm{tr}(G\Psi_\epsilon) + r^\top \big[(A - BK)x + (Bb + \overline{A}\mu + d)\big] + h - \sigma^2 \cdot \mathrm{tr}(R).$$

By comparing the quadratic terms and linear terms on both the LHS and RHS in (E.19), we obtain that

$$G = P_K, \qquad r = 2f_{K,b},$$

where $f_{K,b} = (I - A + BK)^{-\top}[(A - BK)^\top P_K (Bb + \overline{A}\mu + d) - K^\top R b]$. Also, by the definition of $V_{K,b}(x)$ in (B.1), we know that $\mathbb{E}[V_{K,b}(x)] = 0$, where the expectation is taken following the stationary distribution generated by the policy $\pi_{K,b}$ and the state transition. Therefore, we have

$$h = -2f_{K,b}\mu_{K,b} - \mu_{K,b}^\top P_K \mu_{K,b} - \mathrm{tr}(P_K \Phi_K),$$

which shows that

$$V_{K,b}(x) = x^\top P_K x - \mathrm{tr}(P_K \Phi_K) + 2f_{K,b}^\top (x - \mu_{K,b}) - \mu_{K,b}^\top P_K \mu_{K,b}. \tag{E.20}$$

For the action-value function $Q_{K,b}(x,u)$, by plugging (E.20) into (B.2), we obtain that

$$Q_{K,b}(x,u) = \begin{pmatrix} x \\ u \end{pmatrix}^\top \Upsilon_K \begin{pmatrix} x \\ u \end{pmatrix} + 2\begin{pmatrix} p_{K,b} \\ q_{K,b} \end{pmatrix}^\top \begin{pmatrix} x \\ u \end{pmatrix} - \mathrm{tr}(P_K \Phi_K) - \sigma^2 \cdot \mathrm{tr}(R + P_K B B^\top)$$
$$- b^\top R b + 2b^\top R K \mu_{K,b} - \mu_{K,b}^\top (Q + K^\top R K + P_K)\mu_{K,b}$$
$$+ 2f_{K,b}^\top \big[(\overline{A}\mu + d) - \mu_{K,b}\big] + (\overline{A}\mu + d)^\top P_K (\overline{A}\mu + d).$$

We finish the proof of the proposition. $\square$

### E.7 Proof of Proposition B.5

*Proof.* By Proposition B.1, it holds that $Q_{K,b}$ takes the following linear form

$$Q_{K,b}(x,u) = \psi(x,u)^\top \alpha_{K,b} + \beta_{K,b}, \tag{E.21}$$

where $\beta_{K,b}$ is a scalar independent of $x$ and $u$. Note that $Q_{K,b}(x,u)$ satisfies that

$$Q_{K,b}(x,u) = c(x,u) - J(K,b) + \mathbb{E}_{\pi_{K,b}}\big[Q_{K,b}(x',u') \,|\, x,u\big], \tag{E.22}$$

where $(x', u')$ is the state-action pair after $(x,u)$ following the policy $\pi_{K,b}$ and the state transition. Combining (E.21) and (E.22), we obtain that

$$\psi(x,u)^\top \alpha_{K,b} = c(x,u) - J(K,b) + \mathbb{E}_{\pi_{K,b}}\big[\psi(x',u') \,|\, x,u\big]^\top \alpha_{K,b}. \tag{E.23}$$

By left multiplying $\psi(x, u)$ to both sides of (E.23), and taking the expectation, we have

$$\mathbb{E}_{\pi_{K,b}}\Big\{\psi(x, u)\big[\psi(x, u) - \psi(x', u')\big]^\top\Big\} \cdot \alpha_{K,b} + \mathbb{E}_{\pi_{K,b}}\big[\psi(x, u)\big] \cdot J(K, b) = \mathbb{E}_{\pi_{K,b}}\big[c(x, u)\psi(x, u)\big].$$

Combining the definition of the matrix $\Theta_{K,b}$ in (B.7), we have

$$\begin{pmatrix} 1 & 0 \\ \mathbb{E}_{\pi_{K,b}}[\psi(x, u)] & \Theta_{K,b} \end{pmatrix} \begin{pmatrix} J(K, b) \\ \alpha_{K,b} \end{pmatrix} = \begin{pmatrix} J(K, b) \\ \mathbb{E}_{\pi_{K,b}}[c(x, u)\psi(x, u)] \end{pmatrix},$$

which concludes the proof of the proposition. $\qquad\square$

### E.8 PROOF OF PROPOSITION B.6

*Proof.* **Invertibility and Upper Bound.** We denote by $z_t = (x_t^\top, u_t^\top)^\top$ for any $t \geq 0$. Then following from the state transition and the policy $\pi_{K,b}$, the transition of $\{z_t\}_{t\geq 0}$ takes the form of

$$z_{t+1} = L z_t + \nu + \delta_t, \tag{E.24}$$

where $L$, $\nu$ and $\delta$ are defined as

$$L = \begin{pmatrix} A & B \\ -KA & -KB \end{pmatrix}, \qquad \nu = \begin{pmatrix} \overline{A}\mu + d \\ -K(\overline{A}\mu + d) + b \end{pmatrix}, \qquad \delta_t = \begin{pmatrix} \omega_t \\ -K\omega_t + \sigma\eta_t \end{pmatrix}.$$

Note that $L$ also takes the form of

$$L = \begin{pmatrix} I \\ -K \end{pmatrix} \begin{pmatrix} A & B \end{pmatrix}.$$

Combining the fact that $\rho(UV) = \rho(VU)$ for any matrices $U$ and $V$, we know that $\rho(L) = \rho(A - BK) < 1$, which verifies the stability of (E.24). Following from the stability of (E.24), we know that the Markov chain generated by (E.24) admits a unique stationary distribution $\mathcal{N}(\mu_z, \Sigma_z)$, where $\mu_z$ and $\Sigma_z$ satisfy that

$$\mu_z = L\mu_z + \nu, \qquad \Sigma_z = L\Sigma_z L^\top + \Psi_\delta.$$

where

$$\Psi_\delta = \begin{pmatrix} \Psi_\omega & -\Psi_\omega K^\top \\ -K\Psi_\omega & K\Psi_\omega K^\top + \sigma^2 I \end{pmatrix}.$$

Also, we know that $\Sigma_z$ takes the form of

$$\Sigma_z = \mathrm{Cov}\left[\begin{pmatrix} x \\ u \end{pmatrix}\right] = \begin{pmatrix} \Phi_K & -\Phi_K K^\top \\ -K\Phi_K & K\Phi_K K^\top + \sigma^2 I \end{pmatrix} = \begin{pmatrix} 0 & 0 \\ 0 & \sigma^2 I \end{pmatrix} + \begin{pmatrix} I \\ -K \end{pmatrix} \Phi_K \begin{pmatrix} I \\ -K \end{pmatrix}^\top, \tag{E.25}$$

where $\Phi_K$ is defined in (3.3).

The following lemma establishes the form of $\Theta_{K,b}$.

**Lemma E.2.** The matrix $\Theta_{K,b}$ in (B.7) takes the form of

$$\Theta_{K,b} = \begin{pmatrix} 2(\Sigma_z \otimes_s \Sigma_z)(I - L \otimes_s L)^\top & 0 \\ 0 & \Sigma_z(I - L)^\top \end{pmatrix}.$$

*Proof.* See §F.9 for a detailed proof. $\qquad\square$

Note that since $\rho(L) < 1$, both $I - L \otimes_s L$ and $I - L$ are positive definite. Therefore, by Lemma E.2, the matrix $\Theta_{K,b}$ is invertible. This finishes the proof of the invertibility of $\Theta_{K,b}$. Moreover, by (E.25) and Lemma E.2, we upper bound the spectral norm of $\Theta_{K,b}$ as

$$\|\Theta_{K,b}\|_2 \leq 2\max\Big\{\|\Sigma_z\|_2^2 \cdot \big(1 + \|L\|_2^2\big), \ \|\Sigma_z\|_2 \cdot \big(1 + \|L\|_2\big)\Big\} \leq 4\big(1 + \|K\|_\mathrm{F}^2\big)^2 \cdot \|\Phi_K\|_2^2,$$

which proves the upper bound of $\|\Theta_{K,b}\|_2$.

**Minimum singular value.** To lower bound $\sigma_{\min}(\widetilde{\Theta}_{K,b})$, we only need to upper bound $\sigma_{\max}(\widetilde{\Theta}_{K,b}^{-1}) = \|\widetilde{\Theta}_{K,b}^{-1}\|_2$. We first calculate $\widetilde{\Theta}_{K,b}^{-1}$. Recall that the matrix $\widetilde{\Theta}_{K,b}$ in (B.8) takes the form of

$$\widetilde{\Theta}_{K,b} = \begin{pmatrix} 1 & 0 \\ \mathbb{E}_{\pi_{K,b}}[\psi(x,u)] & \Theta_{K,b} \end{pmatrix}.$$

By the definition of the feature vector $\psi(x,u)$ in (B.5), the vector $\widetilde{\sigma}_z = \mathbb{E}_{\pi_{K,b}}[\psi(x,u)]$ takes the form of

$$\widetilde{\sigma}_z = \mathbb{E}_{\pi_{K,b}}[\psi(x,u)] = \begin{pmatrix} \mathrm{svec}(\Sigma_z) \\ \mathbf{0}_{k+m} \end{pmatrix},$$

where $\mathbf{0}_{k+m}$ denotes the all-zero column vector with dimension $k+m$. Also, since $\Theta_{K,b}$ is invertible, the matrix $\widetilde{\Theta}_{K,b}$ is also invertible, whose inverse takes the form of

$$\widetilde{\Theta}_{K,b}^{-1} = \begin{pmatrix} 1 & 0 \\ -\Theta_{K,b}^{-1} \cdot \widetilde{\sigma}_z & \Theta_{K,b}^{-1} \end{pmatrix}.$$

The following lemma upper bounds the spectral norm of $\widetilde{\Theta}_{K,b}^{-1}$.

**Lemma E.3.** The spectral norm of the matrix $\widetilde{\Theta}_{K,b}^{-1}$ is upper bounded by a positive constant $\widetilde{\lambda}_K$, where $\widetilde{\lambda}_K$ only depends on $\|K\|_2$ and $\rho(A - BK)$.

*Proof.* See §F.10 for a detailed proof. $\qquad\square$

By Lemma E.3, we know that $\sigma_{\min}(\widetilde{\Theta}_{K,b})$ is lower bounded by a positive constant $\lambda_K = 1/\widetilde{\lambda}_K$, which only depends on $\|K\|_2$ and $\rho(A - BK)$. This concludes the proof of the proposition. $\quad\square$

# F PROOFS OF LEMMAS

## F.1 PROOF OF LEMMA D.1

*Proof.* Following from (3.4), it holds that

$$y^\top P_{K_2} y = \sum_{t \geq 0} y^\top \left[(A - BK_2)^t\right]^\top (Q + K_2^\top R K_2)(A - BK_2)^t y. \tag{F.1}$$

Meanwhile, by the state transition $y_{t+1} = (A - BK_2)y_t$, we know that

$$y_t = (A - BK_2)^t y_0 = (A - BK_2)^t y. \tag{F.2}$$

By plugging (F.2) into (F.1), it holds that

$$y^\top P_{K_2} y = \sum_{t \geq 0} y_t^\top (Q + K_2^\top R K_2) y_t = \sum_{t \geq 0} (y_t^\top Q y_t + y_t^\top K_2^\top R K_2 y_t). \tag{F.3}$$

Also, it holds that

$$y^\top P_{K_1} y = \sum_{t \geq 0} (y_{t+1}^\top P_{K_1} y_{t+1} - y_t^\top P_{K_1} y_t) \tag{F.4}$$

Combining (F.3) and (F.4), we have

$$y^\top P_{K_2} y - y^\top P_{K_1} y = \sum_{t \geq 0} (y_t^\top Q y_t + y_t^\top K_2^\top R K_2 y_t + y_{t+1}^\top P_{K_1} y_{t+1} - y_t^\top P_{K_1} y_t). \tag{F.5}$$

Also, by the state transition $y_{t+1} = (A - BK_2)y_t$, it holds for any $t \geq 0$ that

$$\begin{aligned}
y_t^\top Q y_t &+ y_t^\top K_2^\top R K_2 y_t + y_{t+1}^\top P_{K_1} y_{t+1} - y_t^\top P_{K_1} y_t \\
&= y_t^\top \left[Q + (K_2 - K_1 + K_1)^\top R (K_2 - K_1 + K_1)\right] y_t \\
&\quad + y_t^\top \left[A - BK_1 - B(K_2 - K_1)\right]^\top P_{K_1} \left[A - BK_1 - B(K_2 - K_1)\right] y_t - y_t^\top P_{K_1} y_t \\
&= 2 y_t^\top (K_2 - K_1)^\top \left[(R + B^\top P_{K_1} B)K_1 - B^\top P_{K_1} A\right] y_t \\
&\quad + y_t^\top (K_2 - K_1)^\top (R + B^\top P_{K_1} B)(K_2 - K_1) y_t \\
&= 2 y_t^\top (K_2 - K_1)^\top (\Upsilon_{K_1}^{22} K_1 - \Upsilon_{K_1}^{21}) y_t + y_t^\top (K_2 - K_1)^\top \Upsilon_{K_1}^{22} (K_2 - K_1) y_t, \tag{F.6}
\end{aligned}$$

where the matrix $\Upsilon_{K_1}$ is defined in (3.7). By plugging (F.6) into (F.5), we have

$$
\begin{aligned}
y^\top & P_{K_2} y - y^\top P_{K_1} y \\
&= \sum_{t \geq 0} 2 y_t^\top (K_2 - K_1)^\top (\Upsilon_{K_1}^{22} K_1 - \Upsilon_{K_1}^{21}) y_t + y_t^\top (K_2 - K_1)^\top \Upsilon_{K_1}^{22} (K_2 - K_1) y_t \\
&= \sum_{t \geq 0} D_{K_1, K_2}(y_t),
\end{aligned}
$$

where $D_{K_1, K_2}(y) = 2 y^\top (K_2 - K_1)(\Upsilon_{K_1}^{22} K_1 - \Upsilon_{K_1}^{21}) y + y^\top (K_2 - K_1)^\top \Upsilon_{K_1}^{22} (K_2 - K_1) y$. We finish the proof of the lemma. $\qquad\square$

## F.2 PROOF OF LEMMA D.2

*Proof.* We prove (D.16) and (D.17) separately in the sequel.

**Proof of** (D.16). From the definition of $J_1(K)$ in (3.6), we have

$$
\begin{aligned}
J_1(K) - J_1(K^*) = \operatorname{tr}(P_K \Psi_\epsilon - P_{K^*} \Psi_\epsilon) &= \mathbb{E}_{y \sim \mathcal{N}(0, \Psi_\epsilon)} (y^\top P_K y - y^\top P_{K^*} y) \\
&= -\mathbb{E}\left[ \sum_{t \geq 0} D_{K, K^*}(y_t) \right],
\end{aligned} \tag{F.7}
$$

where in the last equality, we apply Lemma D.1 and the expectation is taken following the transition $y_{t+1} = (A - BK^*) y_t$ with initial state $y_0 \sim \mathcal{N}(0, \Psi_\epsilon)$. Here we denote by $D_{K, K^*}(y)$ as

$$
D_{K, K^*}(y) = 2 y^\top (K^* - K)(\Upsilon_K^{22} K - \Upsilon_K^{21}) y + y^\top (K^* - K)^\top \Upsilon_K^{22} (K^* - K) y.
$$

Also, we write $D_{K, K^*}(y)$ as

$$
\begin{aligned}
D_{K, K^*}(y) &= 2 y^\top (K^* - K)(\Upsilon_K^{22} K - \Upsilon_K^{21}) y + y^\top (K^* - K)^\top \Upsilon_K^{22} (K^* - K) y \tag{F.8} \\
&= y^\top \left[ K^* - K + (\Upsilon_K^{22})^{-1} (\Upsilon_K^{22} K - \Upsilon_K^{21}) \right]^\top \Upsilon_K^{22} \left[ K^* - K + (\Upsilon_K^{22})^{-1} (\Upsilon_K^{22} K - \Upsilon_K^{21}) \right] y \\
&\quad - y^\top (\Upsilon_K^{22} K - \Upsilon_K^{21})^\top (\Upsilon_K^{22})^{-1} (\Upsilon_K^{22} K - \Upsilon_K^{21}) y.
\end{aligned}
$$

Note that the first term on the RHS of (F.8) is positive, due to the fact that it is a quadratic form of a positive definite matrix, we lower bound $D_{K, K^*}(y)$ as

$$
D_{K, K^*}(y) \geq -y^\top (\Upsilon_K^{22} K - \Upsilon_K^{21})^\top (\Upsilon_K^{22})^{-1} (\Upsilon_K^{22} K - \Upsilon_K^{21}) y. \tag{F.9}
$$

Combining (F.7) and (F.9), it holds that

$$
\begin{aligned}
J_1(K) - J_1(K^*) &\leq \left\| \mathbb{E}\left( \sum_{t \geq 0} y_t y_t^\top \right) \right\|_2 \cdot \operatorname{tr}\left[ (\Upsilon_K^{22} K - \Upsilon_K^{21})^\top (\Upsilon_K^{22})^{-1} (\Upsilon_K^{22} K - \Upsilon_K^{21}) \right] \\
&= \| \Phi_{K^*} \|_2 \cdot \operatorname{tr}\left[ (\Upsilon_K^{22} K - \Upsilon_K^{21})^\top (\Upsilon_K^{22})^{-1} (\Upsilon_K^{22} K - \Upsilon_K^{21}) \right] \\
&\leq \left\| (\Upsilon_K^{22})^{-1} \right\|_2 \cdot \| \Phi_{K^*} \|_2 \cdot \operatorname{tr}\left[ (\Upsilon_K^{22} K - \Upsilon_K^{21})^\top (\Upsilon_K^{22} K - \Upsilon_K^{21}) \right] \\
&\leq \sigma_{\min}^{-1}(R) \cdot \| \Phi_{K^*} \|_2 \cdot \operatorname{tr}\left[ (\Upsilon_K^{22} K - \Upsilon_K^{21})^\top (\Upsilon_K^{22} K - \Upsilon_K^{21}) \right],
\end{aligned}
$$

where the last line comes from the fact that $\Upsilon_K^{22} = R + B^\top P_K B \succeq R$. This complete the proof of (D.16).

**Proof of** (D.17). Note that for any $\widetilde{K}$, it holds by the optimality of $K^*$ that

$$
J_1(K) - J_1(K^*) \geq J_1(K) - J_1(\widetilde{K}) = -\mathbb{E}\left[ \sum_{t \geq 0} D_{K, \widetilde{K}}(y_t) \right], \tag{F.10}
$$

where the expectation is taken following the transition $y_{t+1} = (A - B\widetilde{K}) y_t$ with initial state $y_0 \sim \mathcal{N}(0, \Psi_\epsilon)$. By taking $\widetilde{K} = K - (\Upsilon_K^{22})^{-1} (\Upsilon_K^{22} K - \Upsilon_K^{21})$ and following from a similar calculation as in (F.8), the function $D_{K, \widetilde{K}}(y)$ takes the form of

$$
D_{K, \widetilde{K}}(y) = -y^\top (\Upsilon_K^{22} K - \Upsilon_K^{21})^\top (\Upsilon_K^{22})^{-1} (\Upsilon_K^{22} K - \Upsilon_K^{21}) y. \tag{F.11}
$$

Combining (F.10) and (F.11), it holds that

$$
\begin{aligned}
J(K) - J(K^*) &\geq \mathrm{tr}\big[\Phi_{\widetilde{K}}(\Upsilon_K^{22}K - \Upsilon_K^{21})^\top (\Upsilon_K^{22})^{-1}(\Upsilon_K^{22}K - \Upsilon_K^{21})\big] \\
&\geq \sigma_{\min}(\Psi_\epsilon) \cdot \|\Upsilon_K^{22}\|_2^{-1} \cdot \mathrm{tr}\big[(\Upsilon_K^{22}K - \Upsilon_K^{21})^\top (\Upsilon_K^{22}K - \Upsilon_K^{21})\big],
\end{aligned}
$$

where we use the fact that $\Phi_{\widetilde{K}} = (A - B\widetilde{K})\Phi_{\widetilde{K}}(A - B\widetilde{K})^\top + \Psi_\epsilon \succeq \Psi_\epsilon$ in the last line. This finishes the proof of (D.17). □

### F.3 PROOF OF LEMMA D.3

*Proof.* By Proposition B.2, we have

$$
\big|J_1(\widetilde{K}_{n+1}) - J_1(K_{n+1})\big| = \mathrm{tr}\big[(P_{\widetilde{K}_{n+1}} - P_{K_{n+1}})\Psi_\epsilon\big] \leq \|P_{\widetilde{K}_{n+1}} - P_{K_{n+1}}\|_2 \cdot \|\Psi_\epsilon\|_\mathrm{F}. \quad \text{(F.12)}
$$

The following lemma upper bounds the term $\|P_{\widetilde{K}_{n+1}} - P_{K_{n+1}}\|_2$.

**Lemma F.1.** Suppose that the parameters $K$ and $\widetilde{K}$ satisfy that

$$
\|\widetilde{K} - K\|_2 \cdot (\|A - BK\|_2 + 1) \cdot \|\Phi_K\|_2 \leq \sigma_{\min}(\Psi_\omega)/4 \cdot \|B\|_2^{-1}, \quad \text{(F.13)}
$$

then it holds that

$$
\begin{aligned}
\|P_{\widetilde{K}} - P_K\|_2 &\leq 6 \cdot \sigma_{\min}^{-1}(\Psi_\omega) \cdot \|\Phi_K\|_2 \cdot \|K\|_2 \cdot \|R\|_2 \cdot \|\widetilde{K} - K\|_2 \quad \text{(F.14)} \\
&\quad \cdot \big(\|B\|_2 \cdot \|K\|_2\big) \cdot \|A - BK\|_2 + \|B\|_2 \cdot \|K\|_2 + 1\big).
\end{aligned}
$$

*Proof.* See Lemma 5.7 in Yang et al. (2019) for a detailed proof. □

To use Lemma F.1, it suffices to verify that $\widetilde{K}_{n+1}$ and $K_{n+1}$ satisfy (F.13). Note that from the definitions of $K_{n+1}$ and $\widetilde{K}_{n+1}$ in (D.18) and (D.21), respectively, we have

$$
\begin{aligned}
&\|\widetilde{K}_{n+1} - K_{n+1}\|_2 \cdot \big(\|A - B\widetilde{K}_{n+1}\|_2 + 1\big) \cdot \|\Phi_{\widetilde{K}_{n+1}}\|_2 \\
&\quad \leq \gamma \cdot \|\widehat{\Upsilon}_{K_n} - \Upsilon_{K_n}\|_\mathrm{F} \cdot \big(1 + \|K_n\|_2\big) \cdot \big(\|A - B\widetilde{K}_{n+1}\|_2 + 1\big) \cdot \|\Phi_{\widetilde{K}_{n+1}}\|_2. \quad \text{(F.15)}
\end{aligned}
$$

Now, we upper bound the RHS of (F.15). For the term $\|A - B\widetilde{K}_{n+1}\|_2$, it holds by the definition of $\widetilde{K}_{n+1}$ in (D.21) that

$$
\begin{aligned}
\|A - B\widetilde{K}_{n+1}\|_2 &\leq \|A - BK_n\|_2 + \gamma \cdot \|B\|_2 \cdot \|\Upsilon_{K_n}^{22}K_n - \Upsilon_{K_n}^{21}\|_2 \\
&\leq \|A - BK_n\|_2 + \gamma \cdot \|B\|_2 \cdot \|\Upsilon_{K_n}\|_2 \cdot \big(1 + \|K_n\|_2\big). \quad \text{(F.16)}
\end{aligned}
$$

By the definition of $\Upsilon_{K_n}$ in (3.7), we upper bound $\|\Upsilon_{K_n}\|_2$ as

$$
\begin{aligned}
\|\Upsilon_{K_n}\|_2 &\leq \|Q\|_2 + \|R\|_2 + \big(\|A\|_\mathrm{F} + \|B\|_\mathrm{F}\big)^2 \cdot \|P_{K_n}\|_2 \\
&\leq \|Q\|_2 + \|R\|_2 + \big(\|A\|_\mathrm{F} + \|B\|_\mathrm{F}\big)^2 \cdot J_1(K_0) \cdot \sigma_{\min}^{-1}(\Psi_\epsilon), \quad \text{(F.17)}
\end{aligned}
$$

where the last line comes from the fact that

$$
J_1(K_0) \geq J_1(K_n) = \mathrm{tr}\big[(Q + K_n^\top R K_n)\Phi_{K_n}\big] = \mathrm{tr}(P_{K_n}\Psi_\epsilon) \geq \|P_{K_n}\|_2 \cdot \sigma_{\min}(\Psi_\epsilon).
$$

As for the term $\|\Phi_{\widetilde{K}_{n+1}}\|_2$ in (F.15), from the fact that

$$
J_1(K_0) \geq J_1(\widetilde{K}_{n+1}) = \mathrm{tr}\big[(Q + \widetilde{K}_{n+1}^\top R\widetilde{K}_{n+1})\Phi_{\widetilde{K}_{n+1}}\big] \geq \|\Phi_{\widetilde{K}_{n+1}}\|_2 \cdot \sigma_{\min}(Q),
$$

it holds that

$$
\|\Phi_{\widetilde{K}_{n+1}}\|_2 \leq J_1(K_0) \cdot \sigma_{\min}^{-1}(Q). \quad \text{(F.18)}
$$

Therefore, combining (F.15), (F.16), (F.17), and (F.18), we know that

$$
\begin{aligned}
&\|\widetilde{K}_{n+1} - K_{n+1}\|_2 \cdot \big(\|A - B\widetilde{K}_{n+1}\|_2 + 1\big) \cdot \|\Phi_{\widetilde{K}_{n+1}}\|_2 \\
&\quad \leq \mathrm{poly}_1\big(\|K_n\|_2\big) \cdot \|\widehat{\Upsilon}_{K_n} - \Upsilon_{K_n}\|_\mathrm{F}. \quad \text{(F.19)}
\end{aligned}
$$

From Theorem B.8, it holds with probability at least $1 - T_n^{-4} - \widetilde{T}_n^{-6}$ that

$$\|\widehat{\Upsilon}_{K_n} - \Upsilon_{K_n}\|_F \leq \frac{\text{poly}_3(\|K_n\|_F, \|\mu\|_2)}{\lambda_{K_n} \cdot (1 - \rho)^2} \cdot \frac{\log^3 T_n}{T_n^{1/4}} \tag{F.20}$$
$$+ \frac{\text{poly}_4(\|K_n\|_F, \|b_0\|_2, \|\mu\|_2)}{\lambda_{K_n}} \cdot \frac{\log^{1/2} \widetilde{T}_n}{\widetilde{T}_n^{1/8} \cdot (1 - \rho)},$$

which holds for any $\rho \in (\rho(A - BK_n), 1)$. Note that from the choice of $T_n$ and $\widetilde{T}_n$ in the statement of Theorem B.4 that

$$T_n \geq \text{poly}_5(\|K_n\|_F, \|b_0\|_2, \|\mu\|_2) \cdot \lambda_{K_n}^{-4} \cdot [1 - \rho(A - BK_n)]^{-9} \cdot \varepsilon^{-5},$$
$$\widetilde{T}_n \geq \text{poly}_6(\|K_n\|_F, \|b_0\|_2, \|\mu\|_2) \cdot \lambda_{K_n}^{-2} \cdot [1 - \rho(A - BK_n)]^{-12} \cdot \varepsilon^{-12},$$

it holds that

$$\frac{\text{poly}_3(\|K_n\|_F, \|\mu\|_2)}{\lambda_{K_n} \cdot (1 - \rho)^2} \cdot \frac{\log^3 T_n}{T_n^{1/4}} + \frac{\text{poly}_4(\|K_n\|_F, \|b_0\|_2, \|\mu\|_2)}{\lambda_{K_n}} \cdot \frac{\log^{1/2} \widetilde{T}_n}{\widetilde{T}_n^{1/8} \cdot (1 - \rho)}$$
$$\leq \min \left\{ \left[\text{poly}_1(\|K_n\|_2)\right]^{-1} \cdot \sigma_{\min}(\Psi_\omega)/4 \cdot \|B\|_2^{-1}, \tag{F.21} \right.$$
$$\left. \left[\text{poly}_2(\|K_n\|_2)\right]^{-1} \cdot \varepsilon/8 \cdot \gamma \cdot \sigma_{\min}(\Psi_\epsilon) \cdot \sigma_{\min}(R) \cdot \|\Phi_{K^*}\|_2^{-1} \cdot \|\Psi_\epsilon\|_F^{-1} \right\}.$$

Combining (F.19), (F.20), and (F.21), we know that (F.13) holds with probability at least $1 - \varepsilon^{15}$ for sufficiently small $\varepsilon > 0$. Meanwhile, by (F.16), (F.17), and (F.18), the RHS of (F.14) is upper bounded as

$$6 \cdot \sigma_{\min}^{-1}(\Psi_\omega) \cdot \|\Phi_{\widetilde{K}_{n+1}}\|_2 \cdot \|\widetilde{K}_{n+1}\|_2 \cdot \|R\|_2 \cdot \|\widetilde{K}_{n+1} - K_{n+1}\|_2$$
$$\cdot (\|B\|_2 \cdot \|\widetilde{K}_{n+1}\|_2) \cdot \|A - B\widetilde{K}_{n+1}\|_2 + \|B\|_2 \cdot \|\widetilde{K}_{n+1}\|_2 + 1)$$
$$\leq \text{poly}_2(\|K_n\|_2) \cdot \|\widehat{\Upsilon}_{K_n} - \Upsilon_{K_n}\|_F. \tag{F.22}$$

Now, by Lemma F.1, it holds with probability at least $1 - \varepsilon^{15}$ that

$$\|P_{\widetilde{K}_{n+1}} - P_{K_{n+1}}\|_2 \leq 6 \cdot \sigma_{\min}^{-1}(\Psi_\omega) \cdot \|\Phi_{\widetilde{K}_{n+1}}\|_2 \cdot \|\widetilde{K}_{n+1}\|_2 \cdot \|R\|_2 \cdot \|\widetilde{K}_{n+1} - K_{n+1}\|_2$$
$$\cdot (\|B\|_2 \cdot \|\widetilde{K}_{n+1}\|_2) \cdot \|A - B\widetilde{K}_{n+1}\|_2 + \|B\|_2 \cdot \|\widetilde{K}_{n+1}\|_2 + 1)$$
$$\leq \text{poly}_2(\|K_n\|_2) \cdot \|\widehat{\Upsilon}_{K_n} - \Upsilon_{K_n}\|_F$$
$$\leq \varepsilon/8 \cdot \gamma \cdot \sigma_{\min}(\Psi_\epsilon) \cdot \sigma_{\min}(R) \cdot \|\Phi_{K^*}\|_2^{-1} \cdot \|\Psi_\epsilon\|_F^{-1}, \tag{F.23}$$

where the second inequality comes from (F.22), and the last inequality comes from (F.20) and (F.21). Combining (F.12) and (F.23), it holds with probability at least $1 - \varepsilon^{15}$ that

$$\left| J_1(\widetilde{K}_{n+1}) - J_1(K_{n+1}) \right| \leq \gamma \cdot \sigma_{\min}(\Psi_\epsilon) \cdot \sigma_{\min}(R) \cdot \|\Phi_{K^*}\|_2^{-1} \cdot \varepsilon/4,$$

which concludes the proof of the lemma. $\square$

### F.4 PROOF OF LEMMA D.4

*Proof.* Note that $\Upsilon_{K^*}^{22} K^* - \Upsilon_{K^*}^{21}$ is the natural gradient of $J_1$ at the minimizer $K^*$, which implies that

$$\Upsilon_{K^*}^{22} K^* - \Upsilon_{K^*}^{21} = 0. \tag{F.24}$$

By Lemma D.1, it holds that

$$J_1(K) - J_1(K^*) = \text{tr}(P_K \Psi_\epsilon - P_{K^*} \Psi_\epsilon) = \mathbb{E}_{y \sim \mathcal{N}(0, \Psi_\epsilon)}(y^\top P_K y - y^\top P_{K^*} y)$$
$$= \mathbb{E} \left\{ \sum_{t \geq 0} \left[ 2y_t^\top (K - K^*)(\Upsilon_{K^*}^{22} K^* - \Upsilon_{K^*}^{21}) y_t + y_t^\top (K - K^*)^\top \Upsilon_{K^*}^{22}(K - K^*) y_t \right] \right\}$$
$$= \mathbb{E} \left\{ \sum_{t \geq 0} y_t^\top (K - K^*)^\top \Upsilon_{K^*}^{21}(K - K^*) y_t \right\}, \tag{F.25}$$

where we use (F.24) in the last line. Here the expectations are taken following the transition $y_{t+1} = (A - BK)y_t$ with initial state $y_0 \sim \mathcal{N}(0, \Psi_\epsilon)$. Also, we have

$$
\begin{aligned}
\mathbb{E}\bigg\{ \sum_{t \geq 0} & y_t^\top (K - K^*)^\top \Upsilon_{K^*}^{22}(K - K^*) y_t \bigg\} \\
&= \operatorname{tr}\big[\Phi_K (K - K^*)^\top \Upsilon_{K^*}^{22}(K - K^*)\big] \\
&\geq \|\Phi_K\|_2 \cdot \|\Upsilon_{K^*}^{22}\|_2 \cdot \operatorname{tr}\big[(K - K^*)^\top (K - K^*)\big] \\
&\geq \sigma_{\min}(\Psi_\epsilon) \cdot \sigma_{\min}(R) \cdot \|K - K^*\|_{\mathrm{F}}^2,
\end{aligned}
\tag{F.26}
$$

where we use the fact that $\Phi_K = (A - BK)\Phi_K(A - BK) + \Psi_\epsilon \succeq \Psi_\epsilon$ and $\Upsilon_{K^*}^{22} = R + B^\top P_{K^*} B \succeq R$ in the last line. Combining (F.25) and (F.26), we have

$$
J_1(K) - J_1(K^*) \geq \sigma_{\min}(\Psi_\epsilon) \cdot \sigma_{\min}(R) \cdot \|K - K^*\|_{\mathrm{F}}^2.
$$

We conclude the proof of the lemma. $\qquad\square$

### F.5 Proof of Lemma D.5

*Proof.* Following from Proposition 3.3, we have

$$
\begin{aligned}
J_2(K_N, & b_{h+1}) - J_2(K_N, \widetilde{b}_{h+1}) \\
&\leq \gamma^b \cdot \nabla_b J_2(K_N, \widetilde{b}_{h+1})^\top \big[\nabla_b J_2(K_N, b_h) - \widehat{\nabla}_b J_2(K_N, b_h)\big] \\
&\quad + (\gamma^b)^2 \cdot \nu_{K_N}/2 \cdot \big\|\nabla_b J_2(K_N, b_h) - \widehat{\nabla}_b J_2(K_N, b_h)\big\|_2^2, \\
J_2(K_N, & \widetilde{b}_{h+1}) - J_2(K_N, b_{h+1}) \\
&\leq -\gamma^b \cdot \nabla_b J_2(K_N, \widetilde{b}_{h+1})^\top \big[\nabla_b J_2(K_N, b_h) - \widehat{\nabla}_b J_2(K_N, b_h)\big] \\
&\quad - (\gamma^b)^2 \cdot \iota_{K_N}/2 \cdot \big\|\nabla_b J_2(K_N, b_h) - \widehat{\nabla}_b J_2(K_N, b_h)\big\|_2^2,
\end{aligned}
\tag{F.27}
$$

where $\nu_{K_N}$ and $\iota_{K_N}$ are defined in Proposition 3.3. Also, following from Proposition B.3, it holds that

$$
\big\|\nabla_b J_2(K_N, \widetilde{b}_{h+1})\big\|_2 \leq \operatorname{poly}_1\big(\|K_N\|_{\mathrm{F}}, \|b_h\|_2, \|\mu\|_2, J(K_N, b_0)\big) \cdot \big[1 - \rho(A - BK_N)\big]^{-1}.
\tag{F.28}
$$

Combining (F.27), (F.28), and the fact that $\nu_{K_N} \leq \iota_{K_N} \leq [1 - \rho(A - BK_N)]^{-2} \cdot \operatorname{poly}_2(\|K_N\|_2)$, we know that

$$
\big| J_2(K_N, b_{h+1}) - J_2(K_N, \widetilde{b}_{h+1}) \big|
\tag{F.29}
$$
$$
\begin{aligned}
&\leq (\gamma^b)^2 \cdot \operatorname{poly}_2\big(\|K_N\|_2\big) \cdot \big\|\nabla_b J_2(K_N, b_h) - \widehat{\nabla}_b J_2(K_N, b_h)\big\|_2^2 \cdot \big[1 - \rho(A - BK_N)\big]^{-2} \\
&\quad + \gamma^b \cdot \operatorname{poly}_1\big(\|K_N\|_{\mathrm{F}}, \|b_h\|_2, \|\mu\|_2, J(K_N, b_0)\big) \cdot \big\|\nabla_b J_2(K_N, b_h) - \widehat{\nabla}_b J_2(K_N, b_h)\big\|_2 \\
&\quad \cdot \big[1 - \rho(A - BK_N)\big]^{-1}.
\end{aligned}
$$

Note that from the definition of $\widehat{\nabla}_b J_2(K_N, b_h)$ and $\nabla_b J_2(K_N, b_h)$ in (D.31) and (D.33), respectively, it holds by triangle inequality that

$$
\begin{aligned}
\big\|\nabla_b & J_2(K_N, b_h) - \widehat{\nabla}_b J_2(K_N, b_h)\big\|_2 \\
&\leq \|\widehat{\Upsilon}_{K_N}^{22} - \Upsilon_{K_N}^{22}\|_2 \cdot \|K_N\|_2 \cdot \|\widehat{\mu}_{K_N, b_h}\|_2 + \|\Upsilon_{K_N}^{22}\|_2 \cdot \|K_N\|_2 \cdot \|\widehat{\mu}_{K_N, b_h} - \mu_{K_N, b_h}\|_2 \\
&\quad + \|\widehat{\Upsilon}_{K_N}^{22} - \Upsilon_{K_N}^{22}\|_2 \cdot \|b_h\|_2 + \|\widehat{\Upsilon}_{K_N}^{21} - \Upsilon_{K_N}^{21}\|_2 \cdot \|\widehat{\mu}_{K_N, b_h}\|_2 + \|\widehat{q}_{K_N, b_h} - q_{K_N, b_h}\|_2 \\
&\quad + \|\Upsilon_{K_N}^{21}\|_2 \cdot \|\widehat{\mu}_{K_N, b_h} - \mu_{K_N, b_h}\|_2.
\end{aligned}
$$

By Theorem B.8, combining the fact that $J_2(K_N, b_h) \leq J_2(K_N, b_0)$ and the fact that $\|\mu_{K_N, b}\|_2 \leq J(K_N, b_0)/\sigma_{\min}(Q)$, we know that with probability at least $1 - (T_n^b)^{-4} - (\widetilde{T}_n^b)^{-6}$, it holds for any $\rho \in (\rho(A - BK_N), 1)$ that

$$
\big\|\nabla_b J_2(K_N, b_h) - \widehat{\nabla}_b J_2(K_N, b_h)\big\|_2
\tag{F.30}
$$
$$
\leq \lambda_{K_N}^{-1} \cdot \operatorname{poly}_3\big(\|K_N\|_{\mathrm{F}}, \|b_h\|_2, \|\mu\|_2, J_2(K_N, b_0)\big) \cdot \left[ \frac{\log^3 T_n^b}{(T_n^b)^{1/4}(1 - \rho)^2} + \frac{\log^{1/2} \widetilde{T}_n^b}{(\widetilde{T}_n^b)^{1/8} \cdot (1 - \rho)} \right].
$$

Following from the choices of $\gamma^b$, $T_n^b$, and $\widetilde{T}_n^b$ in the statement of Theorem B.4, it holds that

$$\gamma^b \cdot \text{poly}_1\big(\|K_N\|_{\text{F}}, \|b_h\|_2, \|\mu\|_2, J(K_N, b_0)\big) \cdot \lambda_{K_N}^{-1} \cdot \text{poly}_3\big(\|K_N\|_{\text{F}}, \|b_h\|_2, \|\mu\|_2, J_2(K_N, b_0)\big)$$

$$\cdot \left[ \frac{\log^3 T_n^b}{(T_n^b)^{1/4}(1-\rho)^2} + \frac{\log^{1/2} \widetilde{T}_n^b}{(\widetilde{T}_n^b)^{1/8} \cdot (1-\rho)} \right] \cdot \big[1 - \rho(A - BK_N)\big]^{-1} + \big[1 - \rho(A - BK_N)\big]^{-2}$$

$$\cdot \text{poly}_3\big(\|K_N\|_{\text{F}}, \|b_h\|_2, \|\mu\|_2, J_2(K_N, b_0)\big) \cdot \left[ \frac{\log^6 T_n^b}{(T_n^b)^{1/2}(1-\rho)^4} + \frac{\log \widetilde{T}_n^b}{(\widetilde{T}_n^b)^{1/4} \cdot (1-\rho)^2} \right]$$

$$\cdot (\gamma^b)^2 \cdot \text{poly}_2\big(\|K_N\|_2\big) \cdot \lambda_{K_N}^{-1}$$

$$\leq \nu_{K_N} \cdot \gamma^b \cdot \varepsilon/2.$$

Further combining (F.29) and (F.30), it holds with probability at least $1 - \varepsilon^{15}$ that

$$\left| J_2(K_N, b_{h+1}) - J_2(K_N, \widetilde{b}_{h+1}) \right| \leq \nu_{K_N} \cdot \gamma^b \cdot \varepsilon/2.$$

We then finish the proof of the lemma. $\qquad\square$

## F.6 PROOF OF LEMMA D.6

*Proof.* We show that $\zeta_{K,b} \in \mathcal{V}_\zeta$ and $\xi(\zeta) \in \mathcal{V}_\xi$ for any $\zeta \in \mathcal{V}_\zeta$ separately.

**Part 1.** First we show that $\zeta_{K,b} \in \mathcal{V}_\zeta$. Note that from Definition B.7, we know that $\zeta_{K,b}^1 = J(K, b)$ satisfies that $0 \leq \zeta_{K,b}^1 \leq J(K_0, b_0)$. It remains to show that $\zeta_{K,b}^2 = \alpha_{K,b}$ satisfies that $\|\zeta_{K,b}^2\|_2 \leq M_\zeta$. By the definition of $\alpha_{K,b}$ in (B.6), we know that

$$\|\alpha_{K,b}\|_2^2 \leq \|\Upsilon_K\|_{\text{F}}^2 + \|\Upsilon_K\|_2^2 \cdot \big(\|\mu_{K,b}\|_2^2 + \|\mu_{K,b}^u\|_2^2\big)$$
$$+ \big(\|A\|_2 + \|B\|_2\big)^2 \cdot \big(\|P_K\|_2 \cdot \|\overline{A}\mu + d\|_2 + \|f_{K,b}\|_2\big)^2 \qquad \text{(F.31)}$$

where $f_{K,b} = (I - A + BK)^{-\top}[(A - BK)^\top P_K(Bb + \overline{A}\mu + d) - K^\top Rb]$ and for notational simplicity, we denote by $\mu_{K,b}^u = -K\mu_{K,b} + b$. We only need to bound $\Upsilon_K$, $\mu_{K,b}$, $\mu_{K,b}^u$, $P_K$, and $f_{K,b}$. Note that by Proposition B.2, the expected total cost $J(K, b)$ takes the form of

$$J(K, b) = \text{tr}(P_K \Psi_\epsilon) + \mu_{K,b}^\top Q \mu_{K,b} + (\mu_{K,b}^u)^\top R \mu_{K,b}^u + \sigma^2 \cdot \text{tr}(R) + \mu^\top \overline{Q}\mu.$$

Thus, we have

$$J(K_0, b_0) \geq J(K, b) \geq \sigma_{\min}(\Psi_\omega) \cdot \text{tr}(P_K) \geq \sigma_{\min}(\Psi_\omega) \cdot \|P_K\|_2,$$
$$J(K_0, b_0) \geq J(K, b) \geq \mu_{K,b}^\top Q \mu_{K,b} \geq \sigma_{\min}(Q) \cdot \|\mu_{K,b}\|_2,$$
$$J(K_0, b_0) \geq J(K, b) \geq (\mu_{K,b}^u)^\top R \mu_{K,b}^u \geq \sigma_{\min}(R) \cdot \|\mu_{K,b}^u\|_2,$$

which imply that

$$\|P_K\|_2 \leq J(K_0, b_0)/\sigma_{\min}(\Psi_\omega),$$
$$\|\mu_{K,b}\|_2 \leq J(K_0, b_0)/\sigma_{\min}(Q),$$
$$\|\mu_{K,b}^u\|_2 \leq J(K_0, b_0)/\sigma_{\min}(R). \qquad \text{(F.32)}$$

For $\Upsilon_K$, it holds that

$$\Upsilon_K = \begin{pmatrix} Q & \mathbf{0} \\ \mathbf{0} & R \end{pmatrix} + \begin{pmatrix} A^\top \\ B^\top \end{pmatrix} P_K \begin{pmatrix} A & B \end{pmatrix},$$

which gives

$$\|\Upsilon_K\|_{\text{F}} \leq \big(\|Q\|_{\text{F}} + \|R\|_{\text{F}}\big) + \big(\|A\|_{\text{F}}^2 + \|B\|_{\text{F}}^2\big) \cdot \|P_K\|_{\text{F}},$$
$$\|\Upsilon_K\|_2 \leq \big(\|Q\|_2 + \|R\|_2\big) + \big(\|A\|_2 + \|B\|_2\big)^2 \cdot \|P_K\|_2.$$

Combining (F.32) and the fact that $\|P_K\|_{\text{F}} \leq \sqrt{m} \cdot \|P_K\|_2$, we know that

$$\|\Upsilon_K\|_{\text{F}} \leq \big(\|Q\|_{\text{F}} + \|R\|_{\text{F}}\big) + \big(\|A\|_{\text{F}}^2 + \|B\|_{\text{F}}^2\big) \cdot \sqrt{m} \cdot J(K_0, b_0)/\sigma_{\min}(\Psi_\omega),$$
$$\|\Upsilon_K\|_2 \leq \big(\|Q\|_2 + \|R\|_2\big) + \big(\|A\|_2 + \|B\|_2\big)^2 \cdot J(K_0, b_0)/\sigma_{\min}(\Psi_\omega). \qquad \text{(F.33)}$$

Now, we upper bound the vector $f_{K,b}$. Note that by algebra, the vector $f_{K,b}$ takes the form of

$$f_{K,b} = -P_K \mu_{K,b} + (I - A + BK)^{-T}(Q\mu_{K,b} - K^\top R\mu_{K,b}^u).$$

Therefore, we upper bound $f_{K,b}$ as

$$\|f_{K,b}\|_2 \le J(K_0, b_0)^2 \cdot \sigma_{\min}^{-1}(\Psi_\omega) \cdot \sigma_{\min}^{-1}(Q) + \left[1 - \rho(A - BK)\right]^{-1} \cdot (\kappa_Q + \kappa_R \cdot \|K\|_F) \quad \text{(F.34)}$$

Combining (F.31), (F.32), (F.33), and (F.34), it holds that

$$\|\zeta_{K,b}^2\|_2 = \|\alpha_{K,b}\|_2 \le M_{\zeta,1} + M_{\zeta,2} \cdot (1 + \|K\|_F) \cdot [1 - \rho(A - BK)]^{-1}.$$

Therefore, it holds that $\zeta_{K,b} \in \mathcal{V}_\zeta$.

**Part 2.** Now we show that for any $\zeta \in \mathcal{V}_\zeta$, we have $\xi(\zeta) \in \mathcal{V}_\xi$. Recall that from (D.41), it holds that

$$\xi^1(\zeta) = \zeta^1 - J(K, b), \quad \xi^2(\zeta) = \mathbb{E}_{\pi_{K,b}}\big[\psi(x, u)\big]\zeta^1 + \Theta_{K,b}\zeta^2 - \mathbb{E}_{\pi_{K,b}}\big[c(x, u)\psi(x, u)\big]. \quad \text{(F.35)}$$

Then we have

$$\big|\xi^1(\zeta)\big| = \big|\zeta^1 - J(K, b)\big| \le J(K_0, b_0), \quad \text{(F.36)}$$

where we use the fact that since $\zeta \in \mathcal{V}_\zeta$, we have $0 \le \zeta^1 \le J(K_0, b_0)$ by Definition B.7. Also, by (F.35), we have

$$\big\|\xi^2(\zeta)\big\|_2 \le \underbrace{\Big\|\mathbb{E}_{\pi_{K,b}}\big[\psi(x, u)\big]\zeta^1\Big\|_2}_{B_1} + \underbrace{\|\Theta_{K,b}\|_2 \cdot \|\zeta^2\|_2}_{B_2} + \underbrace{\Big\|\mathbb{E}_{\pi_{K,b}}\big[c(x, u)\psi(x, u)\big]\Big\|_2}_{B_3}. \quad \text{(F.37)}$$

Note that we upper bound $B_1$ as

$$B_1 \le J(K_0, b_0) \cdot \Big\|\mathbb{E}_{\pi_{K,b}}\big[\psi(x, u)\big]\Big\|_2. \quad \text{(F.38)}$$

Following from the definition of $\psi(x, u)$ in (B.5), we know that

$$\Big\|\mathbb{E}_{\pi_{K,b}}\big[\psi(x, u)\big]\Big\|_2 \le \|\Sigma_z\|_F, \quad \text{(F.39)}$$

where $\Sigma_z$ is defined as

$$\Sigma_z = \text{Cov}\left[\begin{pmatrix} x \\ u \end{pmatrix}\right] = \begin{pmatrix} \Phi_K & -\Phi_K K^\top \\ -K\Phi_K & K\Phi_K K^\top + \sigma^2 I \end{pmatrix} = \begin{pmatrix} 0 & 0 \\ 0 & \sigma^2 I \end{pmatrix} + \begin{pmatrix} I \\ -K \end{pmatrix} \Phi_K \begin{pmatrix} I \\ -K \end{pmatrix}^\top.$$

Combining (F.38) and (F.39), we have

$$B_1 \le J(K_0, b_0) \cdot \|\Sigma_z\|_F. \quad \text{(F.40)}$$

By Proposition B.6, we upper bound $B_2$ as

$$B_2 \le 4(1 + \|K\|_F^2)^3 \cdot \|\Phi_K\|_2^2 \cdot (M_{\zeta,1} + M_{\zeta,2}) \cdot \big[1 - \rho(A - BK)\big]^{-1}, \quad \text{(F.41)}$$

where we use the fact that $\zeta \in \mathcal{V}_\zeta$ and Definition B.7. As for the term $B_3$ in (F.37), we utilize the following lemma to provide an upper bound.

**Lemma F.2.** The vector $\mathbb{E}_{\pi_{K,b}}[c(x, u)\psi(x, u)]$ takes the following form,

$$\mathbb{E}_{\pi_{K,b}}\big[c(x, u)\psi(x, u)\big] = \begin{pmatrix} 2\text{svec}\big[\Sigma_z \text{diag}(Q, R)\Sigma_z + \langle\Sigma_z, \text{diag}(Q, R)\rangle\Sigma_z\big] \\ \Sigma_z \begin{pmatrix} 2Q\mu_{K,b} \\ 2R\mu_{K,b}^u \end{pmatrix} \end{pmatrix}$$
$$+ \big[\mu_{K,b}^\top Q\mu_{K,b} + (\mu_{K,b}^u)^\top R\mu_{K,b}^u + \mu^\top \overline{Q}\mu\big] \begin{pmatrix} \text{svec}(\Sigma_z) \\ \mathbf{0}_m \\ \mathbf{0}_k \end{pmatrix}.$$

Here the matrix $\Sigma_z$ takes the form of

$$\Sigma_z = \begin{pmatrix} \Phi_K & -\Phi_K K^\top \\ -K\Phi_K & K\Phi_K K^\top + \sigma^2 \cdot I \end{pmatrix}.$$

*Proof.* See §F.11 for a detailed proof. $\qquad \square$

From Lemma F.2 and (F.32), it holds that

$$B_3 \le 3\big[\|Q\|_{\mathrm{F}} + \|R\|_{\mathrm{F}} + J(K_0, b_0) \cdot \|Q\|_2/\sigma_{\min}(Q) \tag{F.42}$$
$$+ J(K_0, b_0) \cdot \|R\|_2/\sigma_{\min}(R)\big] \cdot \|\Sigma_z\|_2^2.$$

Moreover, by the definition of $\Sigma_z$ in (E.25), combining the triangle inequality, we have the following bounds for $\|\Sigma_z\|_{\mathrm{F}}$ and $\|\Sigma_z\|_2$,

$$\|\Sigma_z\|_{\mathrm{F}} \le 2(d + \|K\|_{\mathrm{F}}^2) \cdot \|\Phi_K\|_2, \qquad \|\Sigma_z\|_2 \le 2(1 + \|K\|_{\mathrm{F}}^2) \cdot \|\Phi_K\|_2. \tag{F.43}$$

Also, we have

$$J(K_0, b_0) \ge J(K, b) \ge \mathrm{tr}\big[(Q + K^\top R K)\Phi_K\big] \ge \|\Phi_K\|_2 \cdot \sigma_{\min}(Q),$$

which gives the upper bound for $\Phi_K$ as follows,

$$\|\Phi_K\|_2 \le J(K_0, b_0)/\sigma_{\min}(Q). \tag{F.44}$$

Therefore, combining (F.37), (F.40), (F.41), (F.42), (F.43), and (F.44), we know that

$$\big\|\xi^2(\zeta)\big\|_2 \le C \cdot (M_{\zeta,1} + M_{\zeta,2}) \cdot J(K_0, b_0)^2/\sigma_{\min}^2(Q) \tag{F.45}$$
$$\cdot \big(1 + \|K\|_{\mathrm{F}}^2\big)^3 \cdot \big[1 - \rho(A - BK)\big]^{-1}.$$

By (F.36) and (F.45), we know that $\xi(\zeta) \in \mathcal{V}_\xi$ for any $\zeta \in \mathcal{V}_\zeta$. We conclude the proof of the lemma. $\qquad \square$

### F.7 PROOF OF LEMMA D.7

*Proof.* Assume that $\widetilde{z}_0 \sim \mathcal{N}(\mu_\dagger, \Sigma_\dagger)$. Following from the fact that

$$\widetilde{z}_{t+1} = L\widetilde{z}_t + \nu + \delta_t,$$

it holds that

$$\widetilde{z}_t \sim \mathcal{N}\left(L^t \mu_\dagger + \sum_{i=0}^{t-1} L^i \cdot \nu, \ (L^\top)^t \Sigma_\dagger L^t + \sum_{i=0}^{t-1} (L^\top)^i \Psi_\delta L^i\right), \tag{F.46}$$

where

$$\Psi_\delta = \begin{pmatrix} \Psi_\omega & K\Psi_\omega \\ K\Psi_\omega & K\Psi_\omega K^\top + \sigma^2 I \end{pmatrix}.$$

From (D.47), we know that $\mu_z$ takes the form of

$$\mu_z = (I - L)^{-1}\nu = \sum_{j=0}^\infty L^j \nu. \tag{F.47}$$

Therefore, combining (F.46) and (F.47), we have

$$\mathbb{E}(\widehat{\mu}_z) = \mu_z + \frac{1}{\widetilde{T}}\sum_{t=1}^{\widetilde{T}} L^t \mu_\dagger - \frac{1}{\widetilde{T}}\sum_{t=1}^{\widetilde{T}}\sum_{i=t}^\infty L^i \nu. \tag{F.48}$$

We denote by

$$\mu_{\widetilde{T}} = \sum_{t=1}^{\widetilde{T}} L^t \mu_\dagger - \sum_{t=1}^{\widetilde{T}}\sum_{i=t}^\infty L^i \nu.$$

Meanwhile, it holds that

$$\left\|\sum_{t=1}^{\widetilde{T}} L^t \mu_\dagger - \sum_{t=1}^{\widetilde{T}} \sum_{i=t}^{\infty} L^i \nu\right\|_2$$

$$\leq \sum_{t=1}^{\widetilde{T}} \rho(L)^t \cdot \|\mu_\dagger\|_2 + \sum_{t=1}^{\widetilde{T}} \sum_{i=t}^{\infty} \rho(L)^i \cdot \|\nu\|_2$$

$$\leq \left[1 - \rho(L)\right]^{-1} \cdot \|\mu_\dagger\|_2 + \left[1 - \rho(L)\right]^{-2} \cdot \|\nu\|_2$$

$$\leq M_\mu \cdot (1-\rho)^{-2} \cdot \|\mu_z\|_2, \tag{F.49}$$

where $M_\mu$ is a positive absolute constant.

For the covariance, note that for any random variables $X \sim \mathcal{N}(\mu_1, \Sigma_1)$ and $Y \sim \mathcal{N}(\mu_2, \Sigma_2)$, we know that $Z = X + Y \sim \mathcal{N}(\mu_1 + \mu_2, \Sigma)$, where $\|\Sigma\|_F \leq 2\|\Sigma_1\|_F + 2\|\Sigma_2\|_F$. Combining (F.46), we know that $\widehat{\mu}_z \sim \mathcal{N}(\mathbb{E}\widehat{\mu}_z, \widetilde{\Sigma}_{\widetilde{T}}/\widetilde{T})$, where $\widetilde{\Sigma}_{\widetilde{T}}$ satisfies that

$$\widetilde{T}/2 \cdot \|\widetilde{\Sigma}_{\widetilde{T}}\|_F \leq \sum_{t=1}^{\widetilde{T}} \rho(L)^{2t} \cdot \|\Sigma_\dagger\|_F + \sum_{t=1}^{\widetilde{T}} \sum_{i=0}^{t-1} \rho(L)^{2i} \cdot \|\Psi_\delta\|_F$$

$$\leq \left[1 - \rho(L)^2\right]^{-1} \cdot \|\Sigma_\dagger\|_F + \widetilde{T} \cdot \left[1 - \rho(L)^2\right]^{-1} \cdot \|\Psi_\delta\|_F,$$

which implies that

$$\|\widetilde{\Sigma}_{\widetilde{T}}\|_F \leq M_\Sigma \cdot (1-\rho)^{-1} \cdot \|\Sigma_z\|_F, \tag{F.50}$$

where $M_\Sigma$ is a positive absolute constant. Combining (F.48), (F.49), and (F.50), we obtain that

$$\widehat{\mu}_z \sim \mathcal{N}\left(\mu_z + \frac{1}{\widetilde{T}}\mu_{\widetilde{T}}, \; \frac{1}{\widetilde{T}}\widetilde{\Sigma}_{\widetilde{T}}\right),$$

where $\|\mu_{\widetilde{T}}\|_2 \leq M_\mu \cdot (1-\rho)^{-2} \cdot \|\mu_z\|_2$ and $\|\widetilde{\Sigma}_{\widetilde{T}}\|_F \leq M_\Sigma \cdot (1-\rho)^{-1} \cdot \|\Sigma_z\|_F$. Moreover, by the Gaussian tail inequality, it holds with probability at least $1 - \widetilde{T}^{-6}$ that

$$\|\widehat{\mu}_z - \mu_z\|_2 \leq \frac{\log \widetilde{T}}{\widetilde{T}^{1/4}} \cdot (1-\rho)^{-2} \cdot \text{poly}\big(\|\Phi_K\|_2, \|K\|_F, \|b\|_2, \|\mu\|_2\big).$$

Then we finish the proof of the lemma. $\qquad\square$

## F.8 Proof of Lemma D.8

*Proof.* We continue using the notations given in §D.3. We define

$$\widehat{F}(\zeta, \xi) = \left\{\mathbb{E}(\widehat{\psi})\zeta^1 + \mathbb{E}\big[(\widehat{\psi} - \widehat{\psi}')\widehat{\psi}^\top\big]\zeta^2 - \mathbb{E}(c\widehat{\psi})\right\}^\top \xi^2 + \big[\zeta^1 - \mathbb{E}(c)\big] \cdot \xi^1 - 1/2 \cdot \|\xi\|_2^2,$$

where $\widehat{\psi} = \widehat{\psi}(x, u)$ is the estimated feature vector. Here the expectation is only taken over the trajectory generated by the state transition and the policy $\pi_{K,b}$, conditioning on the randomness induced when calculating the estimated feature vectors. Thus, the function $\widehat{F}(\zeta, \xi)$ is still random, where the randomness comes from the estimated feature vectors. Note that $|F(\zeta, \xi) - \widetilde{F}(\zeta, \xi)| \leq |F(\zeta, \xi) - \widehat{F}(\zeta, \xi)| + |\widehat{F}(\zeta, \xi) - \widetilde{F}(\zeta, \xi)|$. Thus, we only need to upper bound $|F(\zeta, \xi) - \widehat{F}(\zeta, \xi)|$ and $|\widehat{F}(\zeta, \xi) - \widetilde{F}(\zeta, \xi)|$.

**Part 1.** First we upper bound $|F(\zeta, \xi) - \widehat{F}(\zeta, \xi)|$. Note that by algebra, we have

$$\big|F(\zeta, \xi) - \widehat{F}(\zeta, \xi)\big|$$

$$= \left|\left\{\mathbb{E}(\psi - \widehat{\psi})\zeta^1 + \mathbb{E}\big[(\psi - \psi')\psi^\top - (\widehat{\psi} - \widehat{\psi}')\widehat{\psi}^\top\big]\zeta^2 - \mathbb{E}\big[c(\psi - \widehat{\psi})\big]\right\}^\top \xi^2\right|$$

$$\leq \mathbb{E}\big(\|\psi - \widehat{\psi}\|_2\big) \cdot \Big[|\zeta^1| + \mathbb{E}\big(\|\psi - \psi'\|_2 + 2\|\widehat{\psi}\|_2\big) \cdot \|\zeta^2\|_2 + \mathbb{E}(c)\Big] \cdot \|\xi^2\|_2, \tag{F.51}$$

where the expectation is only taken over the trajectory generated by the state transition and the policy $\pi_{K,b}$. From Lemma D.7, it holds that

$$\mathbb{P}\big(\|\widehat{\mu}_z - \mu_z + 1/\widetilde{T} \cdot \mu_{\widetilde{T}}\|_2 \le C_1\big) \ge 1 - \widetilde{T}^{-6}. \tag{F.52}$$

Therefore, combining (F.52), it holds with probability at least $1 - \widetilde{T}^{-6}$ that

$$\mathbb{E}\big(\|\psi - \psi'\|_2 + 2\|\widehat{\psi}\|_2\big) \le \mathrm{poly}\big(\|\Phi_K\|_2, \|K\|_{\mathrm{F}}, \|b\|_2, \|\mu\|_2, J(K_0, b_0)\big), \tag{F.53}$$

where the expectation is conditioned on the randomness induced when calculating the estimated feature vectors. Also, we know that

$$\mathbb{E}(c) \le \mathrm{poly}\big(\|\Phi_K\|_2, \|K\|_{\mathrm{F}}, \|b\|_2, \|\mu\|_2, J(K_0, b_0)\big). \tag{F.54}$$

Therefore, combining (F.51), (F.53), (F.54), and Definition B.7, it holds with probability at least $1 - \widetilde{T}^{-6}$ that

$$\big|F(\zeta, \xi) - \widehat{F}(\zeta, \xi)\big| \le \mathbb{E}\big(\|\psi - \widehat{\psi}\|_2\big) \cdot \mathrm{poly}\big(\|\Phi_K\|_2, \|K\|_{\mathrm{F}}, \|b\|_2, \|\mu\|_2, J(K_0, b_0)\big). \tag{F.55}$$

Following from the definitions of $\psi(x, u)$ in (B.5) and $\widehat{\psi}(x, u)$ in (B.14), we upper bound $\|\psi(x, u) - \widehat{\psi}(x, u)\|_2$ for any $x$ and $u$ as

$$\|\psi(x, u) - \widehat{\psi}(x, u)\|_2^2 = \|\widehat{\mu}_z - \mu_z\|_2^2 + \big\|z(\widehat{\mu}_z - \mu_z)^\top + (\widehat{\mu}_z - \mu_z)z^\top\big\|_{\mathrm{F}}^2 + \|\mu_z\mu_z^\top - \widehat{\mu}_z\widehat{\mu}_z^\top\|_{\mathrm{F}}^2$$

$$\le \mathrm{poly}\big(\|\Phi_K\|_2, \|K\|_{\mathrm{F}}, \|b\|_2, \|\mu\|_2, J(K_0, b_0)\big) \cdot \|\widehat{\mu}_z - \mu_z\|_2^2, \tag{F.56}$$

where $\mu_z$ is defined in (D.47), $\widehat{\mu}_z$ is defined in (D.48), and $z = (x^\top, u^\top)^\top$. Also, by Lemma D.7, we know that

$$\|\widehat{\mu}_z - \mu_z\|_2 \le \frac{\log \widetilde{T}}{\widetilde{T}^{1/4}} \cdot (1 - \rho)^{-2} \cdot \mathrm{poly}\big(\|\Phi_K\|_2, \|K\|_{\mathrm{F}}, \|b\|_2, \|\mu\|_2, J(K_0, b_0)\big), \tag{F.57}$$

which holds with probability at least $1 - \widetilde{T}^{-6}$. Combining (F.55), (F.56), and (F.57), it holds with probability at least $1 - \widetilde{T}^{-6}$ that

$$\big|F(\zeta, \xi) - \widehat{F}(\zeta, \xi)\big| \le \frac{\log \widetilde{T}}{\widetilde{T}^{1/4}} \cdot (1 - \rho)^{-2} \cdot \mathrm{poly}\big(\|K\|_{\mathrm{F}}, \|b\|_2, \|\mu\|_2, J(K_0, b_0)\big). \tag{F.58}$$

**Part 2.** We now upper bound $|\widehat{F}(\zeta, \xi) - \widetilde{F}(\zeta, \xi)|$ in the sequel. By definitions, we have

$$\big|\widetilde{F}(\zeta, \xi) - \widehat{F}(\zeta, \xi)\big|$$

$$= \bigg|\Big\{\mathbb{E}(\widetilde{\psi} - \widehat{\psi})\zeta^1 + \mathbb{E}\big[(\widetilde{\psi} - \widetilde{\psi}')\widetilde{\psi}^\top - (\widehat{\psi} - \widehat{\psi}')\widehat{\psi}^\top\big]\zeta^2 - \mathbb{E}(\widetilde{c}\widetilde{\psi} - \widehat{c}\widehat{\psi})\Big\}^\top \xi^2 + \mathbb{E}(\widehat{c} - \widetilde{c})\xi^1\bigg|$$

$$\le \bigg|\Big\{\mathbb{E}(\widehat{\psi})\zeta^1 + \mathbb{E}(\widehat{\psi}\widehat{\psi}^\top)\zeta^2 - \mathbb{E}(\widehat{c}\widehat{\psi})\Big\}^\top \xi^2 + \mathbb{E}(\widehat{c})\xi^1\bigg| \cdot \mathbb{1}_{\mathcal{E}^c} \tag{F.59}$$

$$\quad + \bigg|\big[\mathbb{E}(\widehat{\psi}'\widehat{\psi}^\top)\zeta^2\big]^\top \xi^2\bigg| \cdot \mathbb{1}_{(\mathcal{E}' \cap \mathcal{E})^c},$$

where we define the event $\mathcal{E}'$ as

$$\mathcal{E}' = \bigg(\bigcap_{t \in [T]} \Big\{\big|\|z_t' - \mu_z + 1/\widetilde{T} \cdot \mu_{\widetilde{T}}\|_2^2 - \mathrm{tr}(\widetilde{\Sigma}_z)\big| \le C_1 \cdot \log T \cdot \|\widetilde{\Sigma}_z\|_2\Big\}\bigg)\bigcap \mathcal{E}_2,$$

where $\mathcal{E}_2$ is defined in (D.55). Combining the fact that $\mathbb{P}(\mathcal{E}_2) \ge 1 - \widetilde{T}^{-6}$ and Lemma G.3, it holds that $\mathbb{P}(\mathcal{E}') \ge 1 - T^{-5} - \widetilde{T}^{-6}$. Following a similar argument as in **Part 1**, it holds from (F.59) that

$$\big|\widetilde{F}(\zeta, \xi) - \widehat{F}(\zeta, \xi)\big| \le \bigg(\frac{1}{T} + \frac{1}{\widetilde{T}^{1/4}}\bigg) \cdot \mathrm{poly}\big(\|K\|_{\mathrm{F}}, \|b\|_2, \|\mu\|_2, J(K_0, b_0)\big) \tag{F.60}$$

for sufficiently large $T$ and $\widetilde{T}$.

Now, combining (F.58) and (F.60), by triangle inequality, it holds with probability at least $1 - \widetilde{T}^{-6}$ that

$$\big|F(\zeta, \xi) - \widetilde{F}(\zeta, \xi)\big| \le \bigg(\frac{1}{2T} + \frac{\log \widetilde{T}}{\widetilde{T}^{1/4}}\bigg) \cdot (1 - \rho)^{-2} \cdot \mathrm{poly}\big(\|K\|_{\mathrm{F}}, \|b\|_2, \|\mu\|_2, J(K_0, b_0)\big).$$

We finish the proof of the lemma. $\qquad \square$

### F.9 PROOF OF LEMMA E.2

*Proof.* Recall that the feature vector $\psi(x, u)$ takes the following form

$$\psi(x, u) = \begin{pmatrix} \mathrm{svec}\big[(z - \mu_z)(z - \mu_z)^\top\big] \\ z - \mu_z \end{pmatrix}.$$

We then have

$$\psi(x, u) - \psi(x', u') = \begin{pmatrix} \mathrm{svec}\big[yy^\top - (Ly + \delta)(Ly + \delta)^\top\big] \\ y - (Ly + \delta) \end{pmatrix}, \tag{F.61}$$

where we denote by $y = z - \mu_z$, and $(x', u')$ is the state-action pair after $(x, u)$ following the state transition and the policy $\pi_{K,b}$. Therefore, for any symmetric matrices $M$, $N$ and any vectors $m$, $n$, it holds from (B.7) and (F.61) that

$$\begin{pmatrix} \mathrm{svec}(M) \\ m \end{pmatrix}^\top \Theta_{K,b} \begin{pmatrix} \mathrm{svec}(N) \\ n \end{pmatrix}$$

$$= \mathbb{E}_{y,\delta}\left\{ \begin{pmatrix} \mathrm{svec}(M) \\ m \end{pmatrix}^\top \begin{pmatrix} \mathrm{svec}(yy^\top) \\ y \end{pmatrix} \begin{pmatrix} \mathrm{svec}\big[yy^\top - (Ly + \delta)(Ly + \delta)^\top\big] \\ y - (Ly + \delta) \end{pmatrix}^\top \begin{pmatrix} \mathrm{svec}(N) \\ n \end{pmatrix} \right\}$$

$$= \mathbb{E}_{y,\delta}\left\{ \big(\langle M, yy^\top\rangle + m^\top y\big) \cdot \big[\langle N, yy^\top - (Ly + \delta)(Ly + \delta)^\top\rangle + n^\top(y - Ly - \delta)\big] \right\}$$

$$= \underbrace{\mathbb{E}_y\big(\langle yy^\top, M\rangle \cdot \langle yy^\top - Lyy^\top L^\top - \Psi_\delta, N\rangle\big)}_{A_1} + \underbrace{\mathbb{E}_y\big(\langle yy^\top, M\rangle \cdot n^\top(y - Ly)\big)}_{A_2} \tag{F.62}$$

$$+ \underbrace{\mathbb{E}_y\big(m^\top y \cdot \langle yy^\top - Lyy^\top L^\top - \Psi_\delta, N\rangle\big)}_{A_3} + \underbrace{\mathbb{E}_y\big[m^\top y \cdot n^\top(y - Ly)\big]}_{A_4},$$

where the expectations are taken over $y \sim \mathcal{N}(0, \Sigma_z)$ and $\delta \sim \mathcal{N}(0, \Psi_\delta)$. We evaluate the terms $A_1$, $A_2$, $A_3$, and $A_4$ in the sequel.

For the terms $A_2$ and $A_3$ in (F.62), by the fact that $y = z - \mu_z \sim \mathcal{N}(0, \Sigma_z)$, we know that these two terms vanish. For $A_4$, it holds that

$$A_4 = \mathbb{E}_y\big[m^\top y \cdot (y - Ly)^\top n\big] = \mathbb{E}_y\big[m^\top yy^\top(I - L)^\top n\big] = m^\top \Sigma_z(I - L)^\top n. \tag{F.63}$$

For $A_1$, by algebra, we have

$$A_1 = \mathbb{E}_y\big(\langle yy^\top, M\rangle \cdot \langle yy^\top - Lyy^\top L^\top - \Psi_\delta, N\rangle\big)$$

$$= \mathbb{E}_y\big(\langle yy^\top, M\rangle \cdot \langle yy^\top - Lyy^\top L^\top, N\rangle\big) - \mathbb{E}_y\big(\langle yy^\top, M\rangle \cdot \langle \Psi_\delta, N\rangle\big)$$

$$= \mathbb{E}_y\big[y^\top My \cdot y^\top(N - L^\top NL)y\big] - \langle \Sigma_z, M\rangle \cdot \langle \Psi_\delta, N\rangle$$

$$= \mathbb{E}_{u \sim \mathcal{N}(0, I)}\big[u^\top \Sigma_z^{1/2} M \Sigma_z^{1/2} u \cdot u^\top \Sigma_z^{1/2}(N - L^\top NL)\Sigma_z^{1/2} u\big] - \langle \Sigma_z, M\rangle \cdot \langle \Psi_\delta, N\rangle. \tag{F.64}$$

Now, by applying Lemma G.1 to the first term on the RHS of (F.64), we know that

$$A_1 = 2\,\mathrm{tr}\big[\Sigma_z^{1/2} M \Sigma_z^{1/2} \cdot \Sigma_z^{1/2}(N - L^\top NL)\Sigma_z^{1/2}\big]$$

$$+ \mathrm{tr}(\Sigma_z^{1/2} M \Sigma_z^{1/2}) \cdot \mathrm{tr}\big[\Sigma_z^{1/2}(N - L^\top NL)\Sigma_z^{1/2}\big] - \langle \Sigma_z, M\rangle \cdot \langle \Psi_\delta, N\rangle$$

$$= 2\langle M, \Sigma_z(N - L^\top NL)\Sigma_z\rangle + \langle \Sigma_z, M\rangle \cdot \langle \Sigma_z - L\Sigma_z L^\top - \Psi_\delta, N\rangle$$

$$= 2\langle M, \Sigma_z(N - L^\top NL)\Sigma_z\rangle,$$

where we use the fact that $\Sigma_z = L\Sigma_z L^\top + \Psi_\delta$ in the last equality. By using the property of the operator $\mathrm{svec}(\cdot)$ and the definition of the symmetric Kronecker product, we obtain that

$$A_1 = 2\mathrm{svec}(M)^\top \mathrm{svec}\big[\Sigma_z(N - L^\top NL)\Sigma_z\big]$$

$$= 2\mathrm{svec}(M)^\top \big[\Sigma_z \otimes_s \Sigma_z - (\Sigma_z L^\top) \otimes_s (\Sigma_z L^\top)\big]\mathrm{svec}(N)$$

$$= 2\mathrm{svec}(M)^\top \big[(\Sigma_z \otimes_s \Sigma_z)(I - L \otimes_s L)^\top\big]\mathrm{svec}(N). \tag{F.65}$$

Combining (F.62), (F.63), and (F.65), we obtain that

$$\begin{pmatrix} \text{svec}(M) \\ m \end{pmatrix}^\top \Theta_{K,b} \begin{pmatrix} \text{svec}(N) \\ n \end{pmatrix}$$

$$= \text{svec}(M)^\top \left[ 2(\Sigma_z \otimes_s \Sigma_z)(I - L \otimes_s L)^\top \right] \text{svec}(N) + m^\top \Sigma_z (I - L)^\top n$$

$$= \begin{pmatrix} \text{svec}(M) \\ m \end{pmatrix}^\top \begin{pmatrix} 2(\Sigma_z \otimes_s \Sigma_z)(I - L \otimes_s L)^\top & 0 \\ 0 & \Sigma_z(I - L)^\top \end{pmatrix} \begin{pmatrix} \text{svec}(N) \\ n \end{pmatrix}.$$

Thus, the matrix $\Theta_{K,b}$ takes the following form,

$$\Theta_{K,b} = \begin{pmatrix} 2(\Sigma_z \otimes_s \Sigma_z)(I - L \otimes_s L)^\top & 0 \\ 0 & \Sigma_z(I - L)^\top \end{pmatrix},$$

which concludes the proof of the lemma. $\qquad\square$

### F.10 PROOF OF LEMMA E.3

*Proof.* From the definition of $\widetilde{\Theta}_{K,b}$ in (B.9), it holds that

$$\|\widetilde{\Theta}_{K,b}^{-1}\|_2^2 \le 1 + \|\Theta_{K,b}^{-1}\|_2^2 + \|\Theta_{K,b}^{-1}\widetilde{\sigma}_z\|_2^2, \tag{F.66}$$

where $\widetilde{\sigma}_z$ is defined as

$$\widetilde{\sigma}_z = \mathbb{E}_{\pi_{K,b}}\big[\psi(x,u)\big] = \begin{pmatrix} \text{svec}(\Sigma_z) \\ \mathbf{0}_{k+m} \end{pmatrix}.$$

We bound the RHS of (F.66) in the sequel. For the term $\Theta_{K,b}^{-1}\widetilde{\sigma}_z$, combining Lemma E.2, we have

$$\Theta_{K,b}^{-1}\widetilde{\sigma}_z = \begin{pmatrix} 1/2 \cdot (I - L \otimes_s L)^{-\top}(\Sigma_z \otimes_s \Sigma_z)^{-1} \cdot \text{svec}(\Sigma_z) \\ \mathbf{0}_{k+m} \end{pmatrix}$$

$$= \begin{pmatrix} 1/2 \cdot (I - L \otimes_s L)^{-\top}(\Sigma_z^{-1} \otimes_s \Sigma_z^{-1}) \cdot \text{svec}(\Sigma_z) \\ \mathbf{0}_{k+m} \end{pmatrix}$$

$$= \begin{pmatrix} 1/2 \cdot (I - L \otimes_s L)^{-\top} \cdot \text{svec}(\Sigma_z^{-1}) \\ \mathbf{0}_{k+m} \end{pmatrix}, \tag{F.67}$$

where we use the property of the symmetric Kronecker product in the second and last line. By taking the spectral norm on both sides of (F.67), it holds that

$$\|\Theta_{K,b}^{-1}\widetilde{\sigma}_z\|_2 = 1/2 \cdot \left\| (I - L \otimes_s L)^{-\top} \cdot \text{svec}(\Sigma_z^{-1}) \right\|_2$$

$$\le 1/2 \cdot \left\| (I - L \otimes_s L)^{-\top} \right\|_2 \cdot \left\| \text{svec}(\Sigma_z^{-1}) \right\|_2$$

$$\le 1/2 \cdot \big[ 1 - \rho^2(L) \big]^{-1} \cdot \|\Sigma_z^{-1}\|_F$$

$$\le 1/2 \cdot \sqrt{k+m} \cdot \big[ 1 - \rho^2(L) \big]^{-1} \cdot \|\Sigma_z^{-1}\|_2$$

$$= 1/2 \cdot \sqrt{k+m} \cdot \big[ 1 - \rho^2(L) \big]^{-1} \cdot \sigma_{\min}^{-1}(\Sigma_z), \tag{F.68}$$

where in the third line we use Lemma G.2 to the matrix $L \otimes_s L$. Similarly, we upper bound $\|\Theta_{K,b}^{-1}\|_2$ in the sequel

$$\|\Theta_{K,b}^{-1}\|_2 \le \min\Big\{ 1/2 \cdot \big[ 1 - \rho^2(L) \big]^{-1} \sigma_{\min}^{-2}(\Sigma_z), \big[ 1 - \rho(L) \big]^{-1} \sigma_{\min}^{-1}(\Sigma_z) \Big\}. \tag{F.69}$$

Thus, combining (F.66), (F.68), and (F.69), we obtain that

$$\|\widetilde{\Theta}_{K,b}^{-1}\|_2^2 \le 1 + 1/2 \cdot \sqrt{k+m} \cdot \big[ 1 - \rho^2(L) \big]^{-1} \cdot \sigma_{\min}^{-1}(\Sigma_z)$$

$$+ \min\Big\{ 1/2 \cdot \big[ 1 - \rho^2(L) \big]^{-1} \sigma_{\min}^{-2}(\Sigma_z), \big[ 1 - \rho(L) \big]^{-1} \sigma_{\min}^{-1}(\Sigma_z) \Big\}. \tag{F.70}$$

Now it remains to characterize $\sigma_{\min}(\Sigma_z)$. For any vectors $s \in \mathbb{R}^m$ and $r \in \mathbb{R}^k$, we have

$$
\begin{pmatrix} s \\ r \end{pmatrix}^\top \Sigma_z \begin{pmatrix} s \\ r \end{pmatrix} = \mathbb{E}_{x \sim \mathcal{N}(\mu_{K,b}, \Phi_K), u \sim \pi_{K,b}(\cdot \,|\, x)} \Big\{ \big[ s^\top (x - \mu_{K,b}) + r^\top (u + K\mu_{K,b} - b) \big]^2 \Big\}
$$

$$
= \mathbb{E}_{x \sim \mathcal{N}(\mu_{K,b}, \Phi_K), \eta \sim \mathcal{N}(0, I)} \Big\{ \big[ (s - K^\top r)^\top (x - \mu_{K,b}) + \sigma r^\top \eta \big]^2 \Big\}
$$

$$
= \mathbb{E}_{x \sim \mathcal{N}(\mu_{K,b}, \Phi_K)} \Big\{ \big[ (s - K^\top r)^\top (x - \mu_{K,b}) \big]^2 \Big\} + \mathbb{E}_{\eta \sim \mathcal{N}(0, I)} \big[ (\sigma r^\top \eta)^2 \big]. \quad \text{(F.71)}
$$

The first term on the RHS of (F.71) is lower bounded as

$$
\mathbb{E}_{x \sim \mathcal{N}(\mu_{K,b}, \Phi_K)} \Big\{ \big[ (s - K^\top r)^\top (x - \mu_{K,b}) \big]^2 \Big\} = (s - K^\top r)^\top \Phi_K (s - K^\top r)
$$

$$
\geq \|s - K^\top r\|_2^2 \cdot \sigma_{\min}(\Phi_K) \geq \|s - K^\top r\|_2^2 \cdot \sigma_{\min}(\Psi_\omega), \quad \text{(F.72)}
$$

where the last inequality comes from the fact that $\sigma_{\min}(\Phi_K) \geq \sigma_{\min}(\Psi_\omega)$ by (3.3). The second term on the RHS of (F.71) takes the form of

$$
\mathbb{E}_{\eta \sim \mathcal{N}(0, I)} \big[ (\sigma r^\top \eta)^2 \big] = \sigma^2 \|r\|_2^2. \quad \text{(F.73)}
$$

Therefore, combining (F.71), (F.72), and (F.73), we have

$$
\begin{pmatrix} s \\ r \end{pmatrix}^\top \Sigma_z \begin{pmatrix} s \\ r \end{pmatrix} \geq \|s - K^\top r\|_2^2 \cdot \sigma_{\min}(\Psi_\omega) + \sigma^2 \|r\|_2^2
$$

$$
\geq \sigma_{\min}(\Psi_\omega) \cdot \|s\|_2^2 + \big[ \sigma^2 - \|K\|_2^2 \cdot \sigma_{\min}(\Psi_\omega) \big] \cdot \|r\|_2^2.
$$

From this, we know that

$$
\sigma_{\min}(\Sigma_z) \geq \min\big\{ \sigma_{\min}(\Psi_\omega), \sigma^2 - \|K\|_2^2 \cdot \sigma_{\min}(\Psi_\omega) \big\}. \quad \text{(F.74)}
$$

Thus, combining (F.70) and (F.74), we know that $\|\widetilde{\Theta}_{K,b}^{-1}\|_2$ is upper bounded by a constant $\widetilde{\lambda}_K$, where $\widetilde{\lambda}_K$ only depends on $\|K\|_2$ and $\rho(L) = \rho(A - BK)$. This finishes the proof of the lemma. $\qquad \square$

## F.11 PROOF OF LEMMA F.2

*Proof.* First, note that the cost function $c(x, u)$ takes the following form,

$$
c(x, u) = \psi(x, u)^\top \begin{pmatrix} \mathrm{svec}\big[\mathrm{diag}(Q, R)\big] \\ 2Q\mu_{K,b} \\ 2R\mu_{K,b}^u \end{pmatrix} + \big[ \mu_{K,b}^\top Q\mu_{K,b} + (\mu_{K,b}^u)^\top R\mu_{K,b}^u + \mu^\top \overline{Q}\mu \big].
$$

For any matrix $V$ and vectors $v_x$, $v_u$, it holds that

$$
\mathbb{E}_{\pi_{K,b}} \big[ c(x, u)\psi(x, u) \big]^\top \begin{pmatrix} \mathrm{svec}(V) \\ v_x \\ v_u \end{pmatrix}
$$

$$
= \underbrace{\mathbb{E}_{\pi_{K,b}} \left\{ \psi(x, u)^\top \begin{pmatrix} \mathrm{svec}\big[\mathrm{diag}(Q, R)\big] \\ 2Q\mu_{K,b} \\ 2R\mu_{K,b}^u \end{pmatrix} \psi(x, u)^\top \begin{pmatrix} \mathrm{svec}(V) \\ v_x \\ v_u \end{pmatrix} \right\}}_{D_1} \quad \text{(F.75)}
$$

$$
+ \underbrace{\mathbb{E}_{\pi_{K,b}} \left\{ \psi(x, u)^\top (\mu_{K,b}^\top Q\mu_{K,b} + (\mu_{K,b}^u)^\top R\mu_{K,b}^u + \mu^\top \overline{Q}\mu) \begin{pmatrix} \mathrm{svec}(V) \\ v_x \\ v_u \end{pmatrix} \right\}}_{D_2}.
$$

In the sequel, we calculate $D_1$ and $D_2$ respectively.

**Calculation of $D_1$.** Note that by the definition of $\psi(x,u)$ in (B.5), it holds that

$$D_1 = \mathbb{E}_{\pi_{K,b}}\left\{\left[(z-\mu_z)^\top \operatorname{diag}(Q,R)(z-\mu_z) + (z-\mu_z)^\top \begin{pmatrix} 2Q\mu_{K,b} \\ 2R\mu_{K,b}^u \end{pmatrix}\right]\right.$$

$$\left. \cdot \left[(z-\mu_z)^\top V(z-\mu_z) + (z-\mu_z)^\top \begin{pmatrix} v_x \\ v_u \end{pmatrix}\right]\right\}$$

$$= \mathbb{E}_{\pi_{K,b}}\left[(z-\mu_z)^\top \operatorname{diag}(Q,R)(z-\mu_z) \cdot (z-\mu_z)^\top V(z-\mu_z)\right] \tag{F.76}$$

$$+ \mathbb{E}_{\pi_{K,b}}\left[\begin{pmatrix} 2Q\mu_{K,b} \\ 2R\mu_{K,b}^u \end{pmatrix}^\top (z-\mu_z)(z-\mu_z)^\top \begin{pmatrix} v_x \\ v_u \end{pmatrix}\right].$$

Here $z = (x^\top, u^\top)^\top$ and $\mu_z = \mathbb{E}_{\pi_{K,b}}(z)$. For the first term on the RHS of (F.76), note that $z - \mu_z \sim \mathcal{N}(0, \Sigma_z)$. Therefore, by Lemma G.1, we obtain that

$$\mathbb{E}_{\pi_{K,b}}\left[(z-\mu_z)^\top \operatorname{diag}(Q,R)(z-\mu_z) \cdot (z-\mu_z)^\top V(z-\mu_z)\right]$$

$$= 2\big\langle \Sigma_z \operatorname{diag}(Q,R)\Sigma_z, V\big\rangle + \big\langle \Sigma_z, \operatorname{diag}(Q,R)\big\rangle \cdot \big\langle \Sigma_z, V\big\rangle$$

$$= \operatorname{svec}\left[2\Sigma_z \operatorname{diag}(Q,R)\Sigma_z + \big\langle \Sigma_z, \operatorname{diag}(Q,R)\big\rangle \cdot \Sigma_z\right]^\top \operatorname{svec}(V). \tag{F.77}$$

Meanwhile, the second term on the RHS of (F.76) takes the form of

$$\mathbb{E}_{\pi_{K,b}}\left[\begin{pmatrix} 2Q\mu_{K,b} \\ 2R\mu_{K,b}^u \end{pmatrix}^\top (z-\mu_z)(z-\mu_z)^\top \begin{pmatrix} v_x \\ v_u \end{pmatrix}\right] = \left[\Sigma_z \begin{pmatrix} 2Q\mu_{K,b} \\ 2R\mu_{K,b}^u \end{pmatrix}\right]^\top \begin{pmatrix} v_x \\ v_u \end{pmatrix}. \tag{F.78}$$

Combining (F.76), (F.77), and (F.78), we obtain that

$$D_1 = \begin{pmatrix} 2\operatorname{svec}\left[\Sigma_z \operatorname{diag}(Q,R)\Sigma_z + \big\langle \Sigma_z, \operatorname{diag}(Q,R)\big\rangle \Sigma_z\right] \\ \Sigma_z \begin{pmatrix} 2Q\mu_{K,b} \\ 2R\mu_{K,b}^u \end{pmatrix} \end{pmatrix}^\top \begin{pmatrix} \operatorname{svec}(V) \\ v_x \\ v_u \end{pmatrix}. \tag{F.79}$$

**Calculation of $D_2$.** By the definition of the feature vector $\psi(x,u)$ in (B.5), we know that

$$D_2 = (\mu_{K,b}^\top Q\mu_{K,b} + (\mu_{K,b}^u)^\top R\mu_{K,b}^u + \mu^\top \overline{Q}\mu)\begin{pmatrix} \operatorname{svec}(\Sigma_z) \\ \mathbf{0}_m \\ \mathbf{0}_k \end{pmatrix}^\top \begin{pmatrix} \operatorname{svec}(V) \\ v_x \\ v_u \end{pmatrix}. \tag{F.80}$$

Now, combining (F.75), (F.79), and (F.80), it holds that

$$\mathbb{E}_{\pi_{K,b}}\left[c(x,u)\psi(x,u)\right] = \begin{pmatrix} 2\operatorname{svec}\left[\Sigma_z \operatorname{diag}(Q,R)\Sigma_z + \big\langle \Sigma_z, \operatorname{diag}(Q,R)\big\rangle \Sigma_z\right] \\ \Sigma_z \begin{pmatrix} 2Q\mu_{K,b} \\ 2R\mu_{K,b}^u \end{pmatrix} \end{pmatrix}$$

$$+ \left[\mu_{K,b}^\top Q\mu_{K,b} + (\mu_{K,b}^u)^\top R\mu_{K,b}^u + \mu^\top \overline{Q}\mu\right]\begin{pmatrix} \operatorname{svec}(\Sigma_z) \\ \mathbf{0}_m \\ \mathbf{0}_k \end{pmatrix},$$

which concludes the proof of the lemma. $\qquad\square$

# G  AUXILIARY RESULTS

**Lemma G.1.** Assume that the random variable $w \sim \mathcal{N}(0, I)$, and let $U$ and $V$ be two symmetric matrices, then it holds that

$$\mathbb{E}(w^\top U w \cdot w^\top V w) = 2\operatorname{tr}(UV) + \operatorname{tr}(U) \cdot \operatorname{tr}(V).$$

*Proof.* See Magnus et al. (1978) and Magnus (1979) for a detailed proof. $\qquad\square$

**Lemma G.2.** Let $M$, $N$ be commuting symmetric matrices, and let $\alpha_1, \ldots, \alpha_n, \beta_1, \ldots, \beta_n$ denote their eigenvalues with $v_1, \ldots, v_n$ a common basis of orthogonal eigenvectors. Then the $n(n+1)/2$ eigenvalues of $M \otimes_s N$ are given by $(\alpha_i \beta_j + \alpha_j \beta_i)/2$, where $1 \leq i \leq j \leq n$.

*Proof.* See Lemma 2 in Alizadeh et al. (1998) for a detailed proof. $\qquad\square$

**Lemma G.3.** For any integer $m > 0$, let $A \in \mathbb{R}^{m \times m}$ and $\eta \sim \mathcal{N}(0, I_m)$. Then, there exists some absolute constant $C > 0$ such that for any $t \geq 0$, we have

$$\mathbb{P}\Big[\big|\eta^\top A\eta - \mathbb{E}(\eta^\top A\eta)\big| > t\Big] \leq 2 \cdot \exp\Big[-C \cdot \min\big(t^2\|A\|_{\mathrm{F}}^{-2},\ t\|A\|_2^{-1}\big)\Big].$$

*Proof.* See Rudelson et al. (2013) for a detailed proof. $\qquad\square$

