# OpenReview forum: "Actor-Critic Provably Finds Nash Equilibria of Linear-Quadratic Mean-Field Games"
_ICLR.cc/2020/Conference — Accept (Poster)_

### Official Review · AnonReviewer2 · 2019-10-16
**Official Blind Review #2**

**Rating:** 8

**Review:**

Summary and Decision

This paper studied an actor-critic learning algorithm for solving a mean field game. More specifically, the authors showed a particular actor-critic algorithm converges for linear-quadratic games with a quantitative bound in Theorem 4.1. Notably, results on learning algorithms for solving mean field games without prior knowledge of the parameters are rare, so results of this type is highly desirable. However, the algorithm studied in this paper is uniquely tailored for the (very special!) linear-quadratic setting, which is very unsatisfying. We will discuss concern this in detail below.

Overall, I recommend a weak accept for this paper.

Background

Mean field games is a theory of large population games, where we assume each agent has infinitesimal contributions to the dynamics in the limit as number of agents go to infinity. Similar to mean field theory of particles, the limiting dynamics can be completely characterized by a single distribution of agents, commonly know as McKean-Vlasov dynamics. The theory drastically simplifies computation for large population games: while it is essentially impossible to find a Nash equilibrium for a 100 agent game, we can compute the Nash equilibrium for the mean field limit and approximate the finite game.

Mathematically, mean field games remain very difficult to solve even knowing the parameters and dynamics. Therefore it is often important to first study a simple case where we can solve the game analytically. In the context of optimal control and mean field games, we can often recover closed form solutions (up to the solution of a Riccati equation) when the dynamics are linear and the cost is quadratic. We call this class of games linear-quadratic mean field games (LQ-MFG). To interpret the LQ assumption, typical control problems in this setting can be recast into a convex optimization problem in the control (or strategy) using convex analysis techniques. Therefore LQ assumptions provides both theoretical and computational tractability.

Here we will specifically note the paper of Elliot, Li, and Ni, where we can find a closed form solution of the discrete time LQ-MFG with finite horizon.
https://arxiv.org/abs/1302.6416

Furthermore, we will also distinguish between games with a finite horizon and infinite horizon. While there are difficulties associated with both cases, typically an ergodic infinite horizon problem removes the time variable from the equation, making the problem slightly easier. Hence many researchers in MFG prefer to begin by studying the ergodic problem.

In the context of reinforcement learning, we are more interested in solving MFG without knowledge of underlying parameters, dynamics, or even the cost function. This direction is still relatively new for the MFG community, and many problems remain open. The ultimate goal of this line of research is to develop generic and scalable algorithms that can solve general MFGs without knowledge of the game parameters/dynamics/cost etc.


Discussion of Contributions

This work is the first analysis of actor-critic algorithms for solving MFG. At the same time, the paper studies discrete time MFGs, which is generally less popular but no less interesting. Therefore a theoretical convergence result in this setting is highly desired.

Overall, the mathematical set up of this problem is very convoluted. This likely motivated the authors to make more simplifying LQ type assumptions to recover stronger results. Even with these assumptions, to put all the pieces of the puzzle together is no easy task. The authors have to consider the interaction between the agent state and the mean field state, as well as the estimation of optimal controls and how to bound errors from estimation error. This led to a long appendix of proofs - while too lengthy to verify, the results seem sensible.

From this, I believe the mathematical analysis itself is a worthy contribution. This paper will serve as a good starting point for future analysis of more complex problem settings and other learning algorithms in MFGs.


Discussion of Limitations

The main concern regarding this paper is on quantifying how much of a contribution the results add to the broader community. While results on LQ-MFGs are always nice to have, I believe the specific actor-critic algorithm depends too much on the LQ structure for this work to be useful. Two examples of these are:

1. on page 5, above equation (2.1), the actor-critic algorithm will be only seeking policies that are linear in the state x. This is taking advantage of the fact that we know LQ-MFGs have linear optimal policies.
2. on page 7, below equation (3.7), the algorithm requires to know the form of the gradient of the cost function - and therefore leading to a direct estimation of matrices \Upsilon that form the gradient. This is only possible in LQ-MFGs.

Therefore, results from this paper will be very difficult to generalize to other actor-critic algorithms for LQ-MFGs. At the same time, it's also difficult to generalize these results to the same actor-critic algorithm for non-LQ-MFGs.

We also note that if we can assume knowledge of the LQ form of underlying dynamics and the form of the cost function, but no knowledge of the parameters, the problem reduces down to a parameter estimation problem. In this case, we can speculate the results of Elliot, Li, and Ni can be adapted to the ergodic case, and we can recover the approximate optimal controls given estimated parameters. Furthermore, in some sense, this particular actor-critic algorithm is implicitly estimating a sufficient set of parameters (the \Upsilon matrices) to find the optimal control. Essentially, if we rely too much on the LQ structure, the problem is then rendered much less interesting.

In summary, the ultimate goal of this line of research is to approximately solve non-LQ games, therefore the value of this current paper is very incremental in the larger context of learning MFGs. While serving as a reference for future analysis of related algorithms for MFGs, it will be difficult to borrow concrete ideas and results from this paper to build on.

*Edit:* I believe the discussions below have address several important concerns, and I will raise my score to accept.

**Experience Assessment:**

I have read many papers in this area.

**Review Assessment: Checking Correctness Of Derivations And Theory:**

I assessed the sensibility of the derivations and theory.

**Review Assessment: Checking Correctness Of Experiments:**

N/A

**Review Assessment: Thoroughness In Paper Reading:**

I read the paper thoroughly.

---

> ### Author Response · Authors · 2019-11-08
> **Review Response**
>
> We appreciate the valuable comments from the reviewer. We address the concerns in the follows.
>
> 1. (Discussion of Background.) We thank the reviewer for giving such a high-level introduction to mean-field games, and providing some potential future research topics that might be interesting to us. Also, we are sorry that we have missed such an important related work [1], which derives the closed form of the problem (MF-LQ) analytically via solving two AREs. However, we still want to mention that [1] considers the mean-field type linear quadratic optimal control problems, whereas our work focuses on solving linear quadratic mean-field games, which is not exactly the same setting.
>
> 2. (Linear Quadratic Setting.) We agree with the reviewer that the linear quadratic setting is a special setting where the state transition is linear, and the cost function takes a quadratic form. Despite the special formulation, this setting still well approximates many real-world tasks, such as power grids (Minciardi and Sacile, 2011), swarm robots (Fang, 2014; Araki et al., 2017; Doerr et al., 2018), and financial systems (Zhou and Li, 2000; Huang and Li, 2018), as we mentioned in the introduction.
>
> Meanwhile, our linear quadratic setting can be viewed as a model-free RL approach with function approximation (linear approximation of the dynamics and quadratic approximation of the cost function), which may serve as a starting point to deal with more general function approximations. Though the theoretical analysis tailors on the special structure of linear-quadratic setting, we believe that the algorithm itself can be extended to a more general function approximation setting.
>
> Also, our RL approach to the subproblem (drifted LQR) serves as an end-to-end approach, in the sense that we directly aim to minimize the expected total cost, which aligns with the ultimate goal for the agent. On the other hand, the well-studied LQR setting may help theoretically understand the efficiency of RL algorithms (natural/classical AC method in our work).
>
>
> [1] Elliott, Robert, Xun Li, and Yuan-Hua Ni. "Discrete time mean-field stochastic linear-quadratic optimal control problems." Automatica 49.11 (2013): 3222-3233.

---

> > ### Comment · AnonReviewer2 · 2019-11-13
> > **The Linear Quadratic Setting**
> >
> > Allow me to clarify my original point earlier. I do believe the authors have solved a difficult problem mathematically, as it involved putting together a very complex parts of the proof structure. And I am not directly criticizing the linear-quadratic (LQ) assumption, but rather whether the algorithm and proof technique is sufficiently adaptable to a non-LQ setting.
> >
> > Recall my two points listed in the limitations section - these are very specific to LQ mean field games, and it's unclear how to adapt these choices to a non-LQ setting. If we were only interested in solving an LQ problem in the first place, we should be looking for a close form solution and estimate its parameters instead. Using an actor-critic algorithm seems to be an overkill and an uninteresting solution.
> >
> > I believe the authors also recognize that the community most desires theoretical results for general algorithms and proof techniques. Therefore the ideal results should avoid relying on very specific structure of the LQ setting, i.e. not estimating the gradient directly using the Upsilon matrix.
> >
> > To raise the review score, the authors would have to convince me that
> > 1. It's reasonable in a general setting to consider only policies linear in the state, and use the gradient estimation procedure.
> > 2. It's better in the general setting to consider this AC algorithm instead of using an LQ-approximation and plugging a closed form solution.
> > 3. The proof techniques provided in this paper can be generalized to the non-LQ setting.

---

> > > ### Author Response · Authors · 2019-11-14
> > > **Review Response 2**
> > >
> > > We thank the reviewer again for the valuable comments. We address the concerns in the follows.
> > >
> > > 1. (Why linear-Gaussian policy.) Our linear-Gaussian policy belongs to the family of energy-based policies having the form of $\exp( \tau^{-1} \cdot \varrho_{\theta}(x, u) )$. For this class of policies, the function $\varrho_{\theta}$ is usually assumed to be linear in a known feature mapping. Here in the case of our linear-Gaussian policy, $\varrho_{\theta} = \varrho_{K,b}$ is linear in the following feature vector,
> > > $$
> > > \begin{pmatrix}
> > > {\text{vec}}\bigg[\begin{pmatrix}
> > > x\\
> > > u
> > > \end{pmatrix}\begin{pmatrix}
> > > x\\
> > > u
> > > \end{pmatrix}^\top\bigg]\\
> > > x\\
> > > u
> > > \end{pmatrix}.
> > > $$
> > > For the general non-LQ setting, we can directly adopt our analysis framework to energy-based policies and show similar results when the parameterization is compatible.
> > >
> > > 2. (Why AC instead of LQ closed form solution.) Our AC algorithm is readily applicable to general non-LQ mean-field games. In particular, we iteratively update the mean-field state following the current policy, and solve for the optimal policy of the MDP associated with the mean-field state using AC algorithm, which is indeed possible if the policy and value function classes are proper such that we have controllable bias in the estimated policy gradient.
> > >
> > > In the general non-LQ setting, where the feature is unknown and possesses a complicated structure, our AC algorithm may extract the underlying feature and obtains the optimal policy simultaneously. In comparison, the LQ-approximation approach requires learning the model first, which introduces an inevitable error. Meanwhile, the LQ-approximation approach may not be plausible in an online setting, whereas our AC algorithm is an online scheme.
> > >
> > > 3. (Generalization of proof techniques.) In the general non-LQ setting, the convergence of AC algorithm is still not well understood. In particular, the policy optimization in the actor update is generally non-convex, and the policy gradient estimation in the critic update has irremovable bias when the parametrization of the critic is incompatible with that of the actor, which is indeed the case when the critic is a nonlinear function of its parameters  such as a neural network.
> > >
> > > However, the proof techniques in our work may also be extended to general non-LQ setting. Our algorithm indeed falls within the framework of energy-based policy with natural policy gradient method, and we expect that the natural policy gradient still possesses the properties as in the case of quadratic cost  (e.g., the gradient dominance property). Then following from the proof techniques in our work, with compatible parameterization of value function (so that the bias of policy evaluation in the critic update is small), one can show that the AC algorithm still converges to the Nash equilibrium of the mean-field game. We note that a key feature of AC algorithm in the LQ setting is that the parametrization of the critic (value function) is compatible to that of the value function [Sutton et al. (2000)]. Our analysis framework is nevertheless general and can be extended to the general setting under proper assumptions.

---

> > > > ### Comment · AnonReviewer2 · 2019-11-14
> > > > **Linear Quadratic Setting - Continued**
> > > >
> > > > Thanks the authors for the comment. I believe I'm a bit more convinced now, and I will reply to each point separately.
> > > >
> > > > 1. I believe this is a far better justification for linear Gaussian policy than to have assumed that LQ-MFG is known to have solutions of this type. I will agree with this motivation. That being said, there are two issues to expand on:
> > > > (a) Suppose the MFG's optimal policy is not linear in the state, then this AC algorithm can never converge to the optimal policy.
> > > > (b) The AC algorithm remains to rely on an estimation of the matrix \Upsilon_K to obtain the gradient of J_1(K), and hence seeking this optimal policy. I'm not convinced that this is a good idea in the non-LQ setting, as gradients of J_1(K) may not have such an explicit form. In relation to point 2, I'm not sure if this algorithm is readily applicable to non-LQ settings.
> > > >
> > > > 2. I'm not sure if I agree with this response. From a theoretical perspective, LQ and in particular ergodic settings hints at a very likely existence of closed form solution. Now suppose we have a closed form solution to work with, we can iteratively update estimates of the parameters by continuing to play the game. While similarly the AC algorithm proposed in this work relies on estimating the gradient of J_1(K) using the LQ structure, a closed form solution can possibly find this immediately.
> > > >
> > > > The authors claim that  the "AC algorithm may extract the underlying features and obtain the optimal policy simultaneously", this implies that the AC algorithm can extract more complex features than the LQ setting contains - which would be true if the algorithm did not rely on the LQ structure as much. At this point, this claim is at best unclear.
> > > >
> > > > 3. I'm more convinced on the generality of the proof techniques. I will agree with this point.
> > > >
> > > > To summarize, I'm not quite convinced to raise my score to accept, but I believe this discussion has helped. Since there's no intermediate score in this year's ICLR review system, I hope the area chair will see this discussion as a positive.

---

> > > > > ### Author Response · Authors · 2019-11-14
> > > > > **AC algorithm in non-LQ setting**
> > > > >
> > > > > We thank the reviewer for the prompt reply and the valuable comments. We will further address some concerns in below.
> > > > >
> > > > > 1. We thank the reviewer for the agreement with our motivation for using linear-Gaussian policy. In general non-LQ setting, our AC algorithm will evaluate the policy by estimating the value function (Q function) of the policy, and then by policy gradient theorem [Sutton et al. (2000)], the policy gradient takes the form of $Q_{\omega}\nabla \log \pi_{\theta}$, which serves the descent direction in the actor update. The estimation of $\alpha_{K,b}$ (which includes $\Upsilon$, $p$, and $q$) in critic update is only a special case in our LQ setting, where the Q function is linear in $\alpha_{K,b}$. By this, we believe that the AC algorithm can be readily generalized to non-LQ setting.
> > > > >
> > > > > 2. We thank the reviewer for pointing out a possible learning procedure using LQ approximation. And we agree with the reviewer that by using the LQ approximation, one may iteratively update the parameters of the model by continuing playing the game, and then obtain the optimal (linear) policy immediately, which saves computation resources compared with our AC algorithm. We would like to highlight that, the key shortcoming of LQ approximation is that the method possesses the fundamental bias of the model, which may be accumulated so that the resulting optimal policy may deviate much from the true policy (same for the loss function). In comparison, our AC algorithm may learn the optimal policy, or at least we believe that the loss function corresponding to the resulting policy may converge to the one corresponding to the optimal policy. This may outperform the LQ approximation to some extent.
> > > > >
> > > > > We really appreciate the helpful discussion with the reviewer, which is definitely a great benefit for our paper (and future work), and we will revise our paper accordingly to reflect it.

---

> > > > > > ### Comment · AnonReviewer2 · 2019-11-15
> > > > > > **Linear Quadratic Setting - Continued**
> > > > > >
> > > > > > I believe the authors have addressed an additional concern with point 1.
> > > > > >
> > > > > > 1. At this point I'm more convinced that the Upsilon estimate type updates can be used in a general setting. The main concern I original had was whether or not the form of the gradient for J_1(K) were a special artifact of the LQ setting, but as the authors pointed out, it's has a more general motivation.
> > > > > >
> > > > > > I'm still not quite convinced that seeking only linear Gaussian updates will converge to the optimal policy in general, but I believe that's a relatively minor remaining issue now.
> > > > > >
> > > > > > 2. The main concern for this point is that I believe we should always seek a closed form solution before an approximation. I believe the AC algorithm will be far more powerful in general, as most problems in practice will not admit a closed form solution, nor we will have a model for it. Therefore the results here feel less satisfying.
> > > > > >
> > > > > > Putting it together, I'm more convinced that the AC algorithm here is not entirely motivated by the LQ setting, and the proof technique is not as restricted to the LQ setting as I previously thought. Therefore I will increase my score to accept for this paper.

---

### Official Review · AnonReviewer3 · 2019-10-24
**Official Blind Review #3**

**Rating:** 6

**Review:**

Summary
The present work is concerned with providing a provably convergent algorithm for linear-quadratic mean field games. Here, for a given average behavior of the (large number of) players, the optimal strategy is given by an LQR with drift and the solution of the mean field game consists in finding a pair of (π , μ ) such that π = π(μ) is an optimal strategy under the mean field (the average behavior) μ and μ = μ(π, μ) is the mean field obtained if all players use the strategy π, under the mean field μ. First, the authors show that under "standard" assumptions the map μ ↦ μ(π(μ)), μ) is contractive and hence, by the Banach fixed point theorem, has a unique solution, resulting in a unique Nash equilibrium of the mean-field game. Second, they show that by using an actor-critic method to approximate π(μ) for a given μ, this argument can be turned into an algorithm with provably linear convergence rate. The authors prove a natural result that seems to be technically nontrivial (I did not have the time to follow their proof in detail). Thus, I believe the paper should be accepted.

Questions/Suggestions to the author
(1) It might be helpful for the reader to include a rough sketch of the proof and algorithm earlier in the paper

(2) Since, as you mention, Assumption 3.1 (ii) "is standard in the literature", I would assume that it has been used before in order to prove existence of Nash equilibria using the Banach fixed point theorem? If so, I would suggest pointing this out to the reader and briefly mentioning the differences (if any) to the existing proofs in the literature.

(3) On the bottom of page 4 you argue that the expectation of the state converges to a constant vector in the limit of large time, "since the Markov vhain of states ... admits a stationary distribution". In general, the existence of a stationary distribution does not imply convergence to the stationary distribution. Could please explain?

**Experience Assessment:**

I do not know much about this area.

**Review Assessment: Checking Correctness Of Derivations And Theory:**

I assessed the sensibility of the derivations and theory.

**Review Assessment: Checking Correctness Of Experiments:**

N/A

**Review Assessment: Thoroughness In Paper Reading:**

I read the paper at least twice and used my best judgement in assessing the paper.

---

> ### Author Response · Authors · 2019-11-08
> **Review Response**
>
> We appreciate the valuable comments from the reviewer. We address the concerns in the follows.
>
> 1. (Proof Sketch.) We thank the reviewer for pointing it out. We have provided a sketch of the proof of the theorem  in the revision.
>
> 2. (Proof of Existence and Uniqueness.) Yes, Banach fixed point theorem is used in related literature to prove the existence and uniqueness of the Nash equilibrium of mean-field games, and Assumption 3.1(ii) provides the contractibility of the operator $\Lambda(\cdot)$. Similar assumptions also show up in Bensoussan et al., 2016 and Saldi et al., 2018b.
>
> As mentioned in the proof of Proposition 3.2, we follow similar arguments as in the proof of Theorem 1.1 in Sznitman (1991), Theorem 3.2 (and Proposition 3.1 and Theorem 3.3) in Bensoussan et al. (2016), and Theorem 3.3 in Saldi et al., 2018b.
>
> 3. (Convergence to Stationary Distribution.) We agree that in general, only the existence of a stationary distribution does not imply convergence to the stationary distribution. In our case, where the induced Markov chain has discrete-time and continuous state space, we can still rigorously argue as follows. Since it is known that if the Markov chain is irreducible, aperiodic, and positive recurrent, then the Markov chain converges to a stationary distribution. The three properties are easily verified by noting that there is a Gaussian noise term in the state transition. Therefore, the induced Markov chain $\{x_t^*\}_{t\geq 0}$ converges to a stationary distribution, which yields the convergence of $\mathbb{E} x_t^*$ to a constant vector $\mu^*$ as $t\to\infty$.

---

> > ### Comment · AnonReviewer3 · 2019-11-14
> > **Thanks for the response**
> >
> > Thank you for the additional clarification.

---

### Official Review · AnonReviewer1 · 2019-10-27
**Official Blind Review #1**

**Rating:** 6

**Review:**

This paper considers the problem of model-free reinforcement learning in discrete-time, linear-quadratic Markovian mean-field games with ergodic costs. The authors begin by establishing the existence and uniqueness of the Nash equilibrium in such a setting, and then proposes an mean-field actor-critic algorithm with linear function approximation. The proposed algorithm is shown to converge linearly with high probability under certain standard conditions.

The paper is novel in the sense that it extends the recent previous works on model-free learning of MFGs to continuous state-action state spaces, while showing linear global convergence under certain conditions. To my knowledge, the previous works either considers the discrete state-action spaces [Guo et al. (2019)] or has only convergence to local Nash equilibrium (NE) [Jayakumar and Aditya (2019)]. However, I have the following concerns and suggestions for this paper.

1. Some claims of contribution is not very accurate. For example, the paper claims that the proposed algorithm does not require a simulator but only observations of trajectories. However, to invoke Algorithm 2 (mixed actor-critic), one has to fix the mean-field state \mu, which would have required a simulator for running the mixed actor-critic algorithm. Otherwise, the \mu could change if completely following the trajectory. This is the same setting as in some previous works like [Guo et al. (2019)]. The authors may want to double check if this kind of high level claims are accurate or not.

2. The problem setting is not very well stated in Section 2.
1) The authors should better call the problem at the beginning of Section 2 a "linear-quadratic mean-field N_a-player game", instead of a "linear-quadratic mean-field game", to differentiate from Problem 2.1 below.
2) The dimensions and assumptions of A, B, Q, R are also not mentioned until Section 3.1, which is also slightly breaking the reading flow.
3) The policies are also not clearly defined -- e.g., are they stationary or non-stationary, random or deterministic? And are we considering symmetric Nash equilibrium only (which should be so according to the later parts), i.e., all policies \pi^i are the same?
4) In the definition of Problem 2.1, the Nash policy \pi^\star is stated without even defining what the Nash in such a problem is. Similarly, after problem 2.2, Nash equilibrium is mentioned again without defining it. To address the issue, the authors may want to rewrite the cost function in problem 2.1 as J(\pi,\pi'), where \pi and \pi' are two arbitrary policies, and \pi' is not necessarily the Nash policy. Then x_t' (instead of x_t^\star) is the trajectory generated by \pi'.
5) The authors should not mention problem 2.2 right after problem 2.1. Instead, the authors should add a problem called LQ-SMFG (linear-quadratic stationary MFG), which is basically problem 2.2 but the goal is to simultaneously find \mu^\star and \pi_{\pu}^\star. For such a problem, the objective function can be written as J(\pi,\mu), where \mu basically serves the same role as \pi' in the suggested modification to problem 2.1 above. The original problem 2.2, which is the subproblem of finding \pi_{\mu}^\star given \mu according to Section 3, should be put after introducing \Lambda_1, as it is exactly what \Lambda_1 is solving. The paper should then completely focus on this LQ-SMFG instead of LQ-MFG, as explained in the next point.
6) In Definition 2.3, it should refer to LQ-SMFG mentioned above, as \mu does not even appear in problem 2.1. In addition, the definition of \Lambda_2 is also not clear. The authors may want to state it more clearly, e.g., using a one-step definition as in [Guo et al. (2019)].

3. The mixed actor-critic algorithm (with linear approxiamtion) for the subproblem D-LQR for evaluating \Lambda_1 is not well motivated.
1) For example, the authors should better highlight the difficulty of having the drift terms. The authors do show through propositions 3.3 and 3.4 how they decompose the objective into a standard LQR problem (J_1) and the problem w.r.t. a drift term (J_2). However, it is not clear why this is a must. In particular, why can't we just simply apply the natural actor-critic algorithm on the joint space of K and b?
2) Also linear approximation is not mentioned in the main text, which should be discussed given its appearance in the abstract. Otherwise, it seems to be a low-hanging fruit given the previous works like [Yang et al. (2019)].
3) Why do the authors use natural actor-critic for finding K, but just classical actor-critic to find \mu? This should be further explained. And instead of referring to appendix B repeatedly, the authors might want to directly state the assumptions needed for the input parameters on a high level to make the paper more self-contained (e.g., that the subproblem iteration numbers exceed certain threshold).

Some minor suggestions.
1) The discussion about why the Markov chain of states generated by the Nash policy \pi^\star admits a stationary distribution on page 4 is not clear. In general, don't we need additional assumptions like the ergodicity of the induced Markov chain?
2) The claim that there exists a unique optimal policy \pi_{\mu}^\star of Problem 2.2 on page 5 is not clearly stated with the necessary assumptions. The authors should at least mention that under certain standard conditions, etc.
3) In (2.1), there should also be \sigma\in \mathbb{R}.
4) On top of page 6, the authors may want to give an example of the so-called "mild regularity conditions" (e.g., positive definite of Q and R, etc.).
5) At the bottom of page 6, P_K is not defined clearly -- is it the solution to the Riccati equation? And how does it relate to the X in the Riccati equation of assumption 3.1(i)?



**Experience Assessment:**

I have published one or two papers in this area.

**Review Assessment: Checking Correctness Of Derivations And Theory:**

I carefully checked the derivations and theory.

**Review Assessment: Checking Correctness Of Experiments:**

N/A

**Review Assessment: Thoroughness In Paper Reading:**

I read the paper thoroughly.

---

> ### Author Response · Authors · 2019-11-08
> **Review Response**
>
> We appreciate the valuable comments from the reviewer. We address the concerns in the follows.
>
> 1. (Simulator.) We are sorry about the inaccurate claim. Yes, our method requires obtaining the observations of trajectories under a given mean-field state. However, our method is more realizable than Guo et al., 2019, where a simulator is necessary to generate the next state given any state-action pair under any aggregated effect of the population.
>
> 2. (Problem Setting in Section 2.) We thank the reviewer for the valuable suggestions on the structure of Section 2. Let us briefly clarify the setting of Problem 2.1, which follows a similar statement as in Bensoussan et al. (2016). In the linear-quadratic mean-field $N_{\rm a}$-player game, we are interested in finding a Nash equilibrium such that the expected total cost cannot be further decreased by unilaterally deviating one's policy. In the definition of Problem 2.1, we assume that such a Nash equilibrium exists, and since we take the number of players $N_{\rm a}$ to infinity and consider only the mean-field setting, we are only interested in symmetric equilibrium, we then call this Nash equilibrium as Nash policy $\pi^*$. We are sorry that we did not make the definition clearly before the statement of the problem, and we have revised this section according to the reviewer's suggestions to make it more self-contained and uploaded a revision.
>
> 3. (Motivation of AC Algorithm.) We address the concerns as follows.
>
> 1) (Decomposition of Expected Total Cost.) The gradients of the expected total cost $J(K,b)$ take very complicated forms due to the existence of the drift terms, which might not be able to evaluate using our primal-dual gradient temporal difference method to obtain a finite time convergence analysis. Moreover, the gradient dominance property of $J_1(K)$ no longer holds for $J(K,b)$. Thus, we decompose $J(K,b)$ as $J_1(K)$ and $J_2(K,b)$, and carefully design our AC algorithm under the merit of Proposition 3.4. We have revised the paper accordingly to highlight this point.
>
> 2) (Yang et al. (2019) and Function Approximation.) Compared with Yang et al. (2019), our work contains an extra drift term, which enforces the mean of the state to deviate from zero. This extra drift term breaks the elegant structure of classical LQR problem considered in Yang et al. (2019). To overcome the difficulties, we split the expected total cost and use an another independent AC method to deal with the drift term.
>
> Also, We study the linear-quadratic setting here, since this is a well-known problem in the control community, which possessed a specific feature mapping and a explicit form of solution. Whereas our theoretical analysis tailors on the specific structure of linear-quadratic setting, our AC method itself can be extended to mean-field games with general function approximation.
>
> 3) (Natural AC and Classical AC.) The natural gradient of $J_1(K)$ happens to be $2(\Upsilon_K^{22}K - \Upsilon_{K}^{21})$. By using natural AC, we avoid estimating the covariance matrix $\Phi_K$, comparing with using classical AC. On the other hand, the natural gradient of $J_2(K,b)$ w.r.t. $b$ does not take such an elegant form. Thus, we prefer using classical AC to find optimal $b$. However, we can still use natural AC to find $b$, which may require estimating the covariance matrix $\Phi_K$ and bring extra challenge in the proof.

---

> > ### Author Response · Authors · 2019-11-08
> > **Review Response (Cont.)**
> >
> > 4. (Minor Suggestions.) We address the concerns as follows.
> >
> > 1) (Existence of Stationary Distribution.) We thank the reviewer for pointing out this unclear argument. Yes, we agree that the existence of stationary distribution does require the ergodicity of the induced Markov chain. This can be rigorously justified in our discrete-time continuous-state Markov chain setting with a Gaussian noise term in the state transition. Note that a controllable linear system using linear quadratic optimal control is always stable. This together with the fact that the linear closed-loop dynamics is driven by Gaussian noise guarantees the existence of stationary distribution. We have revised the paper accordingly to clarify this point.
> >
> > 2) (Existence of Unique Optimal Policy.) We thank the reviewer for pointing it out. Yes, the existence of the unique optimal policy requires the assumption that the positive definite solution of the algebraic Riccati equation is unique (Anderson and Moore, 2007; Lewis et al., 2012), which requires the conditions stated in the following point 4). We have revised the paper accordingly to clarify this point.
> >
> > 3) (Definition of Linear-Gaussian Policy in (2.1).) We thank the reviewer for pointing it out, we have already specified that $\sigma \in \mathbb{R}$ in the revision. The real number $\sigma$ is a user-defined parameter, which reflects how exploratory the policy is. During the discussion in the paper, we fix $\sigma$ but learn $K$ and $b$ given the data.
> >
> > 4) (Assumption 3.1(i).) Specifically speaking, the uniqueness of the positive definite solution of algebraic Riccati equation needs the following conditions (De Souza et al. (1986); Lewis et al., 2012): (1) $R$ is positive; (2) $Q = C^\top C$ is nonnegative; (3) $(A,C)$ is observable; (4) $(A,B)$ is stabilizable. We have revised the paper accordingly to make it more specific.
> >
> > 5) (Bellman Equation and Algebraic Riccati Equation.) In (3.4), we give the Bellman equation, where $P_K$ is its unique solution. The derivation of this equation follows the dynamic programming principle, which can be found in Lewis et al., 2012, Anderson and Moore, 2007, or https://stanford.edu/class/ee363/lectures/dlqr.pdf, for example.
> >
> > Moreover, in our discrete-time LQR setting, the algebraic Riccati equation is indeed the Bellman optimality equation (Section 11.2 in Lewis et al., 2012). Therefore, if $K$ takes the optimal $K^* = -(B^\top X^* B + R)^{-1} B^\top X^* A$, where $X^*$ is the unique solution to the algebraic Riccati equation, then the resulting $P_{K^*}$ boils down to $X^*$.

---

> > > ### Comment · AnonReviewer1 · 2019-11-14
> > > **Thanks for the detailed clarifications!**
> > >
> > > I'd like to thank the authors for the detailed clarifications and the modifications in the updated draft accordingly. In particular, the modifications to Section 2 make the settings much clearer, and I realized the contribution in addition to (Yang et al., 2019) given the additional difficulty drift terms. The changes have resolved many of my concerns raised above, but there are still two remaining issues.
> > >
> > > Firstly, I still don't agree with the authors' claim about the simulators. The requirement for the simulator in this paper and in (Guo et al., 2019) should be exactly the same -- the difference is nothing but about wording. Both papers require obtaining the observations of trajectories under a given mean-field state, and neither the Q-learning used in (Guo et al., 2019) or the AC used in this paper requires generating the next state given any state-action pair. It seems that (Guo et al., 2019) has stated the requirement in a stronger form than necessary in theory, though. But I don't think in practice there are any situations where you can fix the mean-field/population aggregate states/actions but not the single-agent's states/actions. So I think this is not an improvement that should be claimed in any sense. A real improvement in this direction would be something like a relaxation of the necessity for fixing the mean-field state.
> > >
> > > Secondly, the (linear) function approximation should be elaborated in the main text, even if it's already addressed in (Yang et al., 2019). Either claiming the additional difficulty/contribution related to it or claiming that it is a direct implication from (Yang et al., 2019) is okay. Otherwise, it would be a bit misleading to mention it in the abstract.
> > >
> > > Nevertheless, these two issues mentioned above are not major, and indeed I have improved my mental rating from 5 to 6. But given the rating system of ICLR this year, this change cannot be reflected on the eventual rating, unfortunately.

---

> > > > ### Author Response · Authors · 2019-11-14
> > > > **Address the two furthre comments made the reviewer**
> > > >
> > > > We thank the reviewer for the appreciation of our revision. We further address the concerns below.
> > > >
> > > > 1. (Simulator.) Both [Guo et al., 2019] and our work require a simulator, which runs the system under a fixed mean-field effect.
> > > > However, the simulators in our work differ from that used in [Guo et al., 2019].
> > > > Note that when the mean-field effect is fixed, the agent is faced with a single-agent MDP.  [Guo et al., 2019] solves this MDP by Q-learning and our work solves it by actor-critic.
> > > >
> > > > As shown in page 7 of [Guo et al. 2019] (https://arxiv.org/pdf/1901.09585.pdf) and also Lemma 8 adopted from [Even-Dar, 2003], which is used for proving Theorem 2, the Q-learning algorithm in [Guo et al., 2019] and its analysis is borrowed from [Even-Dar, 2003].
> > > > To implement such a  Q learning algorithm,   [Even-Dar, 2003]  assumes having access to a simulator, which is also known as the ``generative model'' in RL literature [Kakade, 2003]. More importantly,  this simulator assumes that, for any state-action pair $(s,a)$, we can sample the next state $s' \sim P(\cdot | s,a)$ and reward $r(s,a)$. Notice that $(s,a)$ is an arbitrary state-action pair.
> > > >
> > > > In contrast, we only assume that we are only able to sample from the stationary distribution induced by any fixed policy $\pi$. Thus, in our algorithm, we cannot sample an arbitrary state-action pair $(s,a)$ as it is not a stationary distribution induced by any policy. Moreover, the assumption of sampling a stationary distribution of a Markov chain is a realistic assumption as long as the Markov chain mixes rapidly, which is indeed the case in our problem.
> > > >
> > > > Thus, our simulator is different from the generative model used in [Guo et al. 2019] and seems slightly weaker. We agree with the reviewer that it is not a major issue, and we will further revise our wording accordingly in the future revision to make our claim more accurate.
> > > >
> > > > 2. (Function Approximation.) In our work, we use a quadratic function to approximate the value functions due to the specific structure of the LQR problem. We believe that other more complicated function approximation may still work in theory under a similar analysis framework. We are sorry for the misleading statements, and we will further revise it accordingly.
> > > >
> > > > References:
> > > >
> > > > -- [Kakade, 2003] Kakade, Sham Machandranath. On the sample complexity of reinforcement learning. Diss. University of London, 2003.
> > > >
> > > > --[Even-Dar, 2003] E. Even-Dar and Y. Mansour. Learning rates for Q-learning. Journal of Machine Learning Research, 5(Dec):1–25, 2003.

---

> > > > > ### Comment · AnonReviewer1 · 2019-11-14
> > > > > **The further rebuttal related to the simulator does not make sense**
> > > > >
> > > > > 1. Yes, (Guo et al., 2019) utilizes the result from (Even-Dar, 2003). However, please note that there are two versions of Q-learning discussed in (Even-Dar, 2003), and the asynchronous Q-learning does not need the access to a simulator that needs to sample the next state and reward from any state-action pair, but instead only assumptions on the covering time L. Given the assumption mentioned in Lemma 8 in (Guo et al., 2019), they should be using the asynchronous Q-learning. Only the synchronous Q-learning needs the more restricted assumption on the simulator. I do understand that it is easy to be confused about these technical details, but the authors do need to check more carefully when claiming a contribution.
> > > > >
> > > > > I agree that sampling from a stationary distribution is generally weaker and easier than sampling an arbitrary state-action pair. And I encourage the authors to add this explanation to the main text. But this is more of a conceptual contribution instead of being technical, as the previous work has also achieved this goal as explained above.
> > > > >
> > > > > Note that I'm not meaning that this will downgrade the contribution of the current paper in any case. And as mentioned above, my (mental) score improvement is not affected by this point. However, I think such a claim is over-claiming, and can be confusing to the general readers, follow-up works, and is unfair to previous works. And I have been also curious about and puzzled with the simulator assumptions when reading (Yang et al., 2018b), (Guo et al., 2019) and (Jayakumar and Aditya, 2019) and other recent related works. And so I'd like to make the effort for the clarification.
> > > > >
> > > > > 2. Got it! Please add such clarifications and discussions to make the paper clearer.
> > > > >
> > > > > Thanks!

---

> > > > > > ### Author Response · Authors · 2019-11-14
> > > > > > **Further Distinguish the Difference Between the Simulator in our paper and [Guo et al, 2019]**
> > > > > >
> > > > > > We thank the reviewer for the prompt reply. In the following, we further elaborate on the assumptions made on the simulators and address the concern raised by the reviewer in the previous comment. This turns out to be a very subtle issue arising from reinforcement learning itself (and is less related to the mean-field part), which we think is worth clarifying (regardless of the claims of contribution).
> > > > > >
> > > > > > We agree with the reviewer that the Q-learning algorithm used in [Guo et al, 2019] is the asynchronous Q-learning introduced in [Even-Dar,2003] and that asynchronous Q-learning does not use the `"generator" directly, thus allowing the algorithm to be implemented with a trajectory of data.
> > > > > >
> > > > > > However, we would like to emphasize that, for the theory in [Even-Dar, 2003] to work, the trajectory used to implement the asynchronous Q learning has "covering time" bounded by $L$. Which means that, starting from any time-step $t$, $L$ consecutive state-action pairs $(s_t, a_t), (s_{t+1}, a_{t+1}), \ldots, (s_{t+L}, a_{t+L})$  covers all state-action pairs $\{(s,a) \colon s\in \mathcal{S}, a \in \mathcal{A} \}$. Note that $t$ is arbitrary.
> > > > > >
> > > > > > As far as we are concerned, in [Guo et al, 2019], when solving each MDP under a fixed mean-field effect (let us use $\mathcal{M}_k$ to denote the MDP induced by the $k$-th iteration of the mean-field effect), the number of total iterations $T_k$ of asynchronous Q-learning depends on the covering time of the trajectories sampled from $\mathcal{M}_k$. However, in their Theorem 2, they did no explicitly define what kind of assumption is imposed on the covering times of all $\mathcal{M}_k$'s. This leads to two possibilities: 1) Assuming that all these $\mathcal{M}_k$'s have covering times bounded by some $L$, then the $T_k$ has a uniform upper bound over $k$; and 2) $T_k$ is chosen according to the covering number of $\mathcal{M}_k$ so as to make Lemma 8 hold.
> > > > > >
> > > > > > We hypothesize that [Guo et al, 2019] considers the second scenario as they did not impose extra assumptions on the maximum covering times of all $\mathcal{M}_k$'s. However, since $\max_{k}  \{ \text{covering time}(\mathcal{M}_k)\} $ can be arbitrarily large,  $\max_{k} \{T_k\}$ can also be arbitrarily large. In that case, the convergence analysis in [Guo et al, 2019] is only asymptotic.
> > > > > >
> > > > > > Moreover, even focusing on each single MDP $\mathcal{M}_k$, the covering time can be exponentially large because it essentially reflects the problem of exploration in RL. Specifically, consider a chain of $N$ nodes where at any state $i \in \{1,\ldots N\}$, the chain is reset to state $1$ with probability $1/2$ and moves to state $i+1$ with probability $1/2$, regardless of the action taken. Suppose the initial state is the state $1$. Then, any policy would take $2^N$ steps to reach state $N$. Thus, in the worst case, the covering time can be $\Omega(\exp(S) ) $, which makes the asymptotic Q-learning algorithm very inefficient. Here $S$ is the capacity of the state space. A $O(\exp(S)) $ sample complexity upper bound seems very pessimistic.
> > > > > >
> > > > > > Currently, we are unaware of any exploration techniques that ensure a covering time that is polynomial in $S$, without making extra assumptions on the sampling model or the structure of the transition model. It seems that popular exploration techniques such as UCB or $\epsilon$-greedy both fail to ensure this. Nevertheless, having a `"generator" can easily satisfy the condition.
> > > > > >
> > > > > > We are wondering if the reviewer could point out a reference that proves a polynomial $(S)$ upper bound for the covering time under the general MDP, without having a generator or extra assumptions on the transition model. Nonetheless, we thank the reviewer for pointing out this subtle difference and we will revise accordingly.

---

> > > > > > > ### Comment · AnonReviewer1 · 2019-11-14
> > > > > > > **Further comments on the simulators**
> > > > > > >
> > > > > > > Yes the covering time L is kind of a limitation of (Even-Dar, 2003), which is also inherited by (Guo et al., 2019). I think that you are right that (Guo et al., 2019) chooses the second scenario, i.e., T_k is chosen according to the covering time of each M_k. However, I think T_k is not arbitrarily large as the covering time is controlled by the parameter p as specified in Lemma 8 there, which is the probability that the covering time is not bounded by some uniform bound L. And as a result they also obtain a non-asymptotic bound in Theorem 2, although it does not seem to be tight in any sense.
> > > > > > >
> > > > > > > That being said, you are right that covering time bound for general MDP is not easily characterized, let alone showing it's polynomial with high probability. But I would attribute this limitation to the study of general (G)MFG models and the choice of Q-learning in (Guo et al., 2019), as indeed on the high level you (Algorithm 1) basically replaces the Q-learning algorithm in (Guo et al., 2019) with the AC variant. The major contribution is addressing the difficulty of the drift term in the AC algorithm, and obtaining stronger theoretical results by building upon the recent works on AC for LQR settings (Yang et al., 2019). So I think it would be fair to say something like "restricting to the LQR setting, and utilizing and extending the stronger convergence results (cf. Yang et al., 2019) of AC compared to Q-learning in such a settings, we obtain stronger theoretical convergence results than (Guo et al., 2019); in particular, we can get rid of the implicit dependence on the covering times of the Q-learning algorithm in (Guo et al., 2019)".
> > > > > > >
> > > > > > > But anyway, as mentioned several times above, I think it is not really a big issue. And I do think our discussion clarifies several subtleties, which may also be beneficial to other readers and follow-up researchers to your work. Many thanks for detailed discussion!

---

> > > > > > > > ### Author Response · Authors · 2019-11-14
> > > > > > > > **Thanks for the discussion!**
> > > > > > > >
> > > > > > > > We really appreciate the prompt reply from the reviewer. We think that the reviewer's comments and suggestions for the claim of our contributions indeed make sense, and we will revise our work accordingly, hoping it may benefit other readers. We thank the reviewer again for the clarification and the detailed discussion with us!

---

### Decision · Program_Chairs · 2019-12-19

**Decision:**

Accept (Poster)

**Comment:**

The authors propose an actor-critic method for finding Nash equilibrium in linear-quadratic mean field games and establish linear convergence under some assumptions. There were some minor concerns about motivation and clarity, especially with regards to the simulator. In an extensive and interactive rebuttal, the authors were able to argue that their results/methods, which appear to be rather specialized to the LQ setting, offer insight/methods beyond the LQ setting.